# Put on your detective hat: What's wrong in this video? A Dataset for Error Recognition in Procedure Videos

## Abstract

Following step-by-step procedures is an essential component of various activities carried out by individuals in their everyday lives. These procedures serve as a guiding framework that helps achieve goals efficiently, whether assembling furniture or preparing a recipe. However, the complexity and duration of procedural activities inherently increase the likelihood of making errors. Understanding such procedural activities from a sequence of frames is a challenging task that demands an accurate interpretation of visual information and an ability to reason about the structure of the activity. To this end, we collected a new egocentric 4D dataset comprising 384 recordings (94.5 hrs) of people performing recipes in kitchen environments. This dataset consists of two distinct activity types: one in which participants adhere to the provided recipe instructions and another where they deviate and induce errors. We provide 5.3K step annotations and 10K fine-grained action annotations[1] for 20% of the collected data and benchmark it on the following tasks: error recognition, multi-step localization and procedure learning.

## 1 Introduction

*Remember when you prepared your favourite meal after a long day and missed adding that crucial ingredient and then lost your appetite after a few bites?* Such scenarios are quite common because performing long-horizon step-by-step procedural activities increases the likelihood of making errors. These errors can be harmless, provided they can be rectified with little consequence. Nonetheless, when the procedures in question pertain to the medical field or complex chemical experiments, the cost of errors can be substantial. Therefore, there is a pressing need for building AI systems that can guide users in performing procedural activities (Draper, 2021).

A key problem we need to solve in order to build such AI systems is *procedural activity understanding*, a challenging and multi-faceted task that demands interpreting what is happening —specifically, determining whether the person is following the procedure correctly or making an error, anticipating what will happen, and planning the course of action to accomplish the goal. For a system to interpret what is happening, it needs to recognize and segment actions while assessing the current state of the environment (Elhamifar & Huynh, 2020b; Yue Yang et al., 2021; Mengmeng Wang et al., 2021; Xudong Lin et al., 2022; Dvornik, Nikita et al., 2022). To anticipate future events, the system should be able to predict actions at the beginning of an interaction or even beforehand (Damen et al., 2021; Rohit Girdhar & Kristen Grauman, 2021). On the other hand, planning a sequence of actions requires the system to understand the possible outcomes of these interactions (Chang et al., 2019b; Henghui Zhao et al., 2022; Jing Bi et al., 2021). A number of datasets have been introduced to facilitate the understanding of procedural activity. Most of these datasets contain only normal videos of humans performing correct procedures. For an AI system to recognize errors in human procedures, datasets with error annotations are very necessary. Recently, there has been a significant rise in the number of procedural datasets containing errors that have been introduced, most of them primarily focussed on identifying and addressing the errors that occur during assembly (Table 1).

In this work, we present a novel dataset to aid AI systems that solve the procedural activity understanding task, focusing specifically on improving their ability to recognize and anticipate errors. We

---

[1]website: https://error-anonymous-dataset.github.io/ErrorAnonymous/

Table 1: **Ours vs Current Procedural Datasets (with and without errors)** Our dataset not only enhances the study of tasks outlined in procedural activity datasets in existing literature but also enables a systematic investigation of errors occurring during the performance of procedural activities.

| Errors | Dataset Name | Domain | Ego | Depth | Recorded | Error Labels | Errors Type | Videos | Hours | Tasks |
|---|---|---|---|---|---|---|---|---|---|---|
| ✗ | YouCook2(Zhou et al., 2017) | Cooking | ✗ | ✗ | ✗ | - | - | 2000 | 176 | 89 |
| | 50 Salads (Stein & McKenna, 2013) | Cooking | ✗ | ✓ | ✓ | - | - | 50 | 4.5 | 2 |
| | EGTEA Gaze+ (egt, 2018) | Cooking | ✓ | ✗ | ✓ | - | - | 86 | 29 | 7 |
| | MPII Cooking 2 (Rohrbach et al., 2015) | Cooking | ✗ | ✗ | ✓ | - | - | 273 | 27 | 67 |
| | EgoProceL (Bansal, Siddhant et al., 2022) | Assembly | ✓ | ✗ | ✓ | - | - | 329 | 62 | 16 |
| | Breakfast (Kuehne et al., 2014) | Cooking | ✗ | ✗ | ✓ | - | - | 1712 | 77 | 10 |
| ✓ | EgoTV (Rishi Hazra, 2023) | Simulated | ✓ | ✗ | - | ✓ | Intentional | 7673 | 168 | 540 |
| | Assembly101 (Fadime Sener et al., 2022) | Toy Assembly | ✓ | ✗ | ✓ | Partial* | Unintentional | 447 | 53 | 101 |
| | CSV (Qian et al., 2022) | Chemistry Lab | ✗ | ✗ | ✓ | ✗ | Intentional | 1940 | 11.1 | 14 |
| | HoloAssist (Wang et al., 2023) | Assembly* | ✓ | ✓ | ✓ | ✓ | Unintentional | 2221 | 166 | 350 |
| | IndustReal (Schoonbeek et al., 2024) | Toy Assembly | ✓ | ✓ | ✓ | ✓ | Int. and Unint. | 84 | 5.8 | 36 |
| | ATA (Ghoddoosian et al., 2023) | Toy Assembly | ✓ | ✗ | ✓ | ✓ | Intentional | 1152 | 24.8 | 3 |
| ✓ | Ours | Cooking | ✓ | ✓ | ✓ | ✓ | Int. and Unint. | 384 | 94.5 | 24 |

selected cooking as a domain that is sufficiently complex and encompasses different kinds of errors that are compounding in nature and completely alter the current state of the environment with no point of return. We decided to capture data from an egocentric view despite ego motions because it helps minimize occlusions more effectively than third-person videos.

This paper makes the following **contributions**: 1) We collected an egocentric 4D dataset that features individuals following recipes in kitchen settings. Our dataset includes two distinct types of activities: one where the participants precisely follow the given recipe guidelines and another where they deviate, making errors. 2) We provide annotations for (a) Start/End times for each step of the recipe, (b) Start/End times for each action/interaction for 20% of the collected data, and (c) Categorize and provide a detailed description of the error performed by a participant which enabled us to gather a comprehensive overview of different error types and their concise explanations. 3) We provide baselines for the following procedure understanding tasks: supervised error recognition, multi-step localization and procedure learning.

## 2 RELATED WORK

Our dataset is distinguished by four key features: (1) the inclusion of multi-step activities, (2) an egocentric viewpoint, (3) multimodal capabilities, and (4) a diverse set of errors. In Table 1, we offer a comparative analysis with existing datasets, and in the rest of the section, we elaborate on how our dataset is particularly relevant to the various tasks of interest.

**Error Recognition.** Given a video clip, error recognition involves identifying errors present in the clip. This task was initially introduced as mistake detection by Assembly-101 (Fadime Sener et al., 2022) and proposed a 3-class classification on the performed procedure to classify the clip as either correct, mistake, or correction. Anomaly detection, while closely related to error recognition, differentiates itself by utilizing static cameras and backgrounds to identify unusual or abnormal behavior. Our dataset, encompassing a variety of error types, including timing, preparation, temperature, technique, and measurement mishaps, provides researchers with a comprehensive view of error patterns in diverse situations. Cooking is a task that involves continuous changes in the shape and color of ingredients, unlike assembly tasks that usually lack variation. This unique characteristic of cooking activity makes our dataset particularly valuable for developing error recognition methods applicable to procedural tasks in the medical sector or that involve performing chemical experiments.

**Temporal Action Localization.** (TAL) aims to identify temporal boundaries in extended videos and classify each action instance. Broadly, TAL methodologies fall into two categories: two-stage and single-stage approaches. The two-stage method first generates action proposals and then classifies these actions. In contrast, the single-stage approach conducts simultaneous action localization and classification. Several datasets, such as ActivityNet (Fabian Caba Heilbron & Niebles, 2015), THUMOS14 (Jiang et al., 2014), Charades (Sigurdsson et al., 2016), MultiTHUMOS (Yeung et al., 2017), AVA (Gu et al., 2017), EPIC-KITCHENS (Damen et al., 2021), and Ego4D (Grauman et al., 2021), have significantly advanced the field of TAL. While our dataset may be smaller in comparison,

it offers a unique feature: it includes both normal actions and erroneous actions. This makes it especially valuable for evaluating TAL methods' robustness in handling actions with deviations.

**Procedure Learning.** is a two-part process where all video frames are first segregated into K significant steps. Then, a logical sequence of the steps necessary to complete the task is identified (Elhamifar & Huynh, 2020a; Huang et al., 2016; Chang et al., 2019a; Bojanowski et al., 2014; Sener & Yao, 2019; Zhou et al., 2018). Existing procedural activity datasets like CrossTask (Zhukov et al., 2019), COIN (Tang et al., 2019) are predominantly third-person videos; in this light, EgoProceL dataset (Bansal, Siddhant et al., 2022) was compiled from videos of CMU-MMAC (De la Torre et al., 2008), EGTEA (Fathi et al., 2011b), EPIC-Tents (Jang et al., 2019), MECCANO (Ragusa et al., 2020). We observe that our dataset features a greater average step length, posing a substantially more challenging problem for algorithms developed using existing egocentric procedure learning datasets.

## 3 DATA COLLECTION

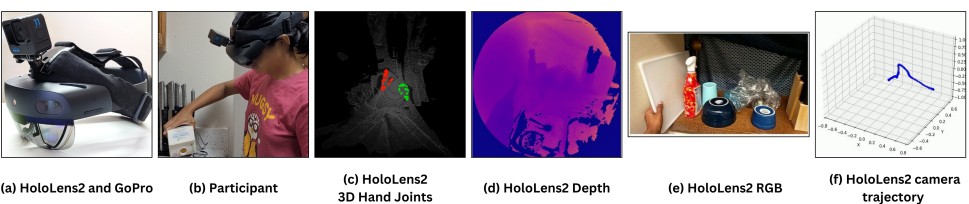

(a) HoloLens2 and GoPro    (b) Participant    (c) HoloLens2 3D Hand Joints    (d) HoloLens2 Depth    (e) HoloLens2 RGB    (f) HoloLens2 camera trajectory

Figure 1: (a-b) display the sensor configuration for recording that includes a GoPro mounted over a HoloLens and a participant making the recipe *Cucumber Raita*, and (c-f) display the synchronized data captured by the HoloLens2 including 3D hand joints, depth, RGB and camera trajectory.

**Sensors.** In order to gather activity data, we employed a combination of the GoPro Hero 11 camera, which was mounted on the user's head, and the HoloLens2 device. To facilitate data collection from the HoloLens2, including its depth sensor, IMU (Inertial Measurement Unit), front RGB camera, and microphone, we utilized a custom tool developed by (Dibene, Juan C. & Dunn, Enrique, 2022). Furthermore, we captured the processed head and hand tracking information provided by the HoloLens2 device. We offer data recorded from HoloLens2 and GoPro, presented separately for each recording. It is important to note that the data from GoPro and HoloLens2 are not synchronized. Figure 1 illustrates the data captured from HoloLens2.

**Recipes.** We curated a selection of 24 cooking recipes sourced from WikiHow (Table 8), specifically focusing on recipes with a preparation time of 30 minutes or less. These recipes encompassed a wide range of culinary traditions, showcasing the diversity of cooking styles across various cuisines. Our main goal was to identify potential errors that could occur when using different cooking tools to prepare recipes sampled from various cuisines.

**Task Graphs.** A task graph visually represents the sequential steps required to accomplish a given recipe. Each node in the task graph (for a recipe) corresponds to a step in a recipe, and a directed edge between a node $x$ and a node $y$ in the graph indicates that $x$ must be performed before $y$. Thus, a task graph is a directed acyclic graph, and a topological sort over it represents a valid completion of the recipe. In order to construct task graphs for our collection of 24 WikiHow recipes, we meticulously identified all the essential steps involved and established their inter-dependencies, thereby establishing a topological order of tasks (see Appendix F for details about constructed task graphs).

### 3.1 PROTOCOL

Our dataset was compiled by 8 participants across 10 distinct kitchens. Each participant selected ten recipes and recorded, on average, 48 videos across 5 different kitchens. During filming, all participants were required to ensure that they were alone in the kitchen and remove any items that could potentially identify them, such as personal portraits, mirrors, and smartwatches with portraits. The participants used a GoPro and a HoloLens2 to record and monitor their footage. Each participant was provided with a tablet-based recording interface accessible through a web browser. To ensure

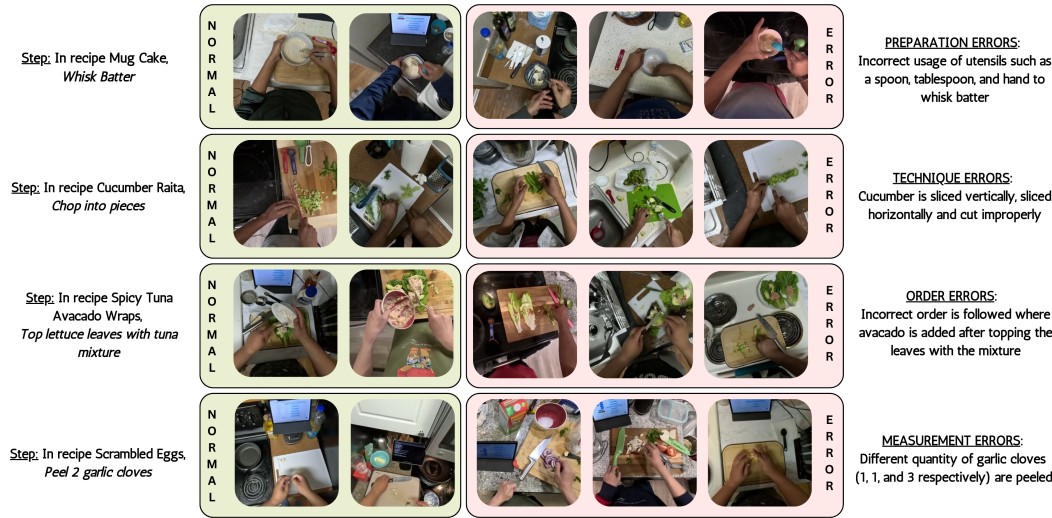

Figure 2: **Error Categories:** Each row displays frames captured from different recordings of recipes, highlighting both correct and erroneous executions, with a focus on specific types of errors.

optimal video quality, we asked the participants to configure the GoPro camera such that it captures videos in 4K resolution at 30 frames per second. The HoloLens2 device was programmed to stream RGB frames at a 360p resolution and a rate of 30 frames per second. It also streamed depth frames in Articulated Hand Tracking mode, referred to as "depth_ahat" mode. The device also streamed three separate IMU sensor data streams and spatial data, including both head and hand poses.

### 3.1.1 NORMAL RECORDINGS

A recording is categorized as a **normal recording** when it is captured as the participant accurately follows the procedure outlined in the recipe. Each participant in the study is tasked with selecting a recipe from the available options, which are scheduled within a kitchen setup using the recording interface. Subsequently, they are presented with one of the pre-established topological orders of the recipe, as determined by the previously constructed task graphs (see Appendix F). Participants then proceed to follow the provided task graph, commencing from the beginning and progressing through each step in accordance with its dependencies and designated timing (see Figure **??**).

### 3.1.2 ERROR RECORDINGS

A recording is termed an **error recording** when it is captured while the individual deviates from the recipe's procedure, thereby inducing errors. Following the terminology used in scientific disciplines such as neuroscience (Chevignard et al., 2010) and chemistry, we will refer to deviations from procedures as *errors*. Note that the term "errors" used here is equivalent to what is commonly called "mistakes" in the AI community (c.f. (Fadime Sener et al., 2022)). Following (Chevignard et al., 2010; Finnanger et al., 2021; Fogel et al., 2020), we classified common errors performed during a cooking activity into the following categories (1) Preparation Error, (2) Measurement Error, (3) Technique Error, (4) Timing Error, (5) Temperature Error, (6) Missing Steps, and (7) Ordering Errors (see Figure 18 in Appendix). We also provide visual illustrations in Figure 2, showcasing the categorization of videos into normal and error recordings.

We devised and implemented three strategies for the participants to follow. Each participant was asked to pick a strategy for performing the recipe in a particular environment and was accordingly guided in preparing for their performance. We list the strategies presented to the participants here (1) **Pre-prepared error scripts**: In this strategy, participants were given pre-prepared error scripts with missing steps and ordering errors. (2) **Prepare error scripts**: Once participants chose this strategy, they were given a web-based interface to create an error script for each error recipe recording and displayed the modified error script on a tablet, enabling participants to perform according to

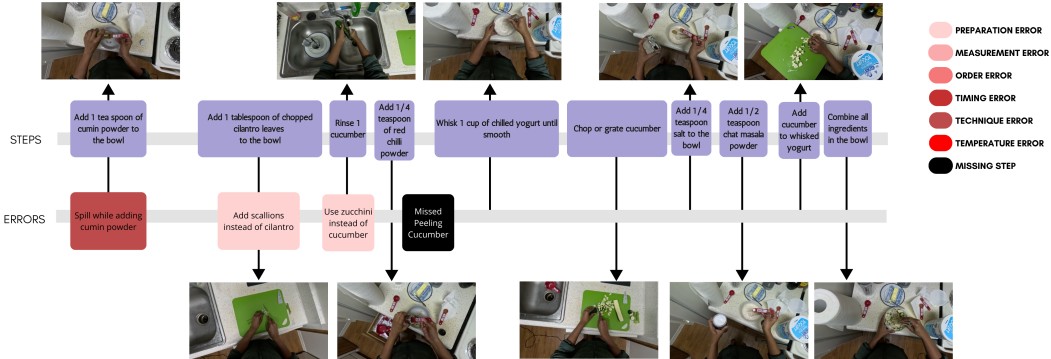

Figure 3: Displays the timeline of recipe steps and recorded errors while preparing the recipe *Cucumber Raita*. Three of four errors were intentional, but the participant unintentionally missed the *Peeling* step. In annotations, we provide step start and end times along with a description and categorization of the error performed during that step.

their modified error scripts (3) **Impromptu**: During the later stages of the recording process, we implemented a strategy where participants were asked to induce errors as they perform the recipe. Following the completion of each recording, participants were given access to a web-based interface to update errors they made during each step. Although we developed a process to capture intentional errors by preparing error scripts, many errors were unintentional (Figure 3 presents such an example).

## 3.2 DATA ANNOTATION

Our annotations comprise (1) Annotations for coarse-grained actions or steps, providing the start and end times for each step within the recorded videos. (2) To support learning semi/weakly supervised approaches for action recognition and action anticipation, we have provided fine-grained action annotations for 20% of the recorded data. These annotations include the start and end times for each fine-grained action. (3) We have also categorized and provided error descriptions for the induced errors. These error descriptions are associated with the corresponding step in the provided annotations, allowing for a comprehensive understanding of the errors. Figures 3, **??** describe the granularity of different categories of annotations provided. To ensure high-quality annotations for our data, we ensured that each recording was annotated by the person who recorded the video and then reviewed by another. The reviewer was asked to double-check that all errors made by the participant in the recording were included in their corresponding step annotations.

**Coarse-Grained Action/Step Annotations.** We designed an interface for performing step annotations in Label Studio [2]. Each annotator is presented with this interface to mark the start and end times for each step. Our steps are significantly longer than a single fine-grained action and encompass multiple fine-grained actions necessary for performing the described step. For example, in order to accomplish the step *{Chop a tomato}*, we include the following (1) Pre-conditional actions of *{opening refrigerator, grabbing a polythene bag of tomatoes, taking a tomato, placing the tomato on cutting board, close fridge}* (2) Post-conditional actions of *{placing down the knife, grabbing the polythene bag of tomatoes, open fridge and place the bag in the fridge}*. Table 2 summarizes and compares coarse-grained action/step annotations across relevant datasets.

Table 2: Comparison of coarse-grained action or step annotations across related datasets. Here, $\mathcal{T}_{avg}$ represents the avg. duration for each video, $\mathcal{N}^{seg}$ shows the total number of segments, $\mathcal{N}^{seg}_{avg}$ reveals the avg. number of segments per video, and $\mathcal{T}^{seg}_{avg}$ shows the avg. duration for all segments.

| Dataset | $\mathcal{T}_{avg}$ (min) | $\mathcal{N}^{seg}$ | $\mathcal{N}^{seg}_{avg}$ | $\mathcal{T}^{seg}_{avg}$ (sec) |
|---|---|---|---|---|
| 50Salads | 6.4 | 899 | 18 | 36.8 |
| Breakfast | 2.3 | 11,300 | 6.6 | 15.1 |
| Assembly 101 | 7.1 | 9523 | 24 | 16.5 |
| CSV | 0.2 | 18488 | 9.53 | 2.1 |
| HoloAssist | 4.48 | 15927 | 7.17 | 39.3 |
| **Ours (Total)** | **14.8** | 5300 | 13.8 | **52.78** |

**Fine-Grained Action Annotations.** Inspired by the pause-and-talk narrator (Damen et al., 2020), we have designed and developed a web-based tool for fine-grained action annotation that utilizes

---

[2]https://labelstud.io/

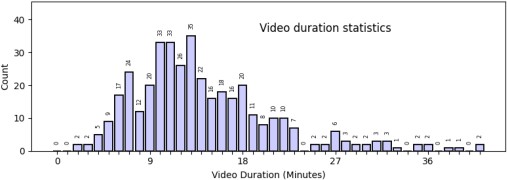 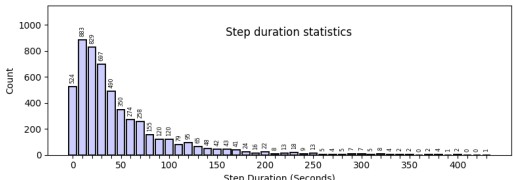

Figure 4: Displays duration statistics for videos and individual steps

Whisper (Radford et al., 2023) for speech-to-text translation. We will release the developed web-based annotation tool as part of our codebase upon acceptance.

## 4 BASELINES

We provide baselines for the following tasks (1) Error Recognition, (2) Multi-Step Localization, and (3) Procedure Learning. In our approach to Error Recognition and Multi-Step Localization tasks, we utilized state-of-the-art pre-trained models originally developed for video recognition tasks to extract relevant features. Once these features were extracted, we proceeded to train distinct heads, each tailored to address a specific task at hand. We used 3D-ResNet (Hara et al., 2017), SlowFast (Feichtenhofer et al., 2019), X3D (Feichtenhofer, 2020), VideoMAE (Tong et al., 2022) and Omnivore (Girdhar et al., 2022) as our backbones for extracting features.

### 4.1 ERROR RECOGNITION

**Supervised Error Recognition.** Error Recognition aims to identify errors within specific segments of a long, untrimmed video that depicts a procedural activity. Here, each segment represents a step in the procedure. We note that the task is challenging due to the presence of a diverse set of errors (summarized in Figures 18 and 19). We set up Error Recognition as a supervised binary classification task, categorizing each step into one of two classes: {*error*, *normal*}. We evaluated our trained models using standard

Table 3: Supervised Error Recognition

| Baseline | Precision | Recall | F1 Score | AUC Score |
|---|---|---|---|---|
| **3D ResNet** | 76.74 | 14.54 | 24.44 | 0.78 |
| **Slowfast** | 64.42 | 29.52 | 40.48 | 0.78 |
| **X3D** | 52.78 | 16.74 | 25.42 | 0.72 |
| **VideoMAE** | 75.34 | 25.7 | 38.33 | 0.82 |
| **Omnivore** | 68.24 | 44.49 | 53.87 | 0.84 |

metrics for binary classification such as Precision, Recall, F1 Score, and AUC Score, and presented results in Table 3. Firstly, we used data constructed by splitting based on recordings (as described in G.1) to prepare training, validation and testing data. Then, we compiled annotated video segments corresponding to the steps of the recipes for each part to prepare a comprehensive dataset which includes 4026 training segments (with 1283 errors), 531 validation segments (179 errors), and 743 testing segments (227 errors). We used the error categorization of each step to generate binary labels for the compiled video segments.

We utilized pre-trained video recognition models to extract features. However, to maintain a fixed-size input to these models, we divided each segment into 1-second sub-segments. Each sub-segment was given the same class label as its parent segment. We used the extracted features to train a neural network with a single hidden layer with ReLU activation and a sigmoid output node. We assigned the majority class among the sub-segments to the entire segment during the inference phase. We trained all classifiers on an NVIDIA A40 GPU using Adam optimizer and set the learning rate to 0.001.Table 3 presents the results obtained for error recognition on our dataset. We observe that our Omnivore(Girdhar et al., 2022)-based model achieves the best recall, F1 and AUC scores. However, the scores are pretty low, which underscores the challenging nature of the task. We also present qualitative results of trained classifiers in figure 5. [3]

---

[3]We also developed methods for solving the zero-shot error recognition task (namely, training data contains only normal recordings and test data has both error and normal recordings) by adapting anomaly detection methods in the literature. However, we found that these methods perform poorly and are only slightly better than random (results are presented in the supplement). These results suggest that zero-shot error recognition is quite challenging and will require methods that seek to understand the context and meaning of errors.

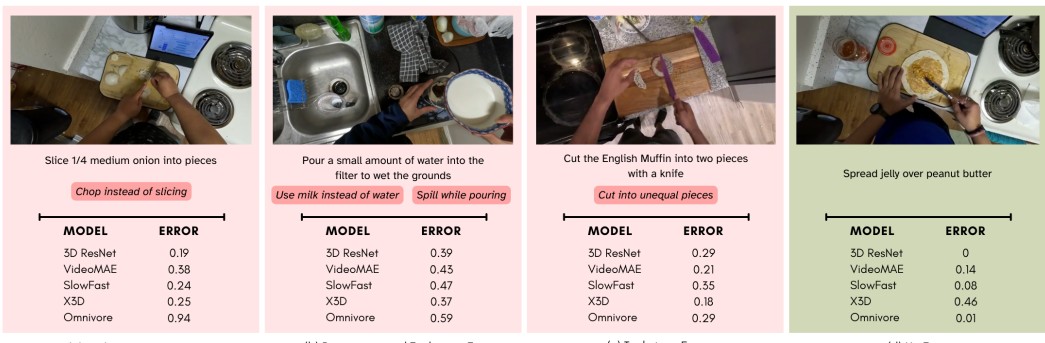

Figure 5: Displays error probabilities predicted by trained classifiers on 4 segments of the video (3 error segments and 1 normal segment) sampled from the compiled test dataset. Although our omnivore-based model outperforms the rest in classifying error segments, we note that all models are adept at distinguishing normal video segments.

**Early Error Recognition.** In this task, we aim to identify errors within segments of a procedural activity when only the first half of the segment is provided as input to the model. Thus, we re-use the datasets compiled for supervised error recognition and train task prediction heads. The results in Table 4 are consistent with supervised error recognition, where our omnivore-based model outperforms other models. We note that the scores for early error recognition are generally lower compared to the error recognition setting, indicating that recognizing errors with less information is a significantly harder setting. We conjecture that to improve these scores significantly, one must employ methods that seek to (semantically) understand the context, meaning, and cause of various errors.

Table 4: Early Error Recognition

| Baseline | Precision | Recall | F1 Score | AUC Score |
|---|---|---|---|---|
| **3D ResNet** | 72.73 | 3.52 | 6.72 | 0.77 |
| **Slowfast** | 72.41 | 9.25 | 16.41 | 0.75 |
| **X3D** | 58.33 | 3.08 | 5.86 | 0.73 |
| **VideoMAE** | 70 | 6.54 | 11.97 | 0.82 |
| **Omnivore** | 72.09 | 13.66 | 22.96 | 0.83 |

**Error Category Recognition.** In this approach, we frame Error Category Recognition as a binary classification task, discerning between errors and non-errors across all error types. We iterate through each error type and construct a dataset where it is designated as the error class, while all other error categories and correct instances are categorized as correct. Table 11 presents the performance metrics for five models, each trained using a distinct pre-trained feature extractor corresponding to different error categories. Despite achieving high accuracy scores, a closer examination of the recall values reveals limitations in the models' ability to identify different types of errors accurately. Additionally, this analysis helps determine the relative hardness of detecting different types of errors.

Table 5: Error Category Recognition

| Method Name | Technique Error | | | | Preparation Error | | | | Measurement Error | | | | Temperature Error | | | | Timing Error | | | |
|---|---|---|---|---|---|---|---|---|---|---|---|---|---|---|---|---|---|---|---|---|
| | Acc. | Prec. | Rec. | F1 | Acc. | Prec. | Rec. | F1 | Acc. | Prec. | Rec. | F1 | Acc. | Prec. | Rec. | F1 | Acc. | Prec. | Rec. | F1 |
| **3D ResNet** | 88.56 | 27.91 | 18.18 | 22.02 | 89.77 | 9.30 | 9.76 | 9.52 | 88.43 | 6.98 | 6.12 | 6.52 | 93.14 | 0.00 | 0.00 | 0.00 | 91.52 | 6.98 | 11.54 | 8.70 |
| **Slowfast** | 82.50 | 19.23 | 30.30 | 23.53 | 83.45 | 10.58 | 26.83 | 15.17 | 82.10 | 9.62 | 20.41 | 13.07 | 85.20 | 0.96 | 12.50 | 1.79 | 84.12 | 5.77 | 23.08 | 9.23 |
| **X3D** | 83.31 | 9.72 | 10.61 | 10.14 | 87.21 | 12.50 | 21.95 | 15.93 | 85.33 | 8.33 | 12.24 | 9.92 | 89.23 | 0.00 | 0.00 | 0.00 | 87.62 | 4.17 | 11.54 | 6.12 |
| **VideoMAE** | 84.39 | 19.18 | 22.22 | 20.59 | 86.99 | 13.70 | 27.03 | 18.18 | 84.39 | 8.22 | 12.77 | 10.00 | 88.58 | 1.37 | 12.50 | 2.47 | 87.14 | 6.85 | 19.23 | 10.10 |
| **Omnivore** | 78.20 | 17.57 | 39.39 | 24.30 | 80.22 | 14.19 | 51.22 | 22.22 | 78.33 | 12.16 | 36.73 | 18.27 | 79.81 | 2.03 | 37.50 | 3.85 | 79.00 | 6.08 | 34.62 | 10.34 |

## 4.2 MULTI STEP LOCALIZATION

**Description.** Given an untrimmed, long video that captures a procedural activity, multi-step localization aims to determine each step's start and end frames and classify them. We've framed the supervised multi-step localization task as an instance of a supervised temporal action localization (TAL) problem. This setup is particularly challenging as our dataset encompasses both normal actions and those with deviations, termed "Technique Errors" (refer to 18), and the duration of steps in our dataset exceeds that of actions in benchmark datasets used for TAL (Table 2). We employ standard metrics used in TAL methods to evaluate trained models and present results. These metrics include temporal Intersection over Union (tIoU), mean Average Precision (mAP), and Recall at x (R@x).

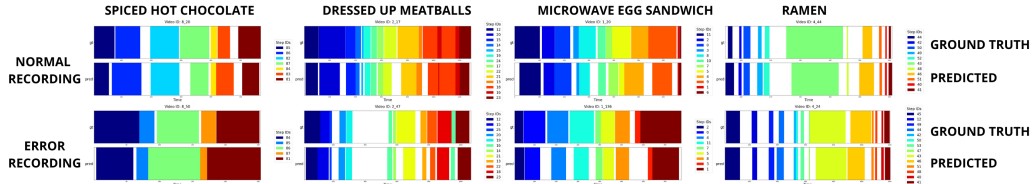

Figure 6: Compares the multi-step localization results for 4 recipes trained using the omnivore-based model on data split constructed using recording environments as a criterion. Each normal/error recording is sampled from the test set containing only normal/error recordings. Top part shows the ground truth segments and the bottom part shows the predicted segments.

**Implementation Details.** We used three different splits of the dataset constructed based on the recording environments ($\mathcal{E}$), recording persons ($\mathcal{P}$) and recordings ($\mathcal{R}$) (as described in G.1). For each of these splits, we extracted features from chosen pre-trained video recognition models. For each of the resulting 12 datasets, we trained an ActionFormer head (Zhang et al., 2022). We modified the default configuration file and set the following hyper-parameters: num_classes to 353, input_dim to 1024, max_seq_len to 4096, learning rate to 0.0001 and trained all 12 models for 16 epochs. During inference, we have further split each test set into two distinct sets where one set contains only normal recordings ($\mathcal{T}_n$), and the other set contains only error recordings ($\mathcal{T}_e$).

Table 13 presents detailed results obtained on the evaluation of trained models on distinct test sets constructed based on the presence of errors. We observe that among all the feature extractors used as backbones for training the ActionFormer head, Omnivore performs much better. In Appendix G.3, we present further benchmarking results where we evaluate models on a combined test set and perform an ablation study on the performance of trained models for features extracted using varying lengths. It can be seen that all the models perform significantly worse on test sets constructed using only error recordings compared to the ones constructed using normal ones. We also present qualitative results in figure 6.

Table 6: Multi Step Localization.

| $\mathcal{B}$ | $\mathcal{D}$ | $\mathcal{T}$ | $\mathcal{I}_t = 0.1$ | | | $\mathcal{I}_t = 0.3$ | | | $\mathcal{I}_t = 0.5$ | | |
|---|---|---|---|---|---|---|---|---|---|---|---|
| | | | mAP | R@1 | R@5 | mAP | R@1 | R@5 | mAP | R@1 | R@5 |
| **3D ResNet** | $\mathcal{E}$ | $\mathcal{T}_n$ | 21.4 | 39.51 | 54.39 | 20.07 | 35.69 | 50.74 | 17.1 | 29.36 | 45.3 |
| | | $\mathcal{T}_e$ | 9.74 | 15.31 | 23.2 | 8.31 | 12.69 | 21.45 | 6.22 | 9.08 | 16.57 |
| | $\mathcal{P}$ | $\mathcal{T}_n$ | 19.57 | 35.93 | 49.39 | 18.68 | 33.2 | 47.44 | 15.99 | 27.58 | 43.22 |
| | | $\mathcal{T}_e$ | 13.82 | 27.14 | 39.57 | 12.94 | 23.4 | 37.32 | 10.88 | 19.21 | 33.64 |
| | $\mathcal{R}$ | $\mathcal{T}_n$ | 20.03 | 35.18 | 47.57 | 19.15 | 32.34 | 46.09 | 16.69 | 27.04 | 41.52 |
| | | $\mathcal{T}_e$ | 13.22 | 25.96 | 37.84 | 12.48 | 23.47 | 36.07 | 10.8 | 19.5 | 31.76 |
| **Slowfast** | $\mathcal{E}$ | $\mathcal{T}_n$ | 22.48 | 39.57 | 54.14 | 20.86 | 35.97 | 50.51 | 17.2 | 28.28 | 44.75 |
| | | $\mathcal{T}_e$ | 10.11 | 16.16 | 23.32 | 9 | 13.02 | 20.39 | 7.53 | 9.54 | 15.83 |
| | $\mathcal{P}$ | $\mathcal{T}_n$ | 23.12 | 36.55 | 50.45 | 22.09 | 34.09 | 49.11 | 19.24 | 28.93 | 45.12 |
| | | $\mathcal{T}_e$ | 14.78 | 26.68 | 39.97 | 14.14 | 24.73 | 37.71 | 12.56 | 21.76 | 34.37 |
| | $\mathcal{R}$ | $\mathcal{T}_n$ | 22.78 | 36.46 | 50.1 | 22.03 | 34.35 | 48.13 | 19.62 | 30.08 | 44.88 |
| | | $\mathcal{T}_e$ | 14.11 | 27.52 | 39.19 | 13.34 | 24.91 | 37.19 | 11.9 | 21.53 | 32.39 |
| **VideoMAE** | $\mathcal{E}$ | $\mathcal{T}_n$ | 24.44 | 38.22 | 52.48 | 22.97 | 34.77 | 49.51 | 18.67 | 28.57 | 42.68 |
| | | $\mathcal{T}_e$ | 7.53 | 13.54 | 20.52 | 6.93 | 11.4 | 18.36 | 5.63 | 8.55 | 15.13 |
| | $\mathcal{P}$ | $\mathcal{T}_n$ | 26.78 | 37.43 | 46.28 | 25.68 | 34.79 | 44.6 | 22.02 | 29.43 | 39.81 |
| | | $\mathcal{T}_e$ | 16.98 | 27.43 | 37.76 | 16.46 | 25.53 | 36.03 | 14.64 | 22.03 | 32.07 |
| | $\mathcal{R}$ | $\mathcal{T}_n$ | 26.27 | 37.15 | 46.93 | 24.71 | 34.06 | 45.03 | 21.51 | 29.36 | 40.44 |
| | | $\mathcal{T}_e$ | 15.43 | 25.94 | 33.97 | 14.44 | 23.23 | 32.35 | 12.96 | 19.83 | 28.99 |
| **Omnivore** | $\mathcal{E}$ | $\mathcal{T}_n$ | 34.65 | 47.91 | 60.63 | 33.06 | 44.77 | 58.36 | 28.59 | 38.38 | 51.9 |
| | | $\mathcal{T}_e$ | 12.51 | 19.6 | 27.06 | 11.66 | 17.54 | 24.45 | 9.94 | 14.63 | 20.96 |
| | $\mathcal{P}$ | $\mathcal{T}_n$ | 32.5 | 44.45 | 52.47 | 31.13 | 41.53 | 50.91 | 28.39 | 37.03 | 47.97 |
| | | $\mathcal{T}_e$ | 21.28 | 31.51 | 40.93 | 20.12 | 28.81 | 39.6 | 18.08 | 24.96 | 36.77 |
| | $\mathcal{R}$ | $\mathcal{T}_n$ | 30.22 | 42.43 | 52.11 | 28.94 | 39.47 | 50.49 | 25.15 | 32.65 | 46.51 |
| | | $\mathcal{T}_e$ | 19.54 | 31.28 | 41.24 | 18.4 | 28.66 | 39.33 | 16.27 | 24.28 | 35.35 |

### 4.3 PROCEDURE LEARNING

**Description.** Procedure learning entails identifying relevant frames across videos of activity and estimating the sequential steps required to complete a given task. To benchmark procedure learning, we employed normal recordings from our dataset and assessed the performance of recently proposed methods (Bansal, Siddhant et al., 2022; Dwibedi et al., 2019).

**Implementation Details.** We followed the setup as described in the work of (Bansal, Siddhant et al., 2022) and trained the embedder networks for each recipe. Specifically, we train two networks, one using the Cycleback Regression loss ($\mathcal{C}$) proposed by (Dwibedi et al., 2019) and the other using a blend of two loss functions: Cycleback Regression loss ($\mathcal{C}$) and Contrastive - Inverse Difference Moment loss ($\mathscr{C}$). The combined loss function is expressed as $\mathcal{C} + \lambda \times \mathscr{C}$, where $\lambda$ is a hyperparameter. (we set it to 0.5). We note that we only train embedded networks using loss functions from these methods and retain the Pro-Cut Module for assigning frames to key steps. We adhered to the hyperparameter settings specified in the original paper to train the embedder network. Utilizing an A-40 GPU, the entire training process was completed in approximately three hours. The results are presented in Table 16 we noticed a significant decline in performance compared to the results from all other datasets reported in the paper (Bansal, Siddhant et al., 2022). Given that our dataset features videos with notably longer key step lengths (as indicated in Table 2), we attribute this drop in performance primarily to this distinguishing characteristic.

Table 7: **Procedure Learning.** Here, $\mathcal{P}$ represents precision, $R$ represents recall, and $I$ represents IOU.

| Recipe | Random | | | $\mathcal{M}_1$ (Dwibedi et al., 2019) | | | $\mathcal{M}_2$ (Bansal, Siddhant et al., 2022) | | |
|---|---|---|---|---|---|---|---|---|---|
| | $\mathcal{P}$ | $\mathcal{R}$ | $\mathcal{I}$ | $\mathcal{P}$ | $\mathcal{R}$ | $\mathcal{I}$ | $\mathcal{P}$ | $\mathcal{R}$ | $\mathcal{I}$ |
| BlenderBananaPancakes | 7.40 | 3.83 | 2.26 | 12.65 | 9.50 | 5.16 | 15.54 | 9.96 | 5.72 |
| Coffee | 6.54 | 3.87 | 2.17 | 13.68 | 9.91 | 5.49 | 15.76 | 10.25 | 5.63 |
| MugCake | 5.45 | 4.00 | 2.12 | 16.12 | 12.95 | 6.87 | 10.32 | 8.85 | 4.40 |
| PanFriedTofu | 5.35 | 3.97 | 1.54 | 8.86 | 10.39 | 3.75 | 9.34 | 12.44 | 3.87 |
| Pinwheels | 6.54 | 4.28 | 2.13 | 13.58 | 11.96 | 5.92 | 16.08 | 13.06 | 7.05 |
| **Average of 24 recipes** | **7.61** | **3.92** | **2.22** | **15.62** | **10.85** | **5.78** | **15.78** | **10.68** | **5.82** |

## 5 DISCUSSION, SUMMARY AND FUTURE WORK

In this paper, we have introduced a large egocentric dataset for procedural activities. Our dataset consists of synchronized egocentric views, audio, and depth information specifically designed for tasks such as Temporal Action Segmentation, 3D activity analysis, Procedure Learning, Error Recognition, Error Anticipation, and more. Additionally, we have provided benchmarks for error recognition and Procedure Learning. While current methods have yielded promising outcomes, they continue to struggle to tackle these challenges adequately with satisfactory results, as demonstrated by our experimental assessment. This indicates the need for further exploration in this domain.

**Limitations.** We intend to capture deviations observed while performing a procedural activity from an egocentric view. First, we note that this type of data cannot be compiled from crowd-sourced platforms. This left us to capture participant data while performing procedural activities. Second, by the nature of the problem, errors that occur when performing procedural activities are combinatorial and can have a compounding effect. Thus, our work has the following limitations: (1) For each activity, the errors captured and presented in the dataset form a subset of the whole combinatorial space; (2) Capturing 4D data in real kitchen environments posed logistical and equipment training challenges. As a result, we were compelled to limit the data collection to a specific geographic area. (3) Compared to datasets curated from the crowd-sourced platforms used for tasks like action/activity recognition, temporal action segmentation, etc., the presented work comprises fewer recipes.

Our work opens up several avenues for future work. First, an exciting direction is the extension of the dataset to include activities from other domains. By incorporating tasks such as performing chemical experiments or executing hardware-related activities (e.g., working with cars or computer parts), the dataset can encompass a wider range of activities and provide insights into error patterns in diverse real-world scenarios. Second, the dataset can be used to compare and develop methods for solving various tasks such as transfer learning, semantic role labelling, video question answering, long video understanding, procedure planning, improving task performance, reducing errors, etc.

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

APPENDICES

## A    MOTIVATION FOR COLLECTING A NEW DATASET WITH ERRORS

Current datasets that study procedural tasks, such as GTEA (Fathi et al., 2011a), Breakfast (Kuehne et al., 2014), CMU-MMAC (De la Torre et al., 2008), 50 Salads (Stein & McKenna, 2013), COIN (Tang et al., 2019), CrossTask (Zhukov et al., 2019), ProceL (Elhamifar & Naing, 2019), EgoProceL(Bansal, Siddhant et al., 2022), Assembly101 (Fadime Sener et al., 2022), and HowTo100M (Miech et al., 2019), encompass temporal variation in the order of the steps performed. But these datasets are predominantly sourced from crowd-sourced online platforms, which results in the videos often containing drastically different steps, with alterations impacting more than 30% of the content.

Our interest lies in understanding errors induced by deviating from the given instruction set. To this end, we require two types of videos —, normal ones that closely follow the instructions and error videos that depict deviations. Moreover, we aim to capture these videos from an ego-centric perspective to minimize the occlusions typical in third-person videos. The absence of such specific video resources on crowd-sourced platforms led us to curate a dataset embodying all our desired characteristics.

In subsequent sections, we first shed light on offered splits for the datasets, then provide detailed descriptions of the performed beIn the following sections, we will start by explaining the data splits that we have used for the datasets. After that, we will provide a detailed description of our benchmarking process and take a closer look at our data collection process. Our data collection consists of three distinct phases: pre-production, production, and post-production. We will then provide a comprehensive overview of the tasks associated with each phase.

## B  DATA COLLECTION—PRE-PRODUCTION PHASE

Our goal is to collect data that can help detect, segment, and analyze errors that may happen during long procedural tasks. To achieve this, we need to answer the following questions: (1) **What to record**; namely, choose the domain and tasks (recipes). (2) **How to record**; namely, choose the sensors and design a data/video capturing system. (3) **Whom to record**; namely, select participants and train them so that they can competently record the videos.

### B.1  WHAT TO RECORD?

Current popular procedural activity understanding datasets encompass recorded and curated ones from crowd-sourced online platforms. Amongst the recorded datasets, Breakfast (Kuehne et al., 2014), 50Salads (Stein & McKenna, 2013), CMU-MMAC (De la Torre et al., 2008), and GTEA (Fathi et al., 2011a) capture people performing cooking activities, and Assembly-101 (Fadime Sener et al., 2022), EPIC-TENTS (Jang et al., 2019) and MECCANO (Ragusa et al., 2020) capture people performing activities related to assembly of toys, tents and lego blocks, respectively. Curated datasets like COIN (Tang et al., 2019), CrossTask (Zhukov et al., 2019), and HowTo100M (Miech et al., 2019) encompass a wide variety of activities from different domains. We introduced a new perspective on understanding procedural activities from the lens of errors made while performing procedural tasks. We embark on an investigation into this new idea by choosing cooking as the domain of interest. This careful choice stems from the fact that activities in cooking often encompass complex procedures and provide an opportunity to capture a plethora of potential errors that are predominantly benign.

#### B.1.1  RECIPES & TASK GRAPHS

We have carefully selected 24 diverse recipes from WikiHow (see Table 8) that represent various cuisines and require the use of different culinary tools during preparation.

Each recipe in our selected set can be subdivided into several atomic steps, where each step involves performing a specific sub-task in the recipe. In general, most recipes available on the web list these sub-tasks in a specific order. However, common sense tells us that each recipe can often be described by a partial order over the sub-tasks rather than a total order.

More formally, we use a task graph to represent the partial order over the steps. Each node in the task graph corresponds to a step and a directed edge between node $i$ and node $j$ denotes that step $i$ must be done before step $j$ (namely $i$ is a pre-condition of $j$). For our selected recipes, the corresponding task graphs are directed acyclic graphs, and therefore a topological sort over them is a valid execution of the recipe. Our task graphs also include two dummy nodes, "START" and "END", which denote the start and end of recipes respectively. The dummy nodes ensure that our task graphs always have one start node and one terminal node.

To simplify the complexity of a recipe, we have adopted a technique that uses a flow graph structure (Yamakata et al., 2020) to represent the dependencies between steps (think of it like a flowchart but designed for recipes). This approach helps us establish a more precise connection between actions and their consequences. Using an action-centric graph, we emphasize the steps involved in the

Table 8: presents 24 selected recipes categorized based on the type of required heating equipment.

| Heating Instrument | Recipe | Heating Instrument | Recipe |
|---|---|---|---|
| Kettle | Coffee | Nothing | Pinwheels |
| Microwave | Breakfast Burritos | | Spicy Tuna Avocado Wraps |
| | Butter Corn Cup | | Tomato Mozzarella Salad |
| | Cheese Pimiento | Pan | Blender Banana Pancakes |
| | Dressed Up Meatballs | | Broccoli Stir Fry |
| | Microwave Egg Sandwich | | Caprese Bruschetta |
| | Microwave French Toast | | Herb Omelet with Fried Tomatoes |
| | Microwave Mug Pizza | | Pan Fried Tofu |
| | Mug Cake | | Sauteed Mushrooms |
| | Ramen | | Scrambled Eggs |
| | Spiced Hot Chocolate | | Tomato Chutney |
| Nothing | Cucumber Raita | | Zoodles |

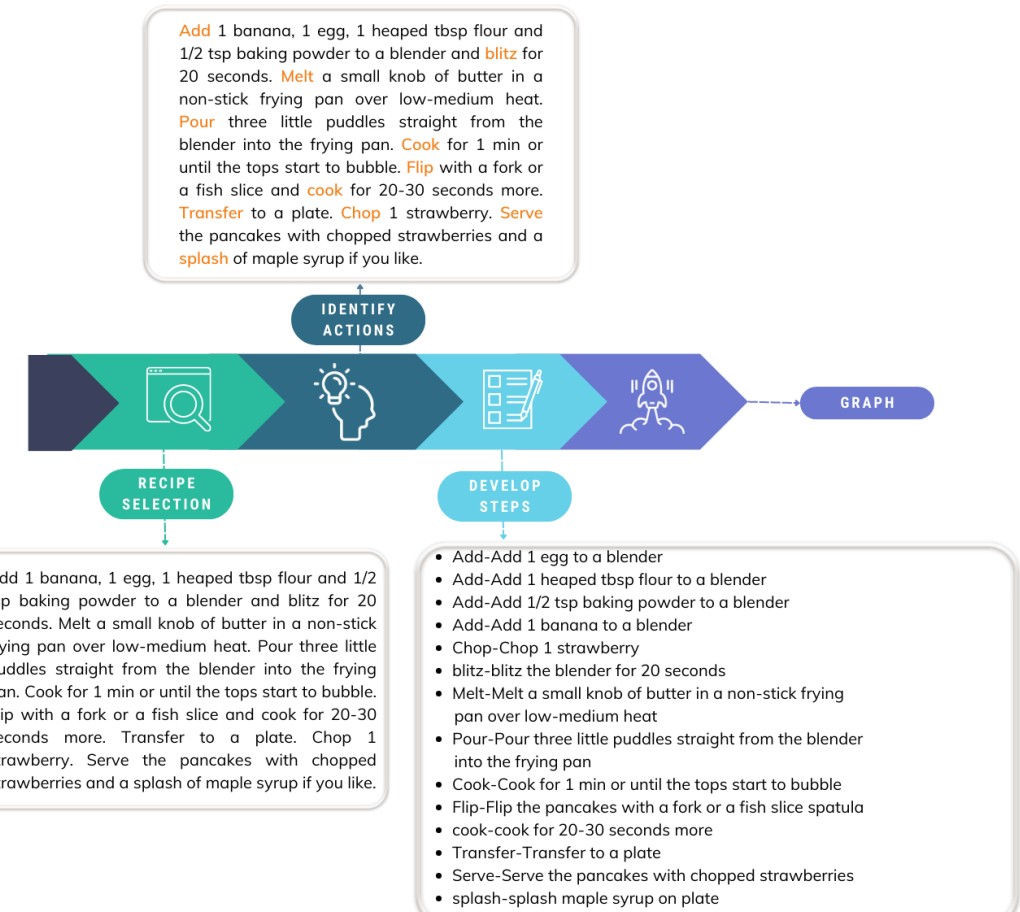

Figure 7: This figure illustrates, with an example, the three steps followed in the creation of an action-centric graph for a recipe. Given the recipe text as shown in *recipe selection*, we identify and mark all the actions necessary for the execution of the recipe as shown in *identify actions*. Once these actions are identified, we develop them into steps (as shown in *develop steps*) where each step contains only one of the actions identified before. These steps are used to construct an action-centric graph for the recipe resulting in a structure as depicted in 8

procedure and illustrate the sequence of operations in an easy-to-understand manner. Each action directly impacts subsequent ones, demonstrating the dependencies between tasks.

We illustrate the process we used to convert a recipe to a task graph using the recipe *Blender Banana Pancakes* (see figures 7 and 8 for a visual guide). Given the recipe description, we first identify all the actions necessary to complete the recipe and develop steps based on the identified actions, where each step contains only one among the identified actions, as shown in figure 7. After we develop steps, we use a relationship annotation tool[4] to represent the implicit order constraints amongst the developed steps. The creation of action-centric graphs serves multiple purposes. These graphs can be utilized to prepare recipe scripts with various orders while still strictly adhering to the constraints present in the graph. Moreover, given a recording, the graph can be used to verify if the individual followed the correct sequence of actions based on the inherent graph structure.ipe *Blender Banana Pancakes*, the developed steps from 7, when represented as an action-centric graph, result in figure 8.

In future, we envision using our dataset to construct more fine-grained task graphs where the meaning of the steps is taken into account as well as how the step changes the environment (post-condition for a step). In literature, different methods have been proposed to illustrate procedural

---

[4]https://www.lighttag.io/

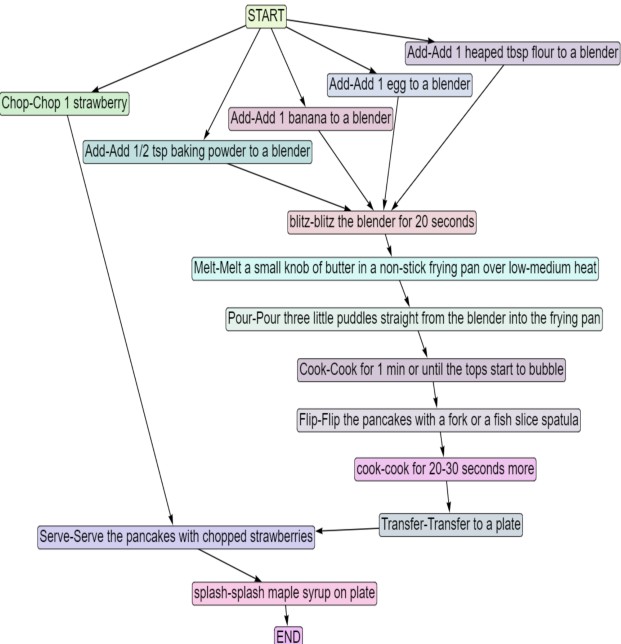

Figure 8: This graph displays the implicit dependency structure of the recipe **Blender Banana Pancakes** where the content of each node can be interpreted as {{action}-{step}} where {action} presents the description of the necessary action to be performed, and {step} presents the description as presented in the recipe text that encompasses the action, ingredients and their quantity required for the execution of the action, necessary tools used in the execution of the action, constraints on the duration of the action, how it is performed, why it is performed and other necessary settings of the environment. e.g., {Add} - {Add one banana to a blender}; here add is the necessary action and the step: Add one banana to a blender describes the action (adding), ingredient (banana), quantity (1)

activities using task graphs and their variations, such as FlowGraphs (Yamakata et al., 2020), Recipe Programs (Papadopoulos et al., 2022), ConjugateTaskGraphs (Hayes & Scassellati, 2016), and ActionDynamicTaskGraphs (Mao et al., 2023) and our dataset can be used to learn these task graphs in an unsupervised manner (or one can use the semantics of these various task graphs to label the videos and solve the problem in a supervised manner).

## B.2 HOW TO RECORD?

### B.2.1 SENSORS

Recognizing the limitations of the Hololens2 augmented reality device in capturing data, despite its advanced technology, we decided to employ a dual-device strategy [5]. While the Hololens2 offers a wealth of data from various sensors, we faced two main challenges. First, the limited field of view of the RGB camera inhibits comprehensive data capture. Second, utilizing all the secondary sensors of the Hololens2 requires operating in research mode, which, unfortunately, leads to a significant frame rate reduction for other sensory data, such as depth and monochrome cameras, when we increase the quality of the captured RGB frames.

To address these issues, we integrated a GoPro into our data-capturing system. Positioned above the Hololens2 on a head mount worn by the participant, the GoPro records 4K videos at 30 frames per second, offering a wider field of view compared to that of the Hololens2's RGB frames. This setup provides us with a more detailed perspective on the participant's activities. We use the Hololens2

---

[5]Although we use a dual-device strategy to record activities, it's important to note that these devices aren't synchronized prior to the start of the recording process. Instead, captured footage from both devices is programmatically synchronized during post-processing using the associated timestamps

in research mode to obtain a diverse range of data, including depth streams and spatial information such as head and hand movements. Additionally, we collect data from three Inertial Measurement Unit (IMU) sensors: an accelerometer, a gyroscope, and a magnetometer. This combined approach enables us to capture complete, high-quality activity data.

### B.2.2 DATA CAPTURING SYSTEM

We have designed a versatile and expandable system for recording procedural activities, which can be readily adapted to meet various needs. This system can function in two distinct ways: (1) as a standalone user application specifically designed for procedural activities and (2) as a comprehensive, plug-and-play system that functions beyond being just a user application.

In its first mode, the application serves a dual role: primarily as a display interface [6] for the procedure of the activity and secondarily as a tool for noting and updating any errors made during the execution of the activity. In its second mode, the system is equipped to capture data streams from various sensors and allows for easy connection and configuration. This dual functionality enhances the system's adaptability, making it an efficient tool for a wide range of procedural activities.

**User application** We will briefly explain how our system can be used as a user interface to capture complex procedural activities, including errors, using several illustrative snippets. This process within our system is divided into four stages to facilitate data collection from participants:

**Stage-1** As depicted in Figure 9, the first stage involves presenting the participant with a list of activities on the left side of the interface. Upon selecting an activity, the corresponding steps for the selected activity are then displayed on the right side of the page.

We offer two options for presenting the steps of an activity, depending on the input provided when information about the activities is loaded into the database: (1) **Recipe Text**: If the input for an activity is plain text, we display the text exactly as provided. (2) **Task Graph**: If the input for an activity is an action-centric task graph, we provide a valid sequence of steps that adheres to the constraints defined by the graph.

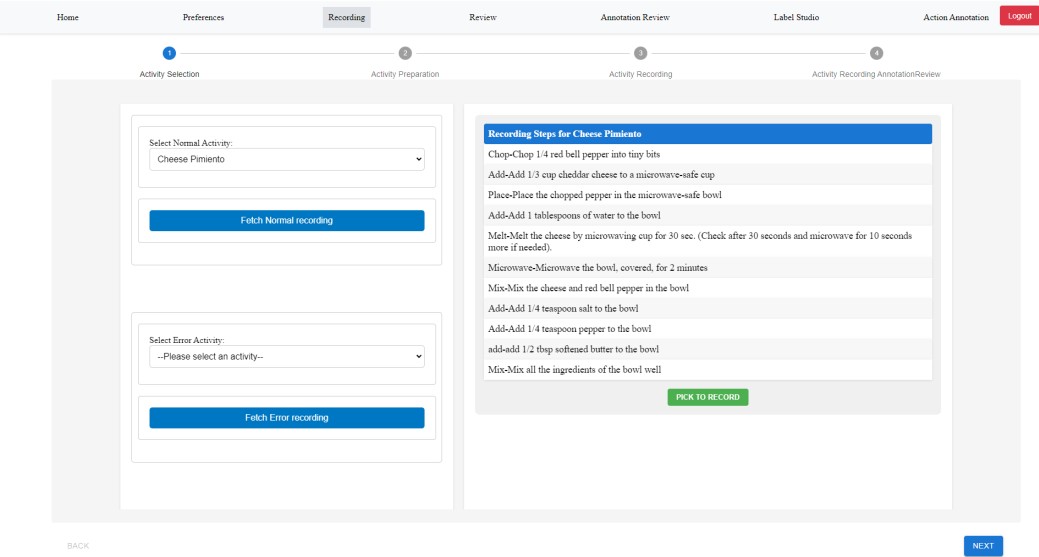

Figure 9: The interface displayed in the figure represents the initial stage of the recording cycle. Here, the participant can choose the activity they want to perform from the options presented on the left. After making a selection, the steps necessary for the chosen recipe will be shown.

---

[6]Please view the tablet that displays the interface, as shown in video snippets posted on our website

**Stage-2**    This stage is referred to as the "activity preparation" stage. Although optional for a normal recording, its primary function is to prepare a script to execute during an error recording. One of our approaches to capturing error recordings involves providing participants with an interface to contemplate the errors they intend to make and modify the description of the steps for a particular activity recording session.

As illustrated in Figure 10, participants can update each step based on different types of errors categorized as described above. When the participant records, they will see the updated step description as part of the sequence of steps. Moreover, GPT-4 has provided suggestions on potential errors that may occur during the activity, now available as static hint options for this recipe. However, we have observed that these generic errors provided by GPT-4 are not particularly helpful, as participants only considered them for script preparation in 20% of cases.

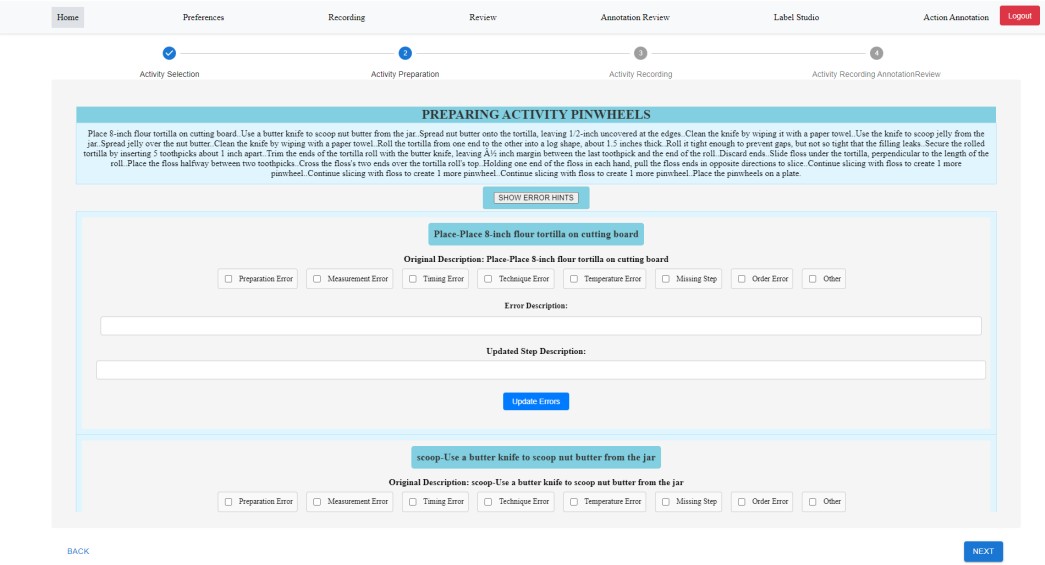

Figure 10: This interface enables participants to update step descriptions easily.

**Stage-3**    During this stage, we present the participants with the sequence of steps for the selected activity that they will perform. As shown in figure 11 we display each step either as plain text or present one topological order of the task graph.

**Stage-4**    After the data is captured either using our system or from a standalone recording system, we provide an interface to participants to review the recording they performed and correspondingly update any unplanned errors they make while performing the activity.

In one of our strategies for capturing error recordings, we asked participants to induce errors *impromptu* while performing the activity. In this situation, participants are given a series of steps corresponding to the task graph's topological order. Subsequently, participants updated information about errors they made while performing the recipe. Figure 12 illustrates a snippet where the participant updates one of the errors made while performing the recipe *Caprese Bruschetta*

**Data Capturing Application**    As introduced earlier, the standalone application described above can be transformed into a data-capturing application by configuring a few plug-and-play modules. Both the user application and its extended data-capturing application are built using the software components depicted in Figure 13.

In developing our data-capturing application, we have utilized data streams from various devices, specifically Hololens2 and a GoPro. The Hololens2 is particularly suited for our needs when set in research mode. It offers a wealth of data from an array of sensors, including a depth sensor, three

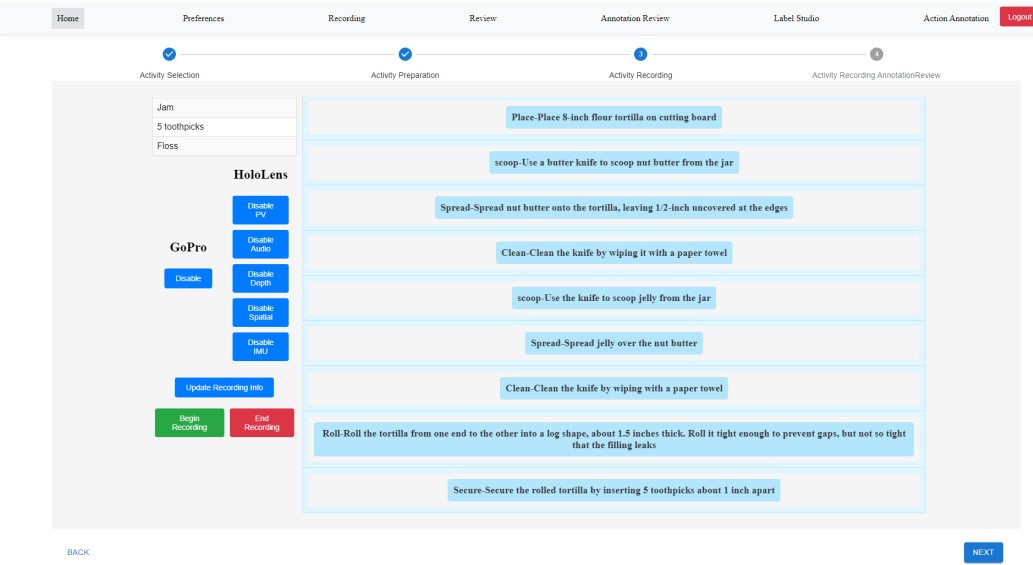

Figure 11: This interface shows the necessary steps to complete an activity.

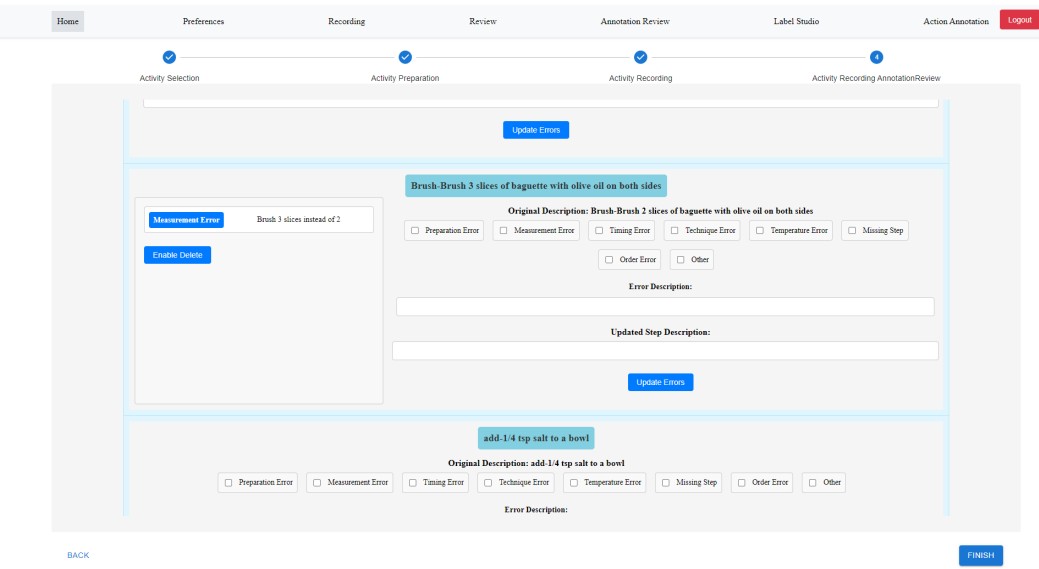

Figure 12: This interface is similar to 10, here the participant can update the information about the errors induced while performing the activity

Inertial Measurement Unit (IMU) sensors - an accelerometer, a gyroscope, and a magnetometer - and spatial information that contains head and hand tracking data.

For the Hololens2, we created a custom Unity streamer application, taking inspiration from (Dibene, Juan C. & Dunn, Enrique, 2022). This application acts as a server, while our Python backend application assumes the role of a client. When we initiate a recording session, we establish one TCP socket connection for each sensor to capture data. As the sensor-specific data

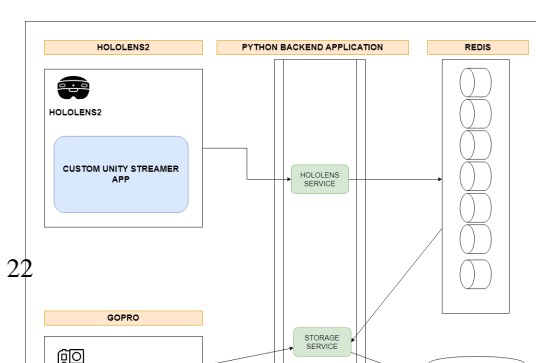

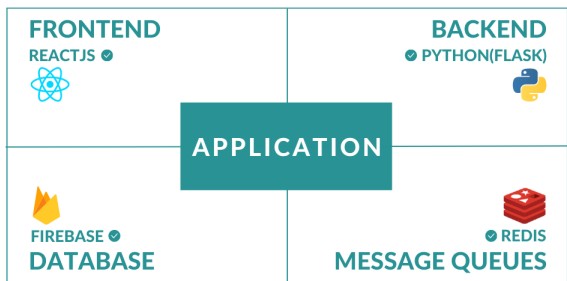

Figure 13: Software components used to build the proposed data capturing system

stream is received, it is immediately pushed onto the sensor-specific Redis message queue [7] Another dedicated Python backend service polls data from these message queues, processes it and subsequently stores it on a locally configured Network Attached Storage (NAS) server. When starting a recording session with GoPro, we utilize the OpenGoPro library to communicate and capture data at the established 4K resolution and 30 FPS. The recorded video is then downloaded from the GoPro through WiFi and saved onto the local NAS server. This system architecture (as shown in figure 14) allows us to capture, process, and securely store vast amounts of data, all in real-time.

### B.3 WHOM TO RECORD

**Participant Statistics.** We present the statistics regarding the participants who performed cooking activities in figure 15. It is important to note that participation in the entire recording process is voluntary, and individuals were not compensated.

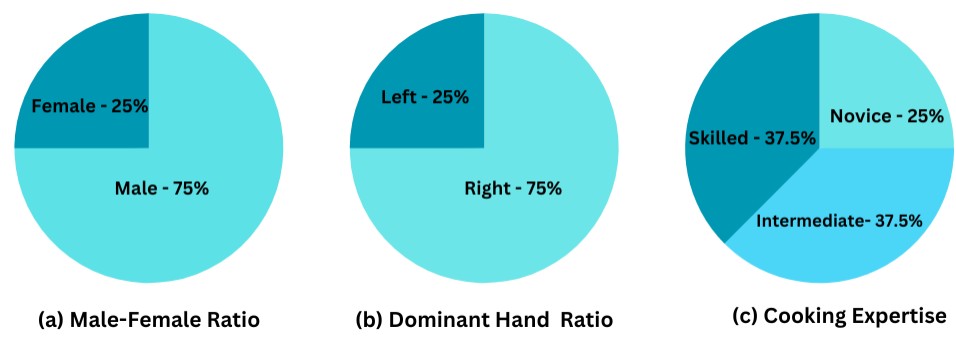

(a) Male-Female Ratio     (b) Dominant Hand Ratio     (c) Cooking Expertise

Figure 15: This data shows information about the individuals who were part of the recordings.

**Participant Training.** To ensure accurate data collection on cooking errors, participants must possess basic culinary skills and have complete knowledge of the recipes they will be preparing. To assist participants, we provided them with a comprehensive list of instructional videos on basic culinary skills and techniques specific to different recipes.

---

[7]Redis (https://redis.com/) is an open-source, in-memory data structure store that's used to implement NoSQL key-value databases, caches, and message brokers.

**Sources of bias.** While this work presents the first attempt at building a comprehensive 4D dataset to study errors in procedural tasks, we acknowledge the dataset's inherent biases. The number of participants contributing to this dataset is noticeably smaller than conventional, large-scale action or activity understanding datasets. Yet, it is important to mention that each participant is asked to perform and record the same recipe four times, and each time, the recording script changes, thus making each recording unique. Finally, note that many errors were intentional because the participants followed a script. However, they also made many unintentional errors in the process, which they annotated later.

## C  DATA COLLECTION - PRODUCTION PHASE

Once we determined what, where, and whom to record, we proceeded to collect data from participants while they engaged in cooking activities. Over the course of 32 days, we recorded in 10 different kitchen settings across the United States. Participants were given the opportunity to schedule their availability to perform these activities in various kitchen environments.

**Inspection & Acclimatisation** Before starting the recording process in every environment, participants must go through a set of steps. Firstly, they are instructed not to have any identifiable information on the body parts that will be exposed during the recording. Additionally, they are checked to ensure that they are not carrying any personal identification tools, such as a smartwatch with personal information. Since participants are filming in unfamiliar kitchen environments, they are given a detailed orientation about the location of all essential ingredients required to finish the recipe.

**Data Capture - Normal Recordings** Participants were given a tablet to access the user application described above. They were instructed to perform normal activities first. Upon selecting a normal activity, each participant is presented with a sequence of steps corresponding to the topological order of the constructed action-centric task graph. The participants were expected to follow the sequence of steps displayed on the tablet precisely and avoid making any errors by deviating from the given steps.

**Data Capture - Error Recordings** We devised and implemented three strategies for the participants to follow. Each participant was asked to choose a recording strategy for a particular environment and was guided accordingly in preparing for his recording. We list the formulated strategies here (1) **Pre-prepared error scripts**: The participants were given pre-prepared error scripts with missing steps and ordering errors. (2) **Prepare error scripts**: Once participants chose this strategy, they were given a web-based interface to create an error script for each error recipe recording and displayed the modified error script on a tablet, enabling participants to perform according to their modified error scripts (3) **Impromptu**: During the later stages of the recording process, we implemented a strategy where participants were asked to induce errors intentionally. Following the completion of each recording, participants were given access to a web-based interface to update any errors they made during each step.

Table 9 describes the statistics for each selected recipe about the number of normal and error recordings and their total durations, respectively.

## D  DATA COLLECTION - POST-PRODUCTION PHASE

### D.1  SYNCHRONIZATION

Following each recording session, data is moved to a local Network Attached Storage (NAS). Upon completion of the recording cycle for each kitchen environment, a synchronization service is employed to align the raw data streams captured by the Hololens2. Data from multiple streams—including RGB, depth, spatial, and three Inertial Measurement Unit (IMU) sensors—are synchronized using timestamps provided by the Hololens2. After synchronization, both the raw and synchronized data are uploaded to the cloud.

Table 9: $\mathcal{S}$ indicates the total number of steps in a given recipe, $\mathcal{N}_n$ shows the count of normal recordings taken for the recipe, $\mathcal{D}_n$ denotes the overall duration of these normal recordings, $\mathcal{N}_e$ shows the count of error recordings taken for the recipe, $\mathcal{D}_e$ denotes the overall duration of these error recordings

| Recipe | $\mathcal{S}$ | $\mathcal{N}_n$ | $\mathcal{D}_n$ (hrs) | $\mathcal{N}_e$ | $\mathcal{D}_e$ (hrs) |
|---|---|---|---|---|---|
| Pinwheels | 19 | 4 | 0.72 | 8 | 1.2 |
| Tomato Mozzarella Salad | 9 | 11 | 1.31 | 7 | 0.64 |
| Butter Corn Cup | 12 | 6 | 1.62 | 8 | 1.49 |
| Tomato Chutney | 19 | 7 | 3.34 | 8 | 2.01 |
| Scrambled Eggs | 23 | 6 | 2.69 | 10 | 3.13 |
| Cucumber Raita | 11 | 12 | 2.9 | 8 | 1.36 |
| Zoodles | 13 | 5 | 1.35 | 10 | 2.19 |
| Microwave Egg Sandwich | 12 | 6 | 1.05 | 12 | 1.67 |
| Sauted Mushrooms | 18 | 6 | 2.73 | 8 | 2.21 |
| Blender Banana Pancakes | 14 | 7 | 1.78 | 12 | 2.57 |
| Herb Omelet with Fried Tomatoes | 15 | 6 | 1.73 | 11 | 2.14 |
| Broccoli Stir Fry | 25 | 11 | 5.74 | 5 | 1.68 |
| Pan Fried Tofu | 19 | 8 | 3.38 | 7 | 2.31 |
| Mug Cake | 20 | 7 | 2.44 | 10 | 2.32 |
| Cheese Pimiento | 11 | 6 | 1.47 | 9 | 1.72 |
| Spicy Tuna Avocado Wraps | 17 | 7 | 2.0 | 11 | 2.66 |
| Caprese Bruschetta | 11 | 6 | 1.92 | 12 | 2.73 |
| Dressed Up Meatballs | 16 | 6 | 2.0 | 10 | 3.09 |
| Microwave Mug Pizza | 14 | 7 | 1.47 | 6 | 1.14 |
| Ramen | 15 | 10 | 2.40 | 7 | 1.45 |
| Coffee | 16 | 8 | 1.97 | 7 | 1.58 |
| Breakfast Burritos | 11 | 6 | 1.22 | 10 | 1.52 |
| Spiced Hot Chocolate | 7 | 6 | 0.82 | 10 | 1.01 |
| Microwave French Toast | 11 | 9 | 1.94 | 5 | 0.66 |
| **Total** | 358 | 173 | 50.05 | 211 | 44.41 |

## D.2 ANNOTATION

### D.2.1 COARSE-GRAINED ACTION/STEP ANNOTATIONS

We designed an interface for performing step annotations in Label Studio [8]. Each annotator is presented with this interface to mark the start and end times for each step. Our steps are significantly longer than a single fine-grained action and encompass multiple fine-grained actions necessary for performing the described step. For example, in order to accomplish the step *{Chop a tomato}*, we include the following (1) Pre-conditional actions of *{opening refrigerator, grabbing a polythene bag of tomatoes, taking a tomato, placing the tomato on cutting board, close fridge}* (2) Post-conditional actions of *{placing down the knife, grabbing the polythene bag of tomatoes, open fridge and place the bag in the fridge}*. Table 2 summarizes and compares coarse-grained action/step annotations for our dataset as well as other popular datasets. (see Figure **??**, which illustrates the key steps for a recording and the corresponding step and action annotations). To perform coarse-grained action/ step annotations, we utilized both our user application and Label Studio. We integrated our application with Label Studio using the APIs provided by Label Studio [9]. This integration allowed for the

---

[8]https://labelstud.io/
[9]https://labelstud.io/

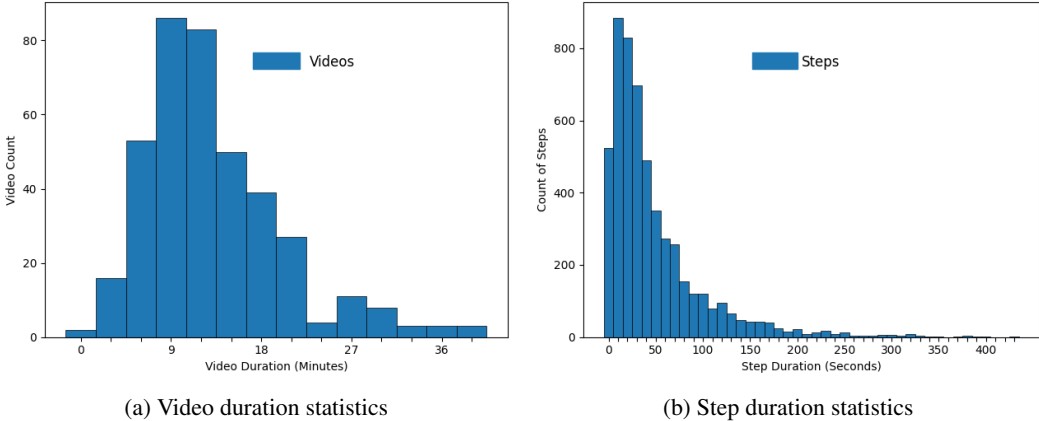

(a) Video duration statistics                    (b) Step duration statistics

Figure 16: Displays duration statistics for videos and individual steps

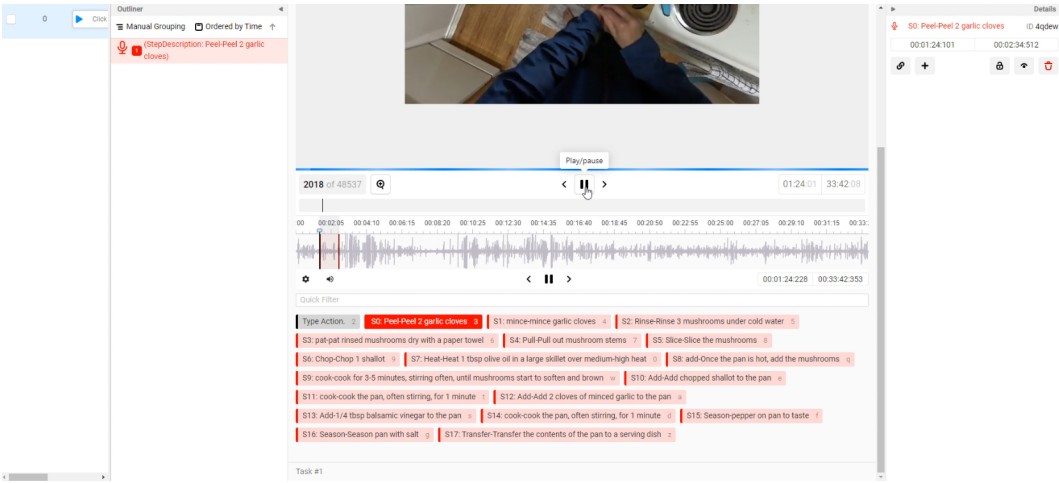

Figure 17: Annotation interface developed to generate step annotations for a recording

seamless creation of a labelling environment for each recording and has provided a way to ensure that generated annotations are reliably stored.

### D.2.2 FINE-GRAINED ACTION ANNOTATIONS

Inspired by the pause-and-talk narrator (Damen et al., 2020), we have designed and developed a web-based tool for fine-grained action annotation that utilizes OpenAI's Whisper APIs for speech-to-text translation. Even though we have built this system around the Whisper API, it is flexible enough to accommodate any automatic speech recognition (ASR) system that can serve transcription requests. We will release the developed web-based annotation tool as part of our codebase upon acceptance.

**Annotation Interface**  We'll briefly explain the reasoning behind the design choices of the annotation interface. Firstly, in step annotations, we are interested in marking the temporal boundaries for each step of the recording. As a result, we've positioned a complete list of all steps corresponding to the activity underneath the video. After identifying a time period as the boundary for a particular activity step, this will become visible on the screen's left-hand side. At the same time, you can see the start and end times of the step on the right side in the corresponding time slot that can be used for minor adjustments in the created annotation.

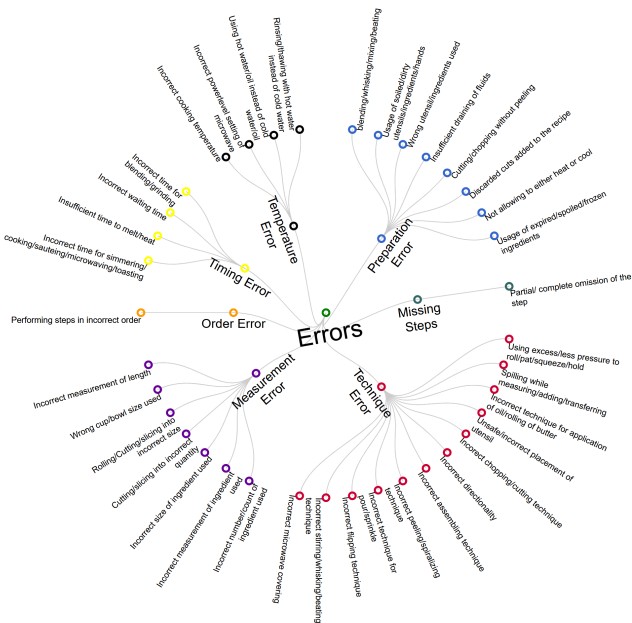

Figure 18: A structured synopsis of various types of errors and their brief descriptions compiled from the annotations

### D.2.3 ERROR ANNOTATIONS

As outlined in Appendix C, each participant is required to update the errors made during every recording step. We gathered all these error annotations and categorized them into distinct error categories. This categorization is concisely presented in a mind map, as depicted in figure 18

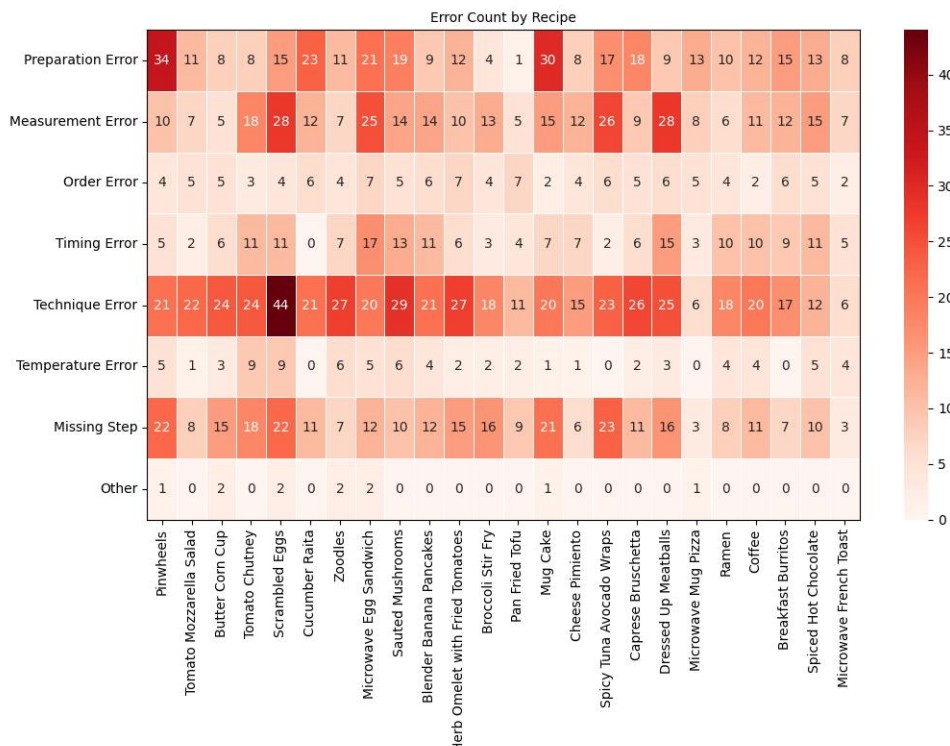

Figure 19: Displays a heatmap that shows the frequency of each type of error in the recordings for a recipe. The X-axis identifies the name of the recipe, while the Y-axis specifies the category of the error type. Each point on the heatmap represents the total count of errors of a specific category performed in all recordings for a recipe.

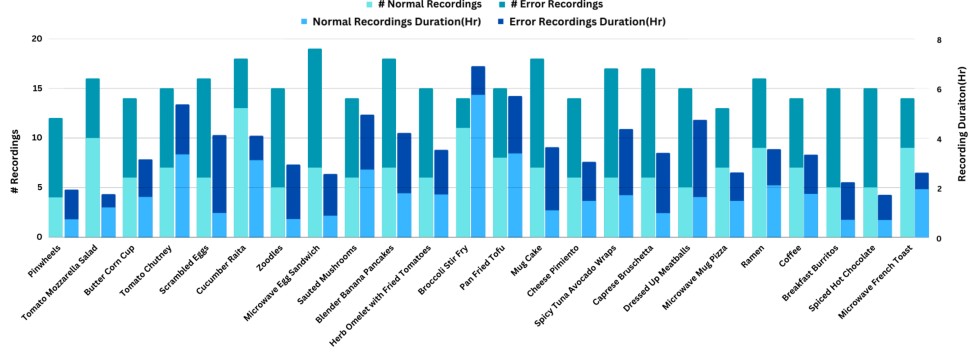

Figure 20: presents statistics for both normal and error recordings. Specifically, the number of recordings for each type is indicated on the left-side Y-axis and represented by the bars on the left side for each recipe. Meanwhile, the corresponding durations for these recordings are shown on the right-side Y-axis and are depicted by the bars on the right side for each recipe.

# E   DATA COMPOSITION

In this section, we list down all the components provided as part of our data. **Raw and synchronized multi-modal data from Hololens2:** The dataset includes raw data captured using the Hololens2 device. This data is multi-modal, which means it contains information in several different forms, including visual (e.g., images or videos), auditory (e.g., sounds or speech), and others (like depth information, accelerometer readings, etc.). **4K videos from GoPro:** Includes high-resolution 4K videos recorded using a GoPro camera. Such high-resolution video can provide much detail, which is particularly useful for tasks like object recognition. **Step annotations** for all the data. **Fine-grained actions for 20% of the data:** Fine-grained actions might include specifics about what objects are being manipulated, exactly what movements are being made, and so on. This data could be helpful for tasks that involve understanding or predicting specific types of actions. **Extracted features using multiple backbones for different tasks:**. We provide a comprehensive overview of the components we release with the dataset in table 10

Table 10: This table presents an overview of the components we release as part of the dataset.

| Device | Type | Component | Sync | Other |
|---|---|---|---|---|
| Hololens2 | Raw | RGB
RGB pose
Depth
Depth pose
Audio
Head pose
Left wrist pose
Right wrist pose
IMU Accelerometer
IMU Gyroscope
IMU Magnetometer | Synchronized | RGB
RGB pose
Depth
Depth pose
Audio
Head pose
Left wrist pose
Right wrist pose
IMU Accelerometer
IMU Gyroscope
IMU Magnetometer |
| Gopro | Raw | RGB
Audio | | |
| Annotations | Step
Fine-grained Action | | | |

# F TASK GRAPHS

## F.1 TASK GRAPHS FOR SELECTED RECIPES

### F.1.1 BREAKFAST BURRITOS

> **Recipe Description** Extract the contents of an egg and add it to a microwave-safe bowl. Whisk the egg. Microwave for 3 minutes, stirring in between. Add 1 tbsp salsa to the bowl. Sprinkle oregano in the bowl. Add 1/2 tbsp sweet and sour sauce to the bowl. Mix the contents of the bowl well. Place 8-inch tortilla on a cutting board. Pour egg mixture on top of the tortilla. Sprinkle 1 tbsp shredded cheddar cheese on top of the egg. Roll the tortilla from one end to another into a log shape, about 1.5 inches thick. Roll it tight enough to prevent gaps but not so tight that the filling leaks.

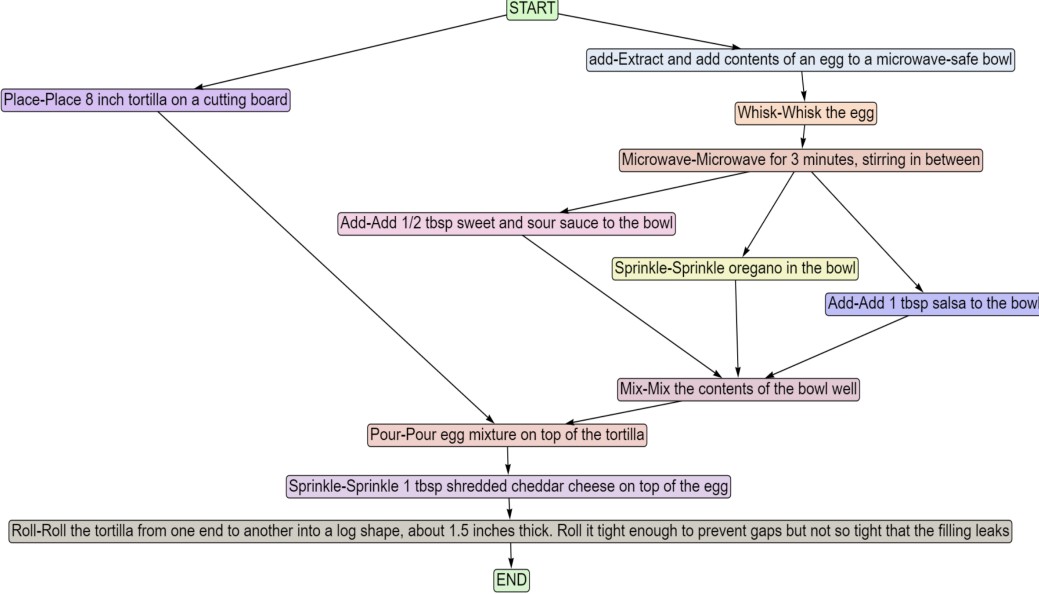

Figure 21: Picture illustrates constructed action-centric task graph for the recipe **Breakfast Burritos**

### F.1.2 BROCCOLI STIR FRY

> **Recipe Description** Peel 2 cloves of garlic and mince them. Combine 1/6 cup water, 1/8 cup soy sauce, 1 tablespoon honey, 2 cloves minced garlic, 1/2 tablespoon minced ginger, 1/8 teaspoon black pepper, and 1 teaspoon cornstarch in a small bowl. Whisk to mix. Set aside. Take 5 in number broccoli florets. Take 2 cremini mushrooms, and slice them. Take 1 bell pepper and slice 1/3 of the bell pepper. Heat 2 tablespoons olive oil in a large skillet over medium-high heat. Add the broccoli and sliced mushrooms and cook, stirring often, for 4 minutes. If the pan gets too hot on medium-high, turn the heat down to medium. Add the bell pepper and continue cooking, stirring often, for 2-3 minutes, until vegetables are crisp-tender. Whisk the sauce again to recombine the ingredients. Pour the sauce into the skillet and cook, stirring, for 1 minute until the sauce thickens.

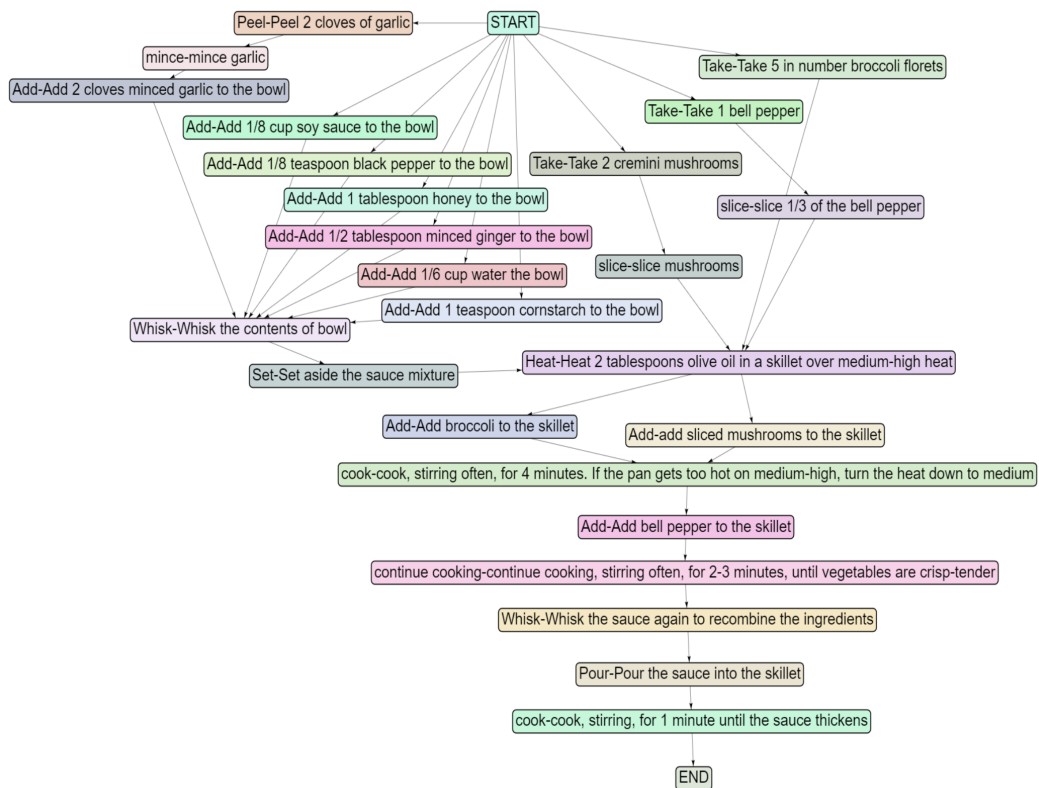

Figure 22: Picture illustrates constructed action-centric task graph for the recipe **Broccoli Stir Fry**

### F.1.3 BUTTER CORN CUP

> **Recipe Description** Measure 2 cups of frozen corn. Thaw the frozen corn by putting it in a sieve and running it under cold water. Add the corn into a microwave-safe bowl. Microwave the corn for 2 minutes. Add 1 teaspoon of butter and 1 teaspoon of pepper powder to the bowl and then stir. Microwave the corn for 3 more minutes. Add 1 teaspoon salt to the bowl. Extract lime juice from 1/3 lime and add it to the bowl. Mix the contents of the bowl well.

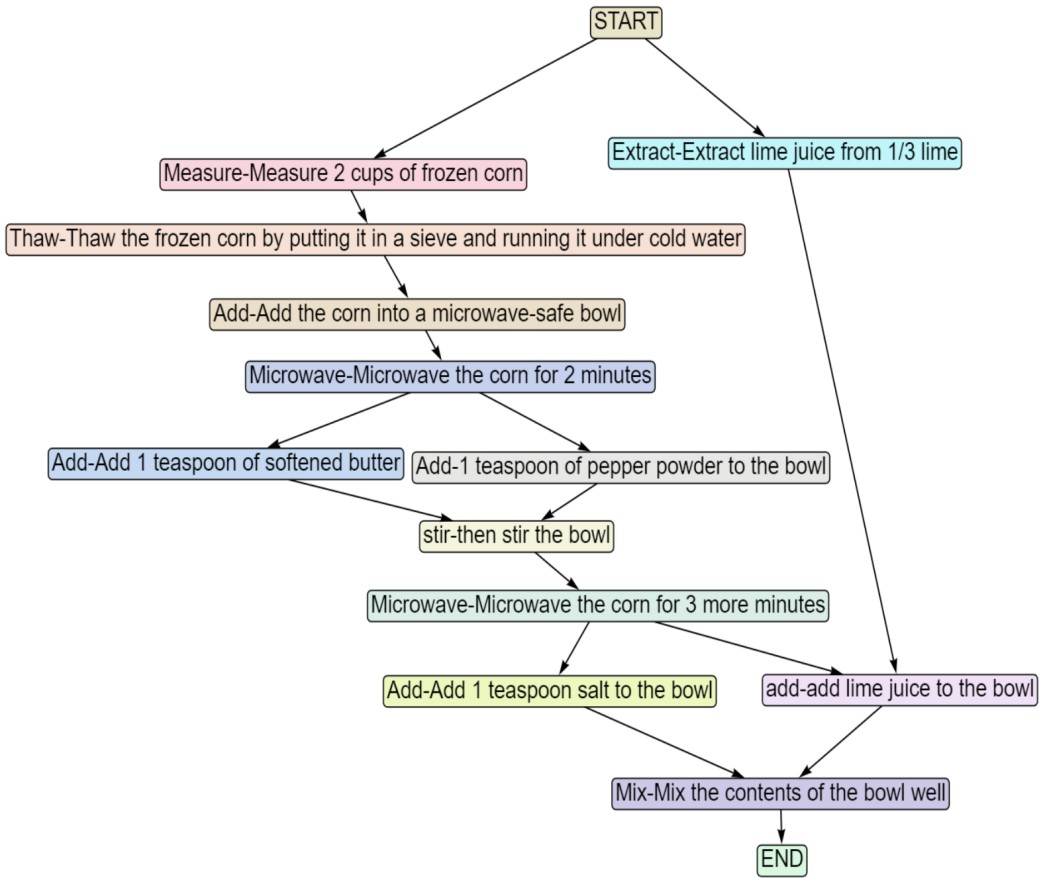

Figure 23: Picture illustrates constructed action-centric task graph for the recipe **Butter Corn Cup**

### F.1.4 CAPRESE BRUSCHETTA

**Recipe Description** Slice Baguette into 1/2 inch thick rounds. Brush 2 slices of baguette with olive oil on both sides. Toast both sides of the slices on the pan for 2 to 3 minutes until lightly charred and crispy. Cut 1/4 cup of cherry tomatoes into halves. In a bowl, add the cut cherry tomatoes, 1/8 cup shredded mozzarella, 1/16 cup basil, 1/4 tsp salt, and 1/4 tsp pepper. Combine the contents of the bowl. Spoon the mixture onto the bread.

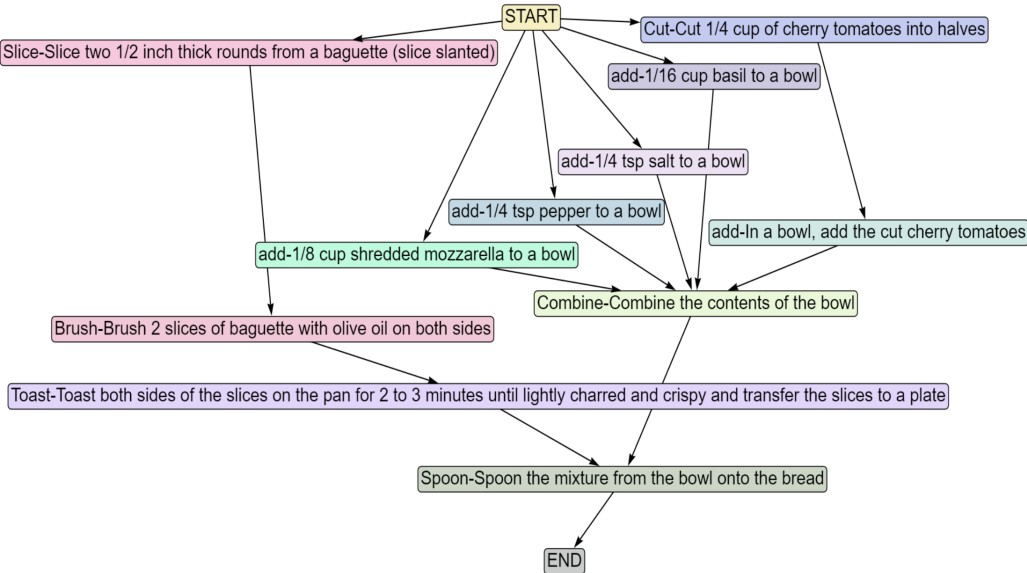

Figure 24: Picture illustrates constructed action-centric task graph for the recipe **Caprese Bruschetta**

### F.1.5 CHEESE PIMIENTO

**Recipe Description**  Chop 1/4 bell pepper into tiny bits. Place the chopped pepper in a microwave-safe bowl. Add 1 tablespoons of water to the bowl. Microwave, covered, for 2 minutes. Add 1/3 cup cheddar cheese to microwave-safe cup. Melt the cheese by microwaving it for 30 seconds(Check after 30 seconds and microwave for 10 seconds more if needed). Mix the cheese and red bell pepper in the bowl and add 1/2 tbsp softened butter. Add 1/4 teaspoon salt. Add 1/4 teaspoon pepper. Mix all the ingredients in the bowl well.

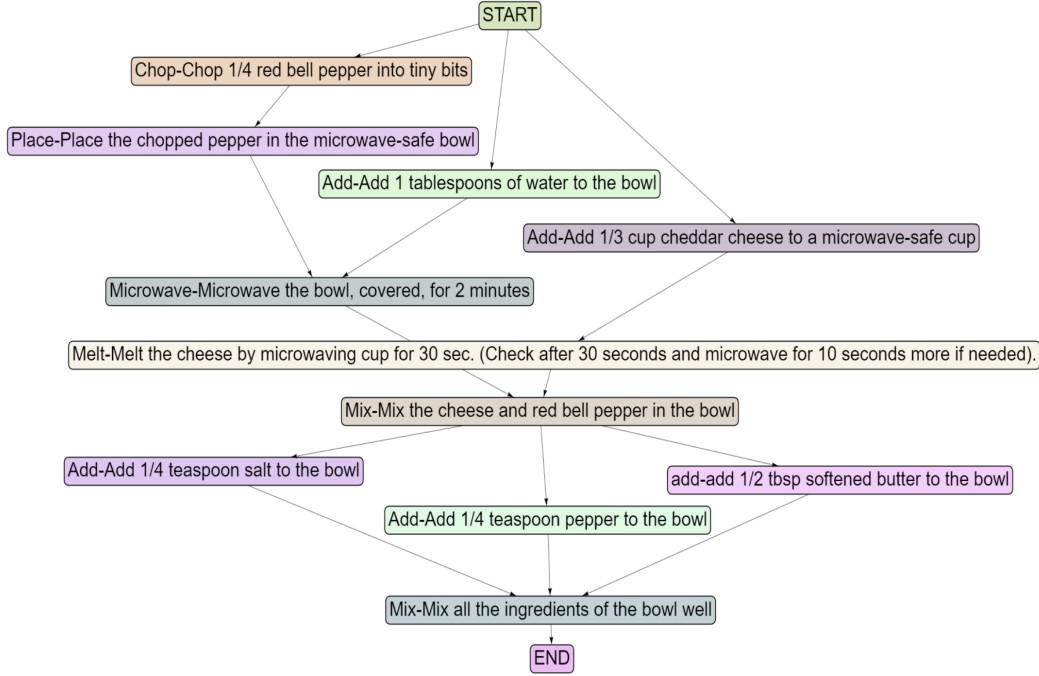

Figure 25: Picture illustrates constructed action-centric task graph for the recipe **Cheese Pimiento**

### F.1.6 COFFEE

> **Recipe Description** Measure 12 ounces of cold water and transfer it to a kettle. Boil the water. While the water is boiling, assemble the filter cone. Place the dripper on top of a coffee mug. Prepare the filter insert by folding the paper filter in half to create a semi-circle, and in half again to create a quarter-circle. Place the paper filter in the dripper and spread open to create a cone. Weigh the coffee beans (8 oz – 12 oz). Grind the coffee beans until the coffee grounds are the consistency of coarse sand, about 20 seconds. Transfer the grounds to the filter cone. Once the water has boiled, check the temperature of the water. (The water should be between 195-205 degrees Fahrenheit or between 91-96 degrees Celsius. If the water is too hot, let it cool briefly.) Pour a small amount of water into the filter to wet the grounds. Wait about 30 seconds for the coffee to bloom. (You will see small bubbles or foam on the coffee grounds during this step.) Slowly pour the rest of the water over the grounds in a circular motion. Do not overfill beyond the top of the paper filter. Let the coffee drain completely into the mug before removing the dripper. Discard the paper filter and coffee grounds.

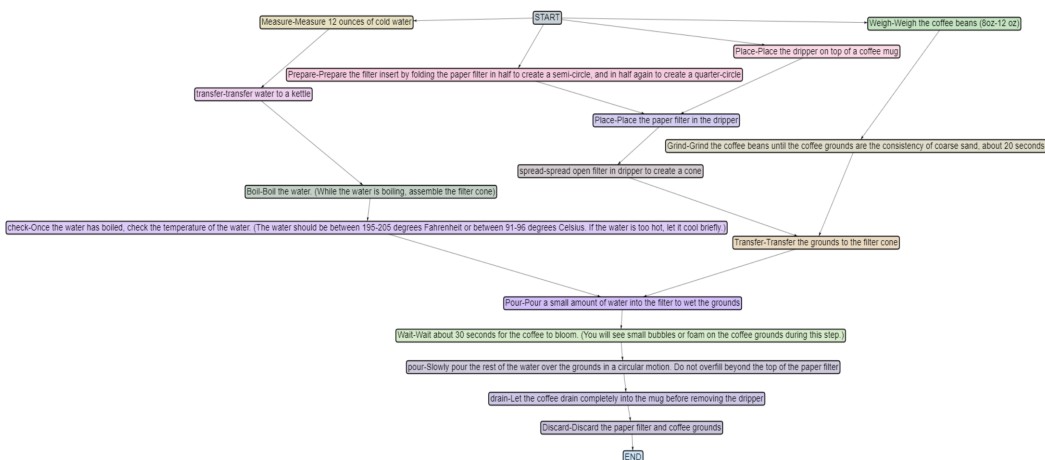

Figure 26: Picture illustrates constructed action-centric task graph for the recipe **Coffee**

### F.1.7 CUCUMBER RAITA

**Recipe Description** In a mixing bowl, whip 1 cup of chilled curd until smooth. Use fresh homemade or packaged curd. Avoid using sour-tasting curd. Rinse, peel and grate 1 medium-sized cucumber. (You need 1 cup of grated cucumber). Add the cucumber to the whisked curd. Add 1 teaspoon of cumin powder, ½ teaspoon of chaat masala powder, ¼ teaspoon of red chilli powder and 1/4 teaspoon salt. Lastly, add 1 tablespoon of chopped coriander leaves (cilantro). Combine all the ingredients.

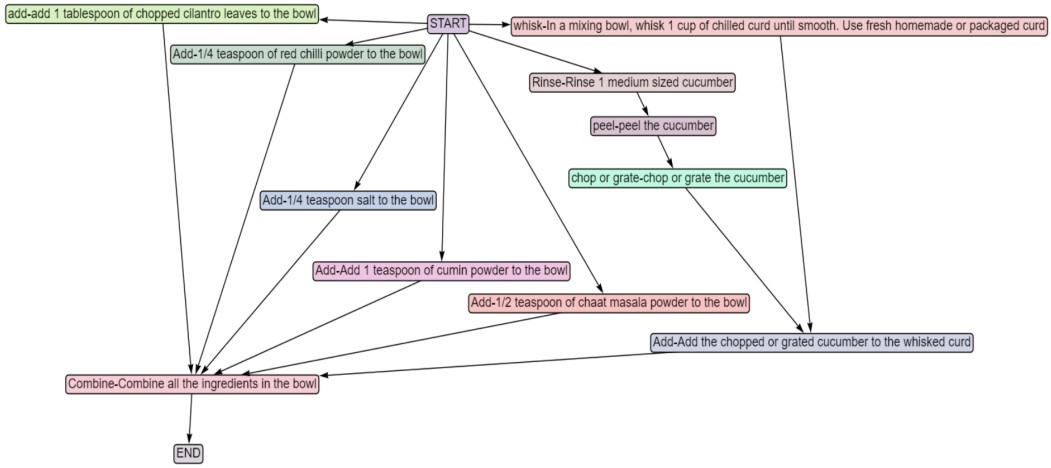

Figure 27: Picture illustrates constructed action-centric task graph for the recipe **Cucumber Raita**

### F.1.8 DRESSED-UP MEATBALLS

> **Recipe Description**  Cut 1/4 medium carrot into short, thin strips. Peel one medium onion and cut it into two pieces. Slice 1/8 medium onion. Peel 1 garlic clove. Cut and Mince 1/8 garlic clove. Place 5 meatballs in a 3-qt Microwave-safe dish. Top with the carrots, onion, garlic and pepper. Mix 1/4 cup sweet-and-sour sauce and 1/2 teaspoon soy sauce in a small bowl. Pour the sauces over the meatballs. Microwave, covered, on high for 1.5 minutes. Stir the contents in the microwave with a spoon. Microwave, covered, on high for 1.5 minutes. Stir the contents in the microwave with a spoon. Microwave for 1 more minute.

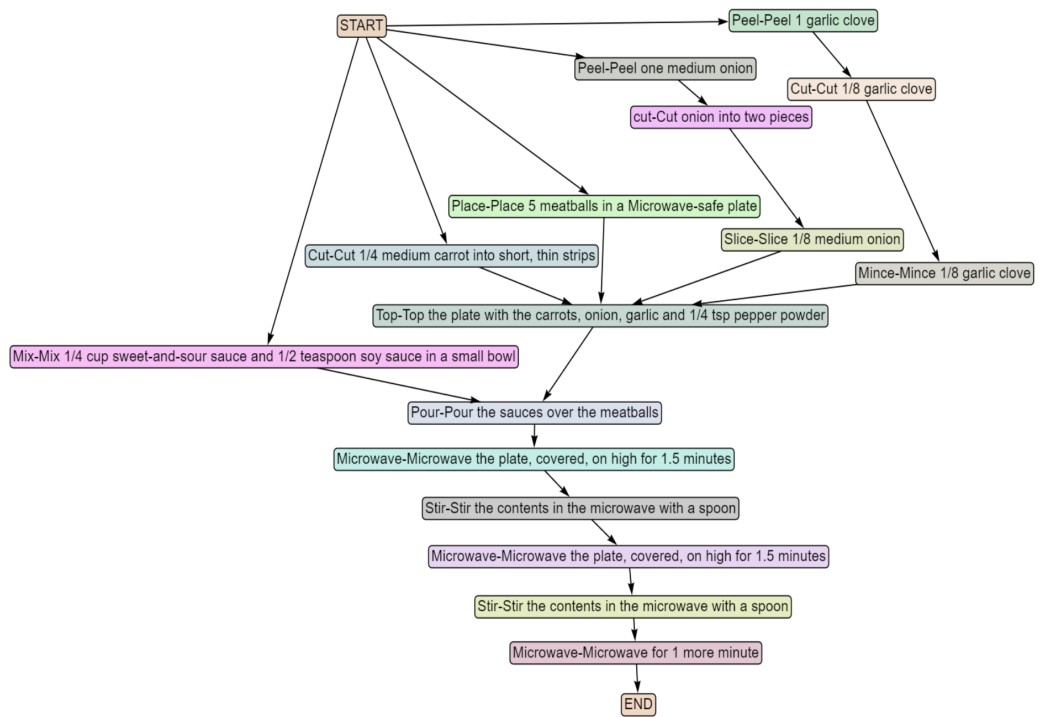

Figure 28: Picture illustrates constructed action-centric task graph for the recipe **Dressed-Up Meatballs**

### F.1.9 HERB OMELETTE WITH FRIED TOMATOES

> **Recipe Description** Take a tomato and chop it into two pieces. Heat 1 tbsp oil in a small non-stick frying pan, then cook the tomatoes cut-side down until they start to soften and colour. Chop 2 tbsp cilantro. Meanwhile, crack one egg in a bowl and add the chopped cilantro and 1/2 tsp ground black pepper. Beat the contents of the bowl. Scoop the tomatoes from the pan and put them on a serving plate. Pour the egg mixture into the pan and stir gently with a wooden spoon so the egg that sets on the base of the pan moves to enable the uncooked egg to flow into the space. Stop stirring when it's nearly cooked to allow it to set into an omelette. Transfer it to the plate and serve with the tomatoes.

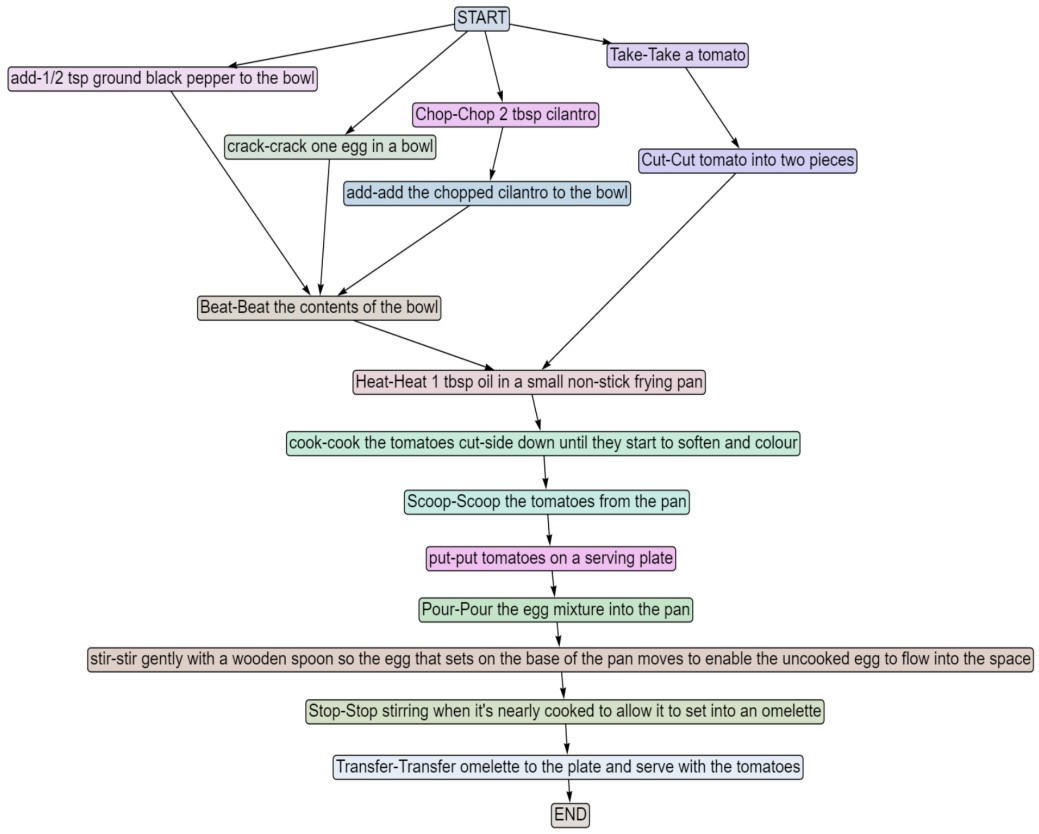

Figure 29: Picture illustrates constructed action-centric task graph for the recipe **Herb omelette with fried tomatoes**

F.1.10   MICROWAVE EGG SANDWICH

**Recipe Description**   Coat a 6-oz. ramekin cup with cooking spray. Pour 1 egg into the ramekin cup. Microwave the ramekin cup uncovered on high for 30 seconds and stir. Continue to Microwave for 15-30 more seconds or until the egg is almost set. Top with 1 tablespoon of salsa and sprinkle with 1 tablespoon of cheese. Microwave just until cheese melts, about 10 seconds. Cut the English muffin into two pieces with a knife. Line the bottom of the English muffin with lettuce. Place the egg from the cup over the lettuce and replace the top of the English muffin.

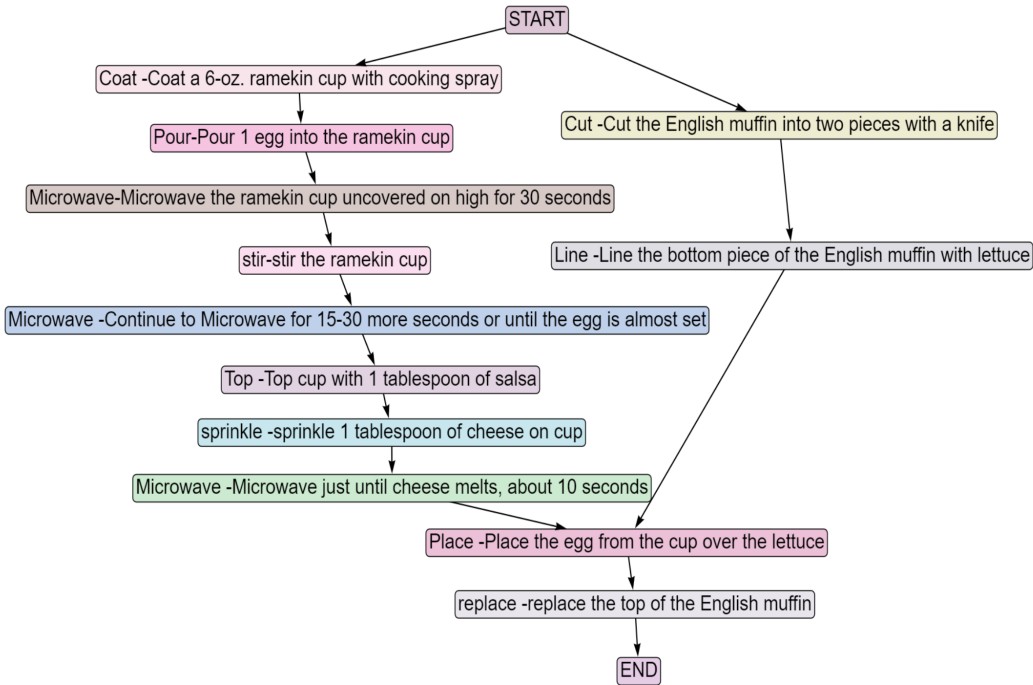

Figure 30: Picture illustrates constructed action-centric task graph for the recipe **Microwave Egg Sandwich**

### F.1.11   MICROWAVE FRENCH TOAST

> **Recipe Description**   In a large mug, melt 1 tablespoon of butter in the microwave for about 30 seconds. Roll the butter around in the cup to coat it. In the mug, whisk one egg with a fork until well blended. Sprinkle 1/4 teaspoon cinnamon over the egg. Add 1/4 teaspoon vanilla extract and stir again. Cut or tear 1 slice of bread into bite-size pieces and add to the egg mixture in the mug, pressing the bread down into the egg. Microwave on high for 90 seconds until the egg is cooked through. Put the cup's contents on a plate, cut it up, and serve.

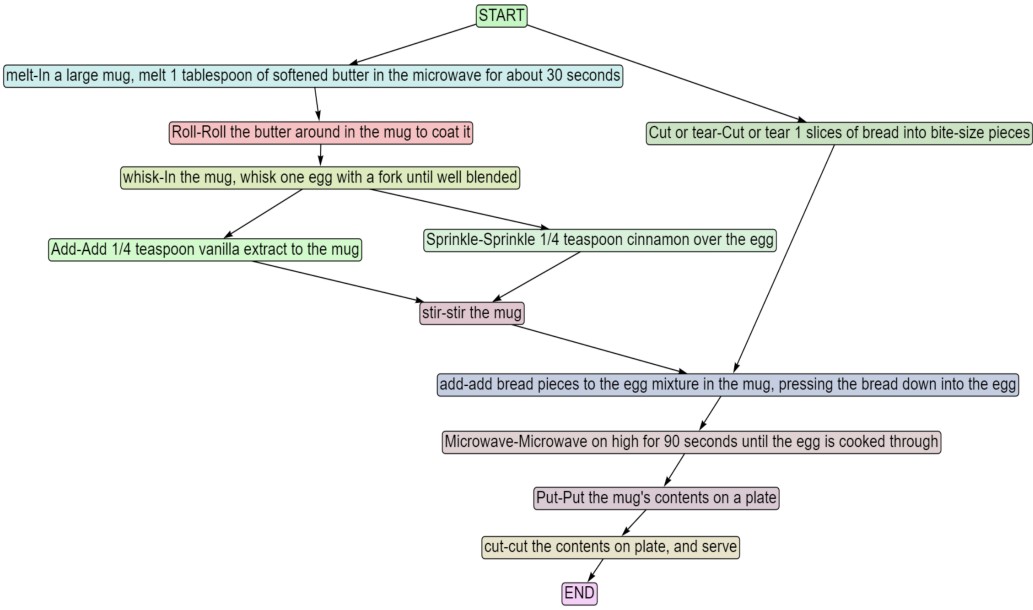

Figure 31: Picture illustrates constructed action-centric task graph for the recipe **Microwave French Toast**

### F.1.12   MICROWAVE MUG PIZZA

**Recipe Description**   Take a microwavable mug. Measure 4 tablespoons of flour and add it to the mug. Measure 1/8 teaspoon of baking powder and add it to the mug. Measure 1/16 teaspoon of baking soda and add it to the mug. Measure 1/8 teaspoon of salt and add it to the mug. Stir the contents in the mug well. Add in 3 tablespoons of milk and 1 tablespoon of olive oil to the mug. Mix the contents of the mug thoroughly. (There might be some lumps, but that is ok.). Take 1 tablespoon of marinara sauce and spread it around the surface of the batter. Sprinkle 1 generous tablespoon of mozzarella cheese on top of the sauce. Sprinkle dried Italian herbs inside the mug. Microwave for 1 minute 20 seconds, or until it rises and the toppings are bubbling.

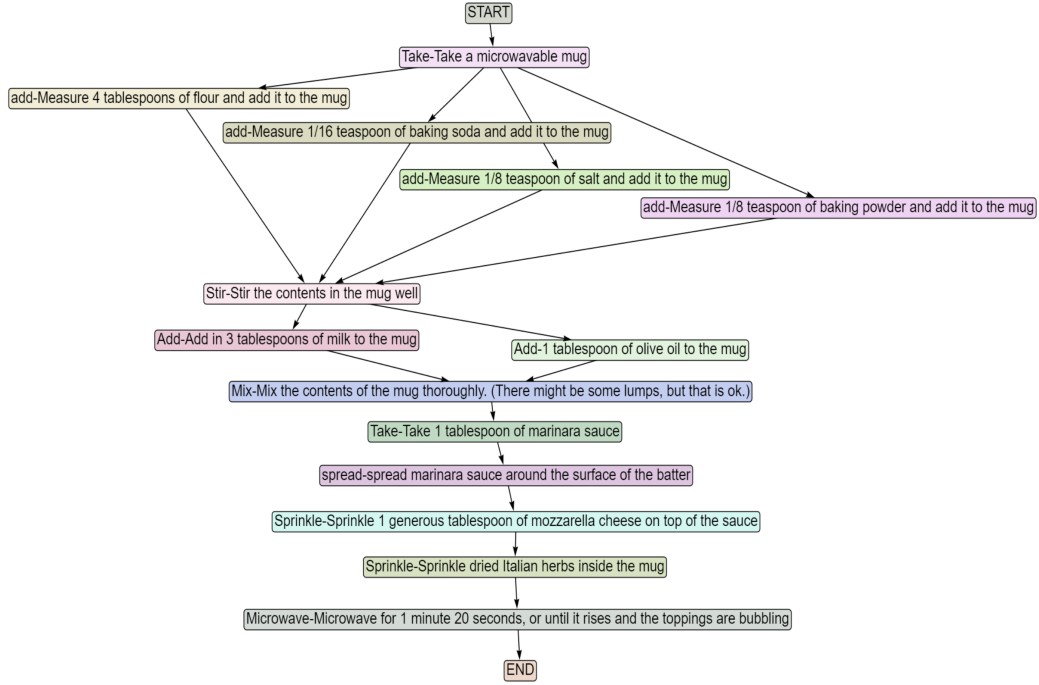

Figure 32: Picture illustrates constructed action-centric task graph for the recipe **Microwave Mug Pizza**

F.1.13 MUG CAKE

> **Recipe Description**   Place the paper cupcake liner inside the mug. Set aside. Measure and add 2 tbsp flour, 1.5 tbsp sugar, 1/4 tsp baking powder, and a pinch of salt to the mixing bowl. Whisk to combine. Measure and add 2 tsp vegetable oil, 2 tbsp water, and 1/4 tsp vanilla extract to the bowl. Whisk batter until no lumps remain. Pour batter into prepared mug. Microwave the mug and batter on high power for 60 seconds. Check if the cake is done by inserting and toothpick into the center of the cake and then removing it. If wet batter clings to the toothpick, microwave for an additional 5 seconds. If the toothpick comes out clean, continue. Invert the mug to release the cake onto a plate. Allow to cool until it is no longer hot to the touch, then carefully remove the paper liner. While the cake is cooling, prepare to pipe the frosting. Scoop 4 spoonfuls of chocolate frosting into a zip-top bag and seal, removing as much air as possible. Use scissors to cut one corner from the bag to create a small opening ¼-inch in diameter. Squeeze the frosting through the opening to apply small dollops of frosting to the plate in a circle around the base of the cake.

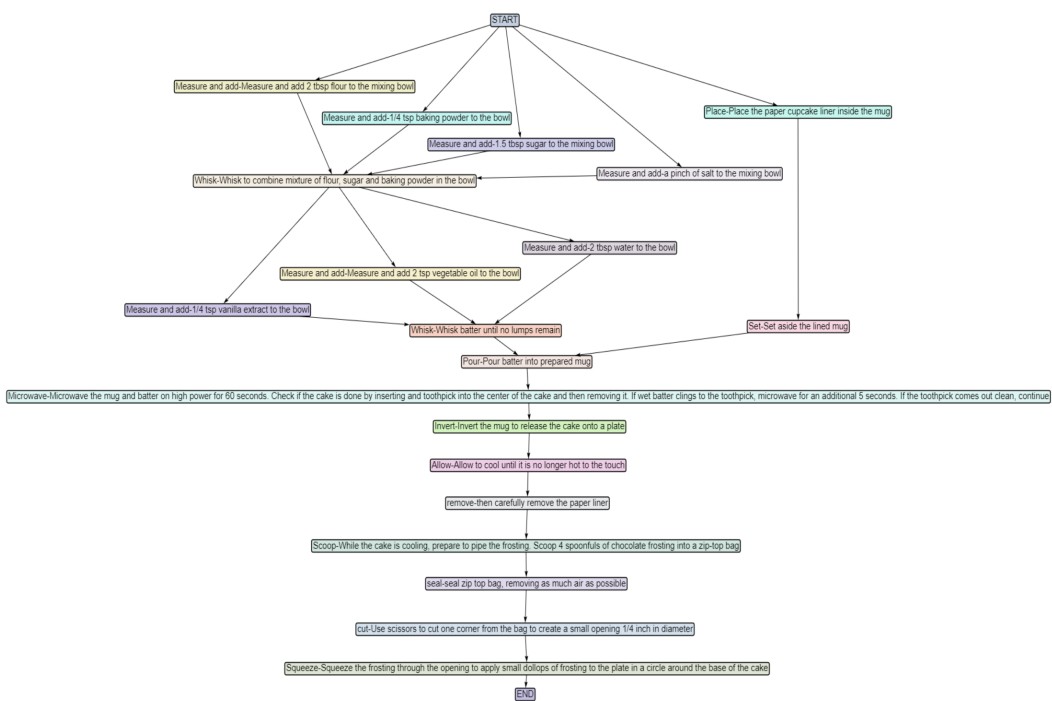

Figure 33: Picture illustrates constructed action-centric task graph for the recipe **Mug Cake**

F.1.14   PAN FRIED TOFU

**Recipe Description**   Cut 1/4 block or 3 ounces of fresh tofu into large cubes (about 1 inch x 1 inch) and pat it dry with a towel. Add 1 tablespoon of olive oil to a non-stick pan and add the tofu cubes and a few pinches of salt. Turn on the heat to medium. Cook 5 to 6 minutes until lightly browned on the bottom. Briefly remove the pan from the heat to reduce spitting. Flip the tofu with tongs. Return the heat to medium and cook until browned. Briefly remove from the heat again and drizzle with 1 tablespoons sesame oil and 1 tablespoon soy sauce (watch for spitting). Return to low heat and cook for 2 minutes, then flip and cook for 2 minutes until the color is darkened. Transfer toa serving dish.

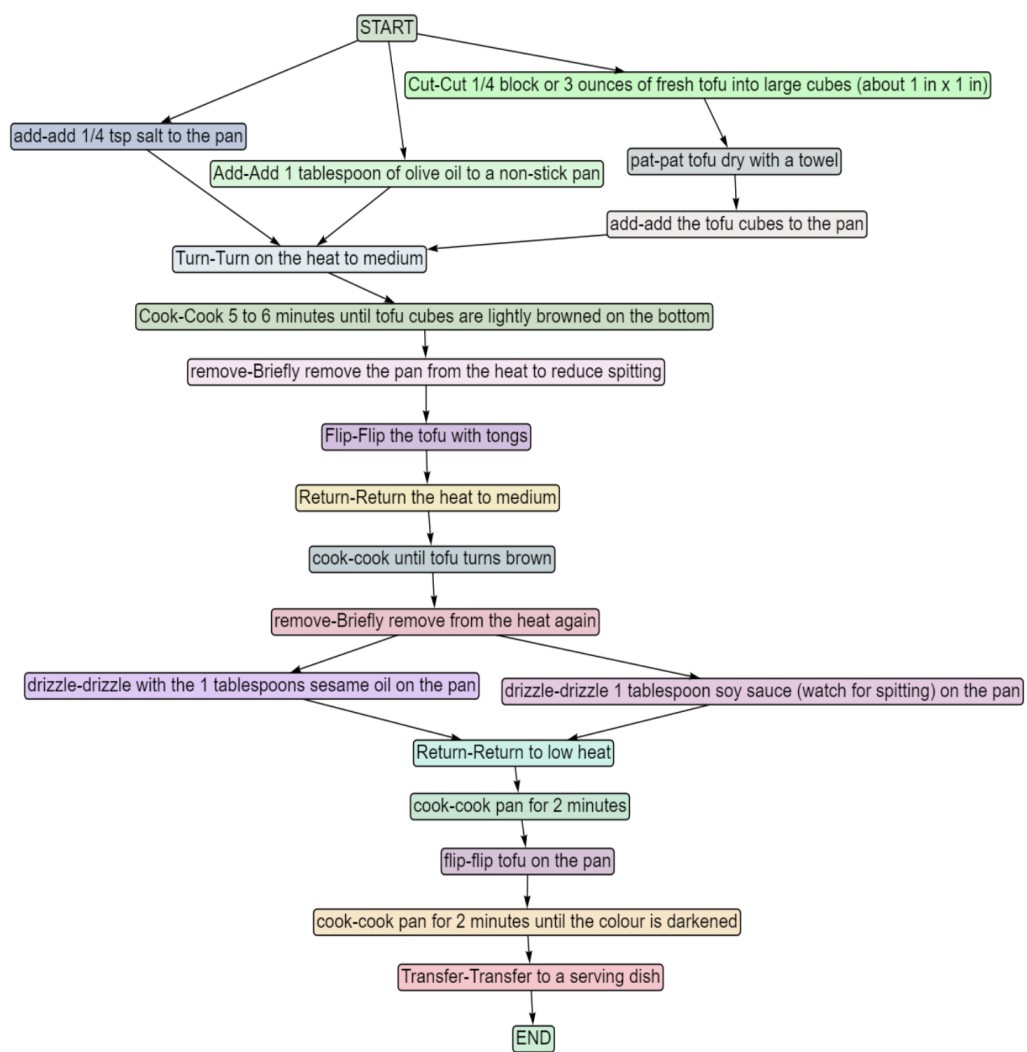

Figure 34: Picture illustrates constructed action-centric task graph for the recipe **Pan Fried Tofu**

F.1.15 PINWHEELS

> **Recipe Description**   Place 8-inch flour tortilla on the cutting board. Use a butter knife to scoop nut butter from the jar. Spread nut butter onto the tortilla, leaving 1/2-inch uncovered at the edges. Clean the knife by wiping it with a paper towel. Use the knife to scoop jelly from the jar. Spread jelly over the nut butter. Clean the knife by wiping it with a paper towel. Roll the tortilla from one end to the other into a log shape, about 1.5 inches thick. Roll it tight enough to prevent gaps, but not so tight that the filling leaks. Secure the rolled tortilla by inserting 5 toothpicks about 1 inch apart. Trim the ends of the tortilla roll with the butter knife, leaving ½ inch margin between the last toothpick and the end of the roll. Discard ends. Slide floss under the tortilla, perpendicular to the length of the roll. Place the floss halfway between two toothpicks. Cross the floss's two ends over the tortilla roll's top. Holding one end of the floss in each hand, pull the floss ends in opposite directions to slice. Continue slicing with floss to create 1 more pinwheel. Continue slicing with floss to create 1 more pinwheel. Continue slicing with floss to create 1 more pinwheel. Place the pinwheels on a plate.

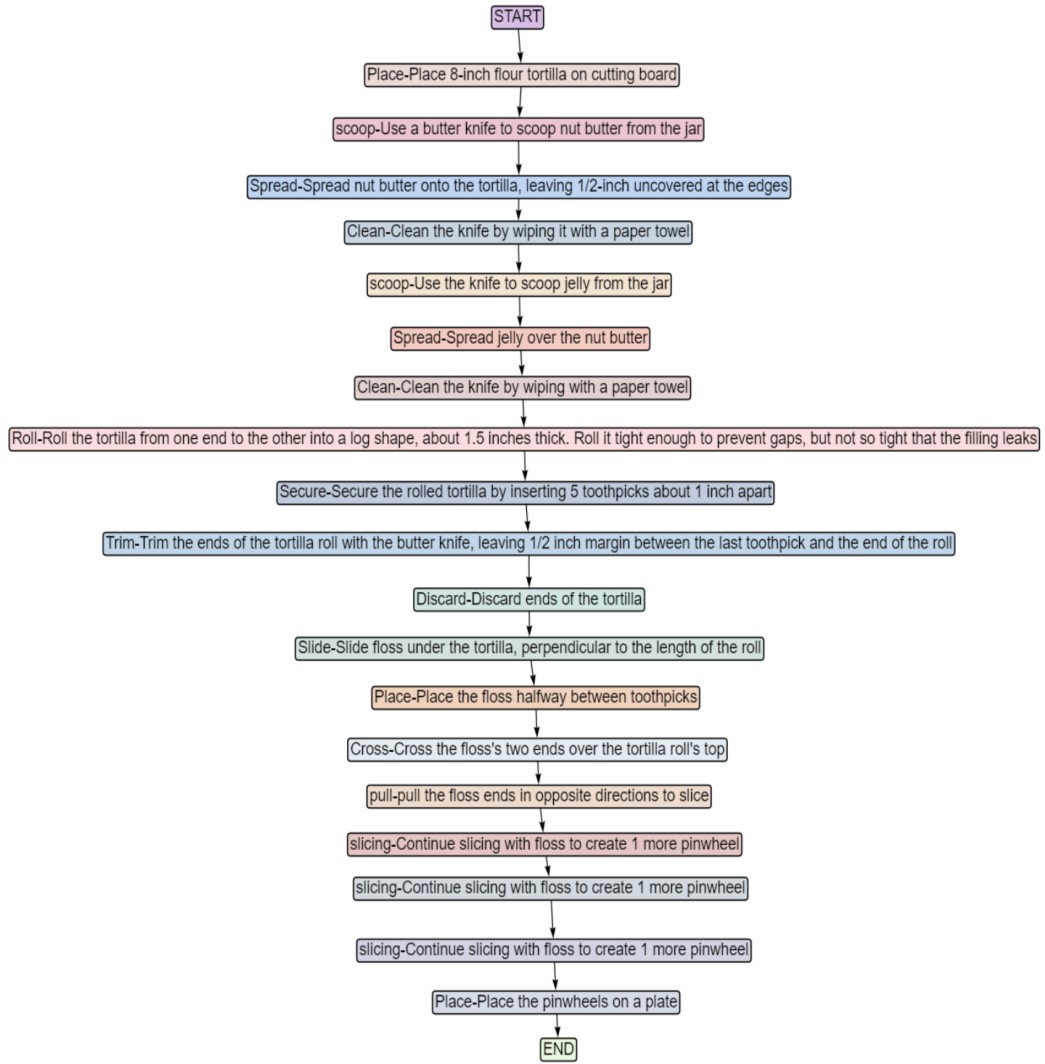

Figure 35: Picture illustrates constructed action-centric task graph for the recipe **Pinwheels**

F.1.16   RAMEN

**Recipe Description**    Remove the noodles from the package(Break Noodles / Keep them as a block). Peel and Chop 1 garlic clove on a cutting board. Peel 1 medium onion and slice 1/4 medium onion into pieces. Put all the Vegetables in a microwave-safe bowl. Add the noodles to the bowl and cover them with water. Cover with a lid (or paper towel) to prevent splattering. Microwave the ramen for 4 minutes. Add basil and cilantro to the bowl. Let the noodles sit for about 1 minute after the microwave stops. Mix in the flavour packet. Stir noodles with a spoon or fork until the flavouring dissolves.

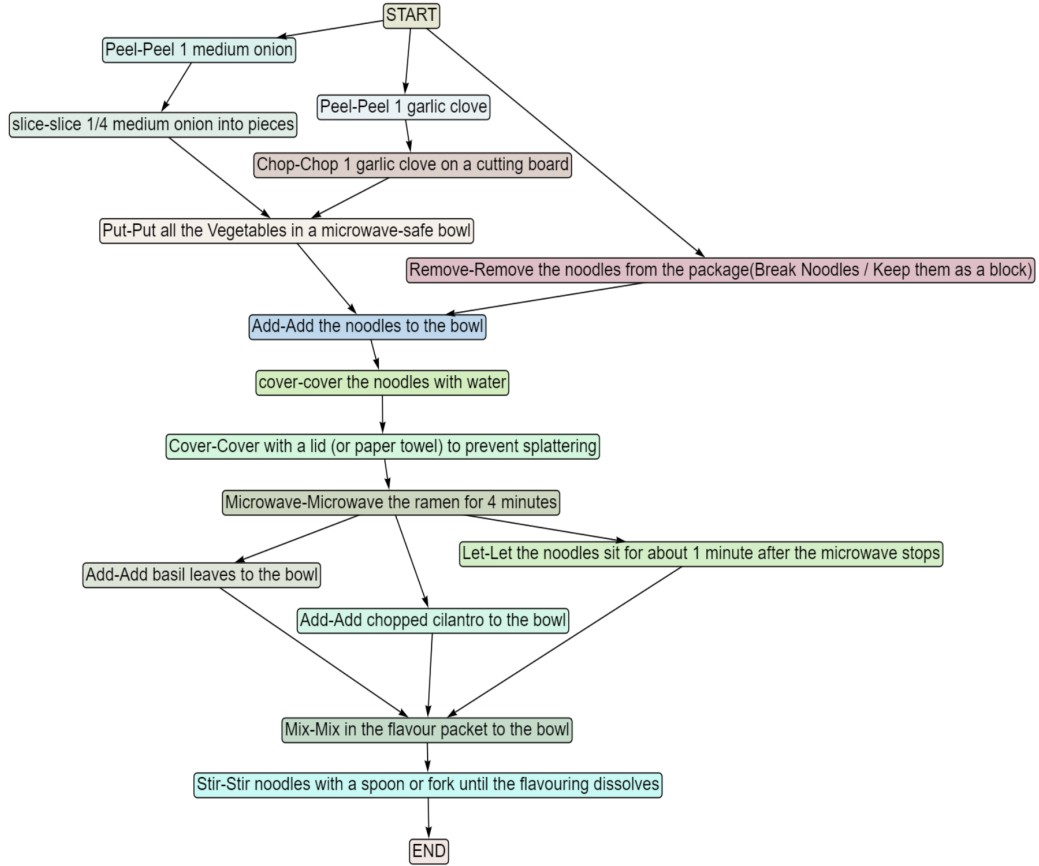

Figure 36: Picture illustrates constructed action-centric task graph for the recipe **Ramen**

### F.1.17  SAUTED MUSHROOMS

> **Recipe Description**  Chop 1 shallot. Rinse 3 mushrooms under cold water and immediately pat them dry with a paper towel. Pull out the stems. Slice the mushrooms. Heat 1 tbsp olive oil in a large skillet over medium-high heat. Once the pan is hot, add the mushrooms and cook for 3-5 minutes, stirring often, until they start to soften and brown. Add chopped shallot and cook, often stirring, for 1 minute. Peel 2 garlic cloves and mince them. Add 2 cloves of minced garlic and 1/2 tbsp balsamic vinegar and cook, often stirring, for 1 minute. Season with salt and pepper to taste. Serve.

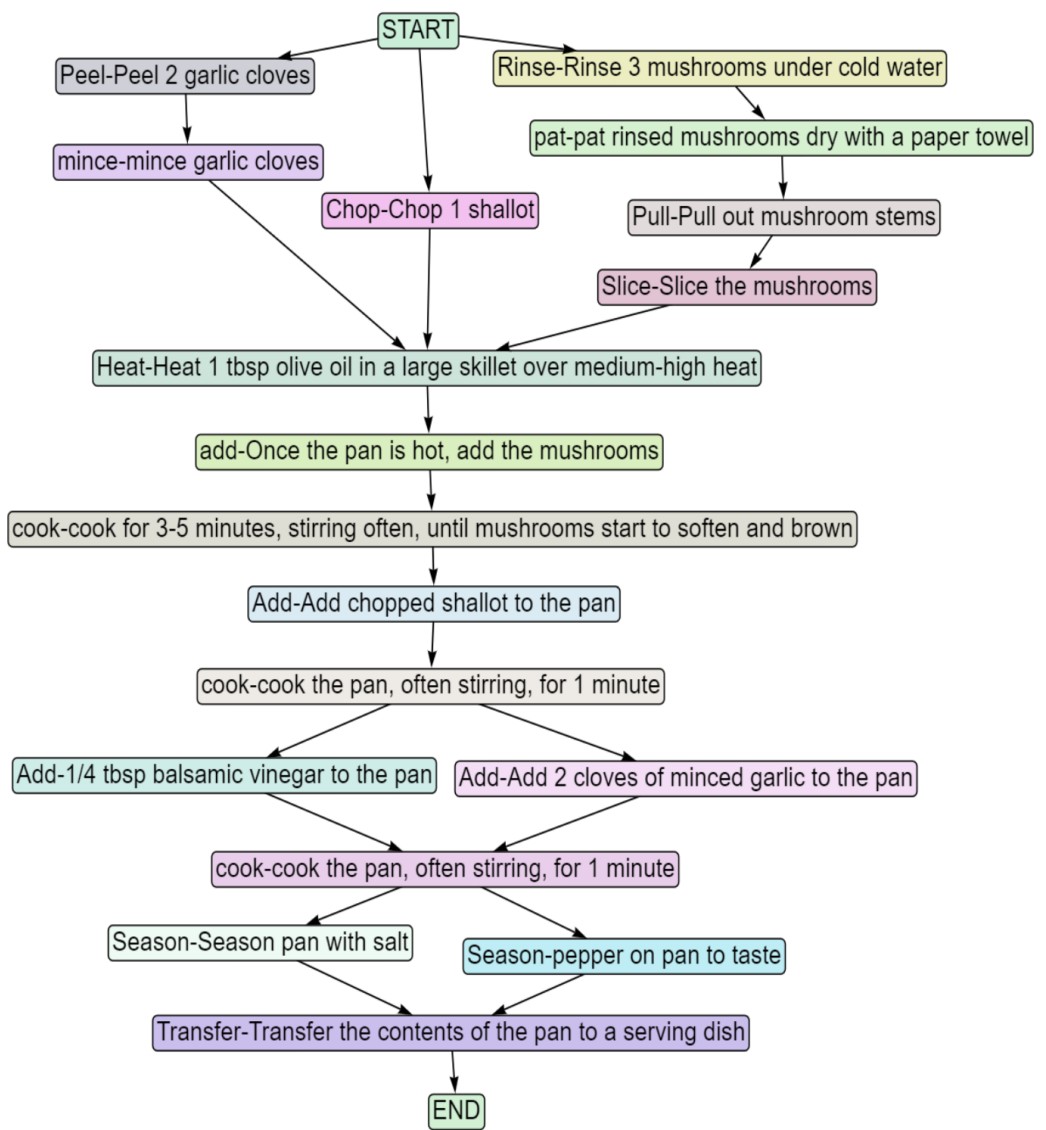

Figure 37: Picture illustrates constructed action-centric task graph for the recipe **Sauted Mushrooms**

### F.1.18   SCRAMBLED EGGS

**Recipe Description**   Chop 1/4 medium onion. Chop 1/4 tomato. Chop 1 green chili. Peel 2 garlic cloves. Mince peeled garlic cloves. Chop 1 tsp cilantro. Heat 2 tbsp oil in a heavy-bottomed or nonstick pan, and add chopped onions and 1/3 tsp salt. Sauté the onions on medium heat until they are soft and translucent. Add garlic, and chili to the pan. Cook for 1 minute, mixing everything. Add tomatoes and 1/8 tsp of turmeric to the pan. Cook covered for 1 minute or until the tomatoes soften. Whisk one egg with 1 tbsp milk and 1/3 tsp salt. Slowly pour the whisked eggs into the pan. Keep mixing with a spatula for 3 minutes or until the eggs are almost cooked. Garnish with 1 tbsp chopped cilantro and serve.

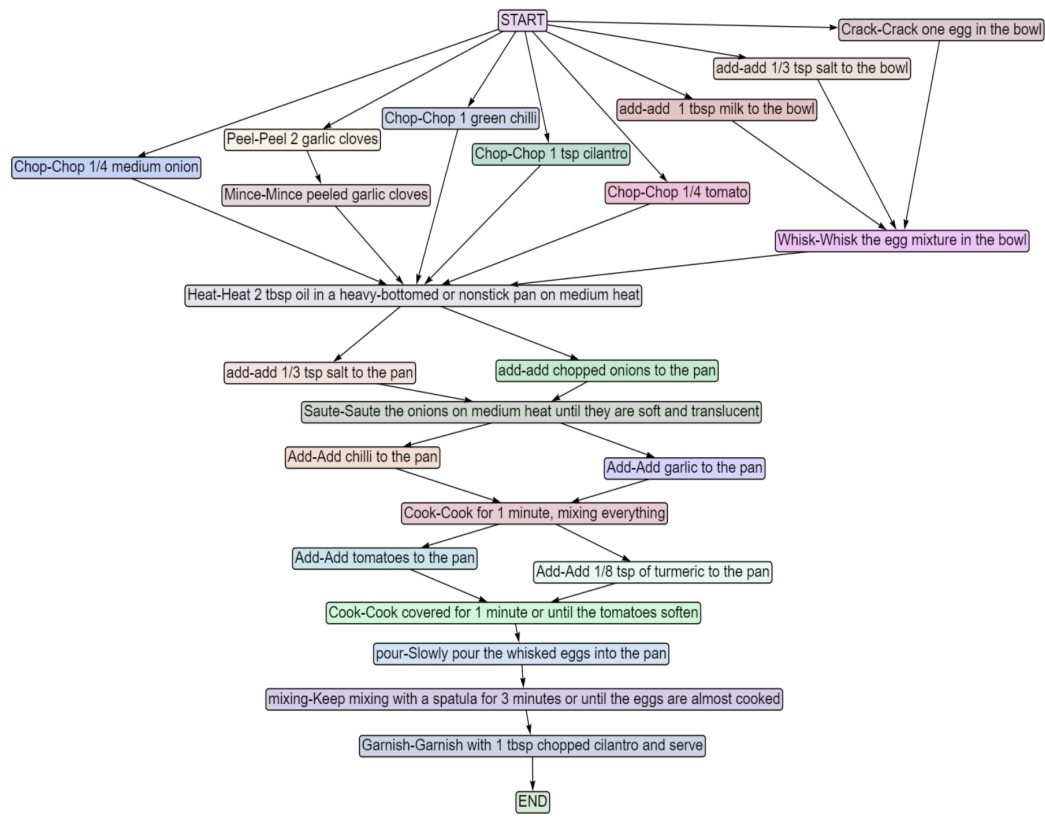

Figure 38: Picture illustrates constructed action-centric task graph for the recipe **Scrambled Eggs**

### F.1.19 SPICED HOT CHOCLATE

> **Recipe Description** Fill a microwave-safe mug with skimmed milk. Microwave the contents of the mug for 1 minute. Add 2 pieces of chocolate to the mug. Add a 1/5 teaspoon of cinnamon to the mug. Add 1 teaspoon of white sugar. Mix the contents of the mug. Heat the contents of the mug for 1 minute and serve.

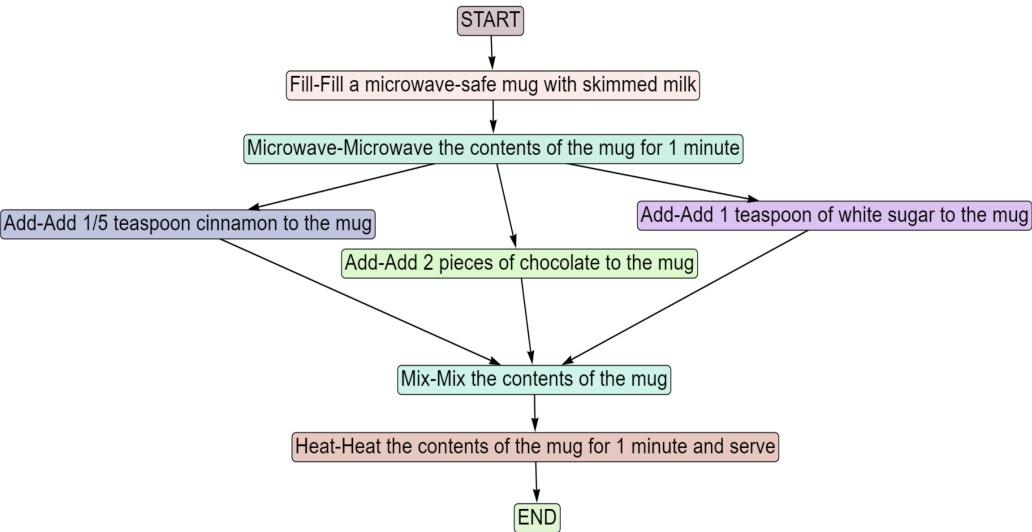

Figure 39: Picture illustrates constructed action-centric task graph for the recipe **Spiced Hot Choclate**

### F.1.20 SPICY TUNA AVOCADO WRAPS

**Recipe Description** Take 1 ripe avocado and cut it into thin slices. Chop 1 scallion. Open a can of tuna and drain excess water. Add 1 can drained tuna, 1/4 cup mayonnaise, 1 tsp Sriracha sauce, and chopped scallion in a bowl. Mix the contents of the bowl. Season with 1/4 tsp salt and 1/4 tsp pepper. Lay out 2 large lettuce leaves, place avocado slices on each leaf, and top with the tuna mixture. Roll up the lettuce wraps and secure the wrap with a toothpick.

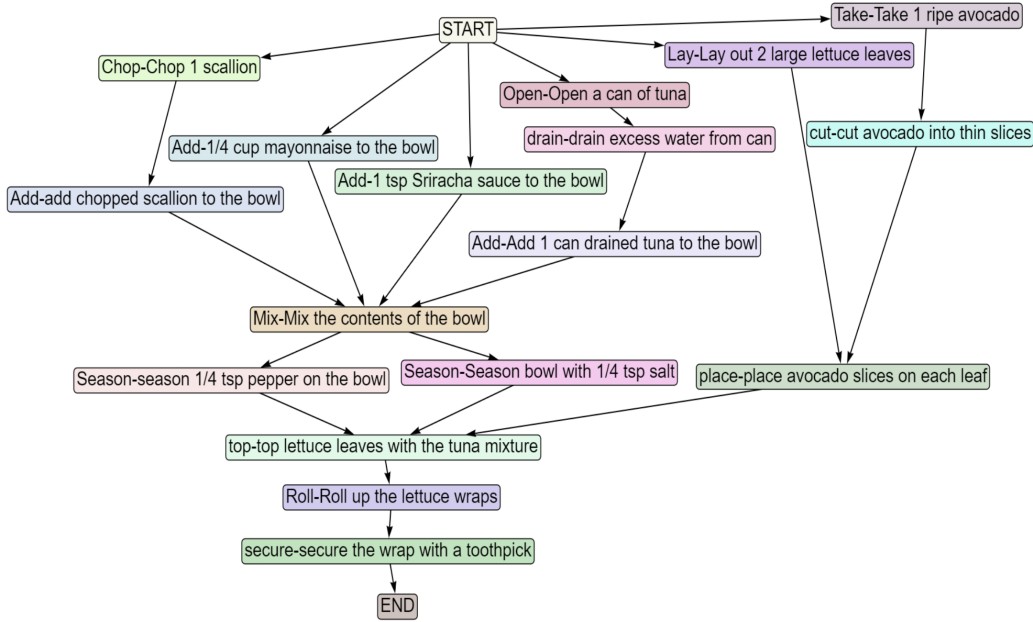

Figure 40: Picture illustrates constructed action-centric task graph for the recipe **Spicy Tuna Avocado Wraps**

### F.1.21 TOMATO CHUTNEY

> **Recipe Description**   Peel 4 large garlic cloves. mince the garlic. Take 1 tomato. Chop them up roughly and puree them without any water. Heat 3 tbsp oil in a pan over medium heat. Add 1/4 tsp mustard and 1/2 tsp cumin seeds. When the seeds begin to sizzle, add minced garlic. Lower the heat. Saute the garlic for 2-3 minutes. Add 2 tbsp red chili powder and mix well. Add tomato puree and 1/2 tsp salt. Mix well. Allow the mixture to simmer over low heat for 5 minutes or until the mixture becomes thick. Take the pan off the heat. Transfer it to a serving bowl.

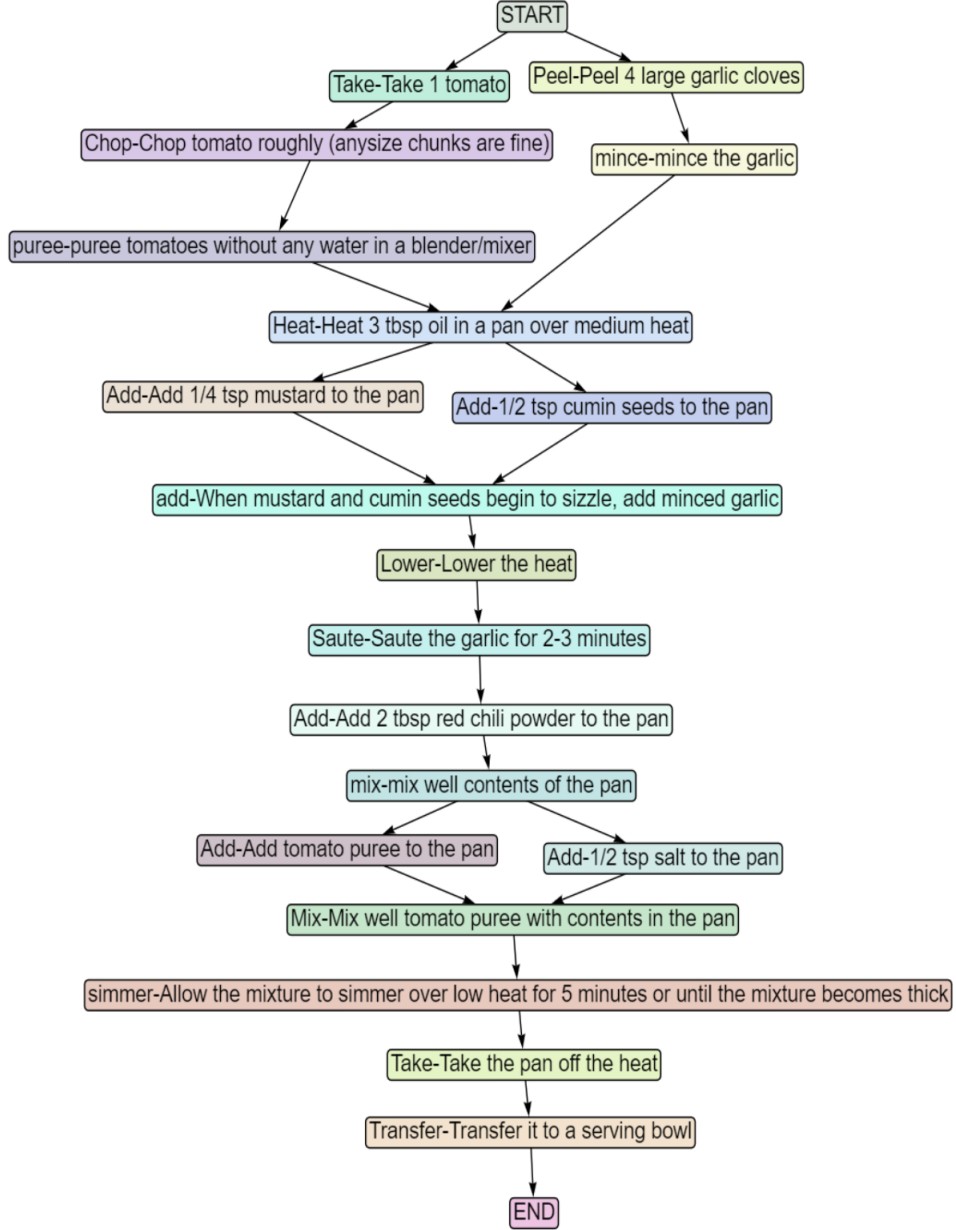

Figure 41: Picture illustrates constructed action-centric task graph for the recipe **Tomato Chutney**

### F.1.22 TOMATO MOZZARELLA SALAD

**Recipe Description** Rinse a tomato and gently dry it with a tea towel. Slice one tomato into about ½ inch thick slices. Place the thick slices of tomatoes on a platter, ensuring they only make a single layer. Season the tomato slices with salt. Sprinkle mozzarella cheese on top of the tomato throughout the platter. Garnish with italian seasoning. Season with ¼ teaspoon black pepper. Add a drizzle of extra-virgin olive oil, about 1 tablespoon, over the entire platter.

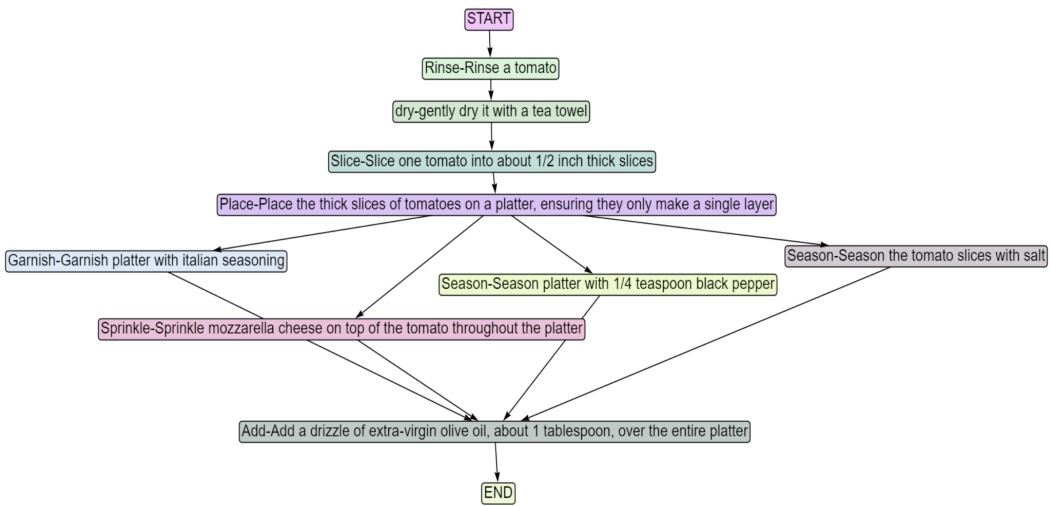

Figure 42: Picture illustrates constructed action-centric task graph for the recipe **Tomato Mozzarella Salad**

F.1.23 ZOODLES

> **Recipe Description**   Spiralize 1 medium zucchini into thin noodles using a spiralizer. Heat a large pan on medium-high heat. Melt 1 tablespoon of butter. Peel 1 garlic clove. Add 1 large minced garlic cloves. Cook garlic until fragrant (about 1-2 minutes). Be careful not to burn garlic. Add the zucchini noodles and 1/6 cup grated parmesan cheese and season with salt and pepper to taste. Cook for 2 minutes or until the zoodles are al dente. Remove from heat and serve immediately. Top with more parmesan if desired.

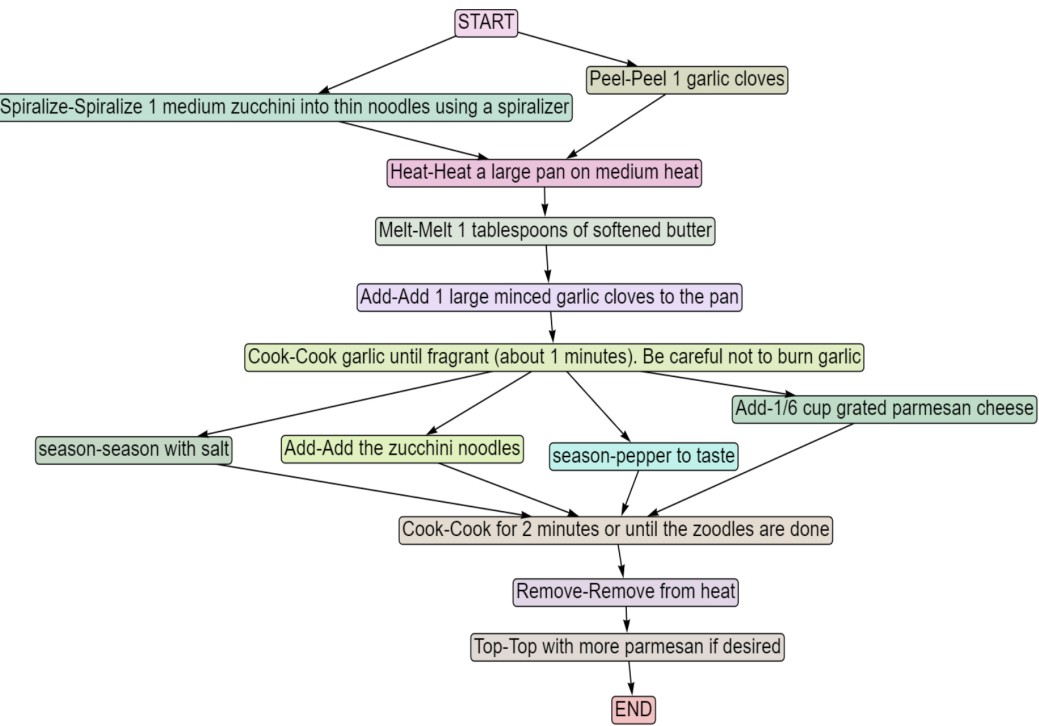

Figure 43: Picture illustrates constructed action-centric task graph for the recipe **Zoodles**

# G BENCHMARKING

## G.1 DATA SPLITS

In this paper, we evaluated our dataset on the following tasks: Supervised Error Recognition, Error Early Prediction, Multi-Step Localization, Procedure Learning and Zero-Shot Error Recognition. The following sections briefly describe the methods employed in learning models for the above tasks.

We provide splits based on recording environments, persons, recipes, and complete recordings to train supervised methods using our dataset.

- Split by recording environment
  - Our dataset comprises recordings from ten environments. But the bulk of our recordings are from 5 environments. So, we provide a split where we include recordings from environments with a larger proportion of data in the train/validation sets and recordings from environments with smaller proportions in the test set, leading to a split of 80% - 20%
- Split by recording persons
  - Eight participants compiled our dataset, and each participant recorded an equal number of videos. So, we provide a split where we include recordings of two participants who performed all the recipes in the test set and the recordings by the rest of the participants in the train/validation sets
- Split by recordings
  - Here, we categorize all the recordings of a recipe into train, validation and test sets based on a given ratio. This split based on recordings is generated randomly and varies each time.
- Split by recipes
  - This split is useful for learning tasks that require skill transfer. It was carefully created based on the skills needed for each recipe.

Furthermore, in tasks that require a semantic understanding of errors, methods can be employed to learn and distinguish between normal and error recordings. In such scenarios, learning only normal recordings and identifying errors in error recordings can be incorporated.

**Caveats** Since we rely on a tablet-based interface to provide the sequence of steps, we also provide 4K videos for the recordings. We are aware that an OCR-based system might be able to recognize the content in the tablet. To tackle that, we ensured that the test set included videos where participants looked at the complete text, not just the sequence of steps.

## G.2 SUPERVISED ERROR RECOGNITION

The objective of error recognition is to identify and flag errors in a given video. In our specific dataset, we divided each video into segments, where each segment corresponds to a distinct step in the recipe. For benchmarking error recognition, we mark a segment as *normal* if the corresponding (recipe) step was performed correctly; otherwise, we tag it as *error*.

### G.2.1 TRAINING AND EVALUATING BENCHMARK MODELS

We set up the error recognition task (namely given a video segment, classify it either as error or normal) as a supervised binary classification problem. The presence of a variety of error types makes solving this task particularly challenging.

**Datasets and Feature Extractors.** We employ features extracted by methods detailed in (Feichtenhofer et al., 2019; Feichtenhofer, 2020; Girdhar et al., 2022; Tong et al., 2022; Hara et al., 2017) to train our error recognition models.

Since the feature extractors require fixed-sized inputs (they are neural networks), we divided each video segment into contiguous 1-second sub-segments. The video segment may not always be

perfectly divisible by 1 second, and as a result the last sub-segment might be shorter than 1 second. To make this last sub-segment uniform, we use zero-padding, namely we add zeros at the end of the sub-segment to extend its duration to 1 second.

**Training.**    At training time, we assign the class label of a segment to all its sub-segments. This yields training and validation data for learning our proposed classifiers. Specifically, in our study, we used five pre-trained models: 3D-ResNet (Hara et al., 2017), SlowFast (Feichtenhofer et al., 2019), X3D (Feichtenhofer, 2020), VideoMAE (Tong et al., 2022) and Omnivore (Girdhar et al., 2022), which were used to solve video recognition tasks in prior studies, and replaced their output layer by a hidden layer followed by a sigmoid node (corresponding to the class). Then, we retrained these models using the training set and tuned the hyperparameters using the validation set.

In future, we will explore other options such as sub-sampling to ensure fixed-sized inputs for our feature extractors.

**Prediction.**    At prediction time, we again divide each segment into 1-second sub-segments, and after applying any necessary zero-padding, designate the class of the segment as the majority class of its sub-segments. Note that since our proposed method does not reason about order and missing step errors, we remove them from our evaluation set. In future, we plan to use neuro-symbolic methods that leverage background knowledge about error types in order to improve the performance of error recognition.

### G.2.2    Hyper-parameters

Throughout our experimental configuration, we maintained a uniform minibatch size of 512 instances.We employed ReLU activation functions in the hidden layers and sigmoid activation in the output layer. These networks were trained using the PyTorch framework (Paszke et al., 2019), with the training process executed on a single NVIDIA A40 GPU. We employed the Adam optimizer (Kingma & Ba, 2017) for training and a learning rate of 0.001. All models were trained for 100 epochs.

### G.2.3    Evaluating Baselines for Individual Error Types

Table 11 presents the performance metrics of five distinct models, each utilizing a unique feature extractor, across various error types for the Error Recognition Task. Despite high accuracy scores attributable to class imbalance, a close look at the recall values indicates that these models exhibit limitations in accurate error recognition. They perform well in classifying "normal" videos but fall short in detecting erroneous ones. Notably, the Omnivore model consistently excels over other models in both recall and F1 scores across different error types, thereby affirming its superior capability in identifying videos with errors.

Table 12 presents the performance of the models for the Early Error Prediction task. The low recall values reveal that these models face difficulty accurately predicting errors when presented with only partial video segments.

In summary, specialized approaches are crucial for effective error detection, especially for challenging error types such as "Temperature Error," "Measurement Error," and "Timing Error". Augmenting models with semantic information, task graphs, and other relevant attributes associated with errors is essential for achieving substantial improvements in performance.

### G.2.4    Further Evaluation of Error Recognition and Early Error Prediction Baselines

Figure 44a and 44b illustrate the Receiver Operating Characteristic (ROC) Curve for the baseline methods. In these visuals, a similar trend emerges where Omnivore outperforms other methods while the remaining approaches exhibit comparable performance levels. Moreover, the figure underscores that video understanding methods are not optimally suited for the error recognition task. Consequently, it emphasizes the requirement for more advanced and refined methods tailored to this task.

Figure 45 and 46 display the confusion matrices for all baseline methods in both Error Recognition and Early Error Prediction tasks. A critical observation is that all methods excel in correctly identifying

Table 11: Supervised Error Recognition: Baseline Evaluation Across Different Error Types

| Type of Error | Method Name | Accuracy | Precision | Recall | F1 Score |
|---|---|---|---|---|---|
| **Technique Error** | 3D ResNet | 88.56 | 27.91 | 18.18 | 22.02 |
| | Slowfast | 82.50 | 19.23 | 30.30 | 23.53 |
| | X3D | 83.31 | 9.72 | 10.61 | 10.14 |
| | VideoMAE | 84.39 | 19.18 | 22.22 | 20.59 |
| | Omnivore | 78.20 | 17.57 | 39.39 | 24.30 |
| **Preparation Error** | 3D ResNet | 89.77 | 9.30 | 9.76 | 9.52 |
| | Slowfast | 83.45 | 10.58 | 26.83 | 15.17 |
| | X3D | 87.21 | 12.50 | 21.95 | 15.93 |
| | VideoMAE | 86.99 | 13.70 | 27.03 | 18.18 |
| | Omnivore | 80.22 | 14.19 | 51.22 | 22.22 |
| **Measurement Error** | 3D ResNet | 88.43 | 6.98 | 6.12 | 6.52 |
| | Slowfast | 82.10 | 9.62 | 20.41 | 13.07 |
| | X3D | 85.33 | 8.33 | 12.24 | 9.92 |
| | VideoMAE | 84.39 | 8.22 | 12.77 | 10.00 |
| | Omnivore | 78.33 | 12.16 | 36.73 | 18.27 |
| **Temperature Error** | 3D ResNet | 93.14 | 0.00 | 0.00 | 0.00 |
| | Slowfast | 85.20 | 0.96 | 12.50 | 1.79 |
| | X3D | 89.23 | 0.00 | 0.00 | 0.00 |
| | VideoMAE | 88.58 | 1.37 | 12.50 | 2.47 |
| | Omnivore | 79.81 | 2.03 | 37.50 | 3.85 |
| **Timing Error** | 3D ResNet | 91.52 | 6.98 | 11.54 | 8.70 |
| | Slowfast | 84.12 | 5.77 | 23.08 | 9.23 |
| | X3D | 87.62 | 4.17 | 11.54 | 6.12 |
| | VideoMAE | 87.14 | 6.85 | 19.23 | 10.10 |
| | Omnivore | 79.00 | 6.08 | 34.62 | 10.34 |

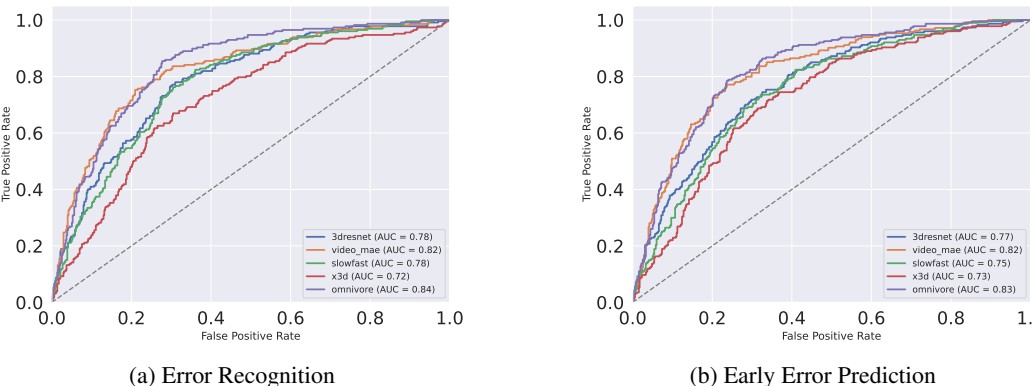

(a) Error Recognition  (b) Early Error Prediction

Figure 44: Evaluating Baseline Methods via ROC Curves

non-erroneous videos, as evidenced by high values in the first row and first column. Conversely, the detection of erroneous videos remains a significant challenge for all methods, highlighting the intrinsic complexity of the task.

### G.2.5  QUALITATIVE RESULTS FOR ERROR RECOGNITION

Figure 47 presents qualitative findings across seven examples, four of which contain errors, and three are from normal segments. The second column outlines the specific steps and highlights errors in red. Among these frames, the first, third, and fifth are from normal videos, which all models successfully classified. For the frames containing errors, one example was misclassified by all

Table 12: Early Error Prediction: Baseline Evaluation Across Different Error Types

| Type of Error | Method Name | Accuracy | Precision | Recall | F1 Score |
|---|---|---|---|---|---|
| **Technique Error** | 3D ResNet | 90.44 | 27.27 | 4.55 | 7.79 |
| | Slowfast | 88.83 | 20.69 | 9.09 | 12.63 |
| | X3D | 89.77 | 8.33 | 1.52 | 2.56 |
| | VideoMAE | 89.74 | 30 | 9.52 | 14.46 |
| | Omnivore | 87.75 | 20.93 | 13.64 | 16.51 |
| **Preparation Error** | 3D ResNet | 93 | 0 | 0 | 0 |
| | Slowfast | 92.19 | 20.69 | 14.63 | 17.14 |
| | X3D | 93.41 | 16.67 | 4.88 | 7.55 |
| | VideoMAE | 92.34 | 10 | 5.41 | 7.02 |
| | Omnivore | 90.85 | 18.6 | 19.51 | 19.05 |
| **Measurement Error** | 3D ResNet | 91.92 | 0 | 0 | 0 |
| | Slowfast | 89.77 | 3.45 | 2.04 | 2.56 |
| | X3D | 91.79 | 0 | 0 | 0 |
| | VideoMAE | 90.32 | 0 | 0 | 0 |
| | Omnivore | 88.16 | 4.65 | 4.08 | 4.35 |
| **Temperature Error** | 3D ResNet | 94.58 | 0 | 0 | 0 |
| | Slowfast | 95.4 | 5 | 25 | 8.33 |
| | X3D | 89.94 | 0 | 0 | 0 |
| | VideoMAE | 94 | 3.57 | 20 | 6.06 |
| | Omnivore | 94.39 | 7.14 | 40 | 12.12 |
| **Timing Error** | 3D ResNet | 95.29 | 9.09 | 3.85 | 5.41 |
| | Slowfast | 93.14 | 6.9 | 7.69 | 7.27 |
| | X3D | 95.15 | 8.33 | 3.85 | 5.26 |
| | VideoMAE | 93.64 | 5 | 3.85 | 4.35 |
| | Omnivore | 90.98 | 2.33 | 3.85 | 2.9 |

(a) 3D Resnet     (b) Slowfast     (c) X3D     (d) Omnivore

Figure 45: Confusion Matrices Demonstrating Baseline Method Performance in Supervised Error Recognition

baseline methods. Importantly, no error examples were correctly classified by all the methods. In one particularly challenging case, all methods failed to identify an error where butter fell outside the bowl, likely obscured by the background. These observations emphasize the inherent difficulty in error detection and indicate that future improvements may necessitate methods designed for semantic context understanding.

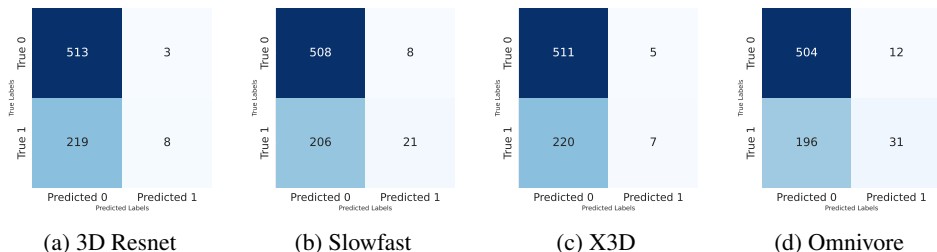

|     |     |     |     |
| --- | --- | --- | --- |
| (a) 3D Resnet | (b) Slowfast | (c) X3D | (d) Omnivore |

Figure 46: Confusion Matrices Demonstrating Baseline Method Performance in Early Error Detection

| Backbone | DivisionType | VideoType | tIoU 0.1 mAP | Recall@1x | Recall@5x | 0.3 mAP | Recall@1x | Recall@5x | 0.5 mAP | Recall@1x | Recall@5x |
| --- | --- | --- | --- | --- | --- | --- | --- | --- | --- | --- | --- |
| 3D ResNet | environment | Normal | 21.4 | 39.51 | 54.39 | 20.07 | 35.69 | 50.74 | 17.1 | 29.36 | 45.3 |
| | | Error | 9.74 | 15.31 | 23.2 | 8.31 | 12.69 | 21.45 | 6.22 | 9.08 | 16.57 |
| | person | Normal | 19.57 | 35.93 | 49.39 | 18.68 | 33.2 | 47.44 | 15.99 | 27.58 | 43.22 |
| | | Error | 13.82 | 27.14 | 39.57 | 12.94 | 23.4 | 37.32 | 10.88 | 19.21 | 33.64 |
| | recordings | Normal | 20.03 | 35.18 | 47.57 | 19.15 | 32.34 | 46.09 | 16.69 | 27.04 | 41.52 |
| | | Error | 13.22 | 25.96 | 37.84 | 12.48 | 23.47 | 36.07 | 10.8 | 19.5 | 31.76 |
| Slowfast | environment | Normal | 22.48 | 39.57 | 54.14 | 20.86 | 35.97 | 50.51 | 17.2 | 28.28 | 44.75 |
| | | Error | 10.11 | 16.16 | 23.32 | 9 | 13.02 | 20.39 | 7.53 | 9.54 | 15.83 |
| | person | Normal | 23.12 | 36.55 | 50.45 | 22.09 | 34.09 | 49.11 | 19.24 | 28.93 | 45.12 |
| | | Error | 14.78 | 26.68 | 39.97 | 14.14 | 24.73 | 37.71 | 12.56 | 21.76 | 34.37 |
| | recordings | Normal | 22.78 | 36.46 | 50.1 | 22.03 | 34.35 | 48.13 | 19.62 | 30.08 | 44.88 |
| | | Error | 14.11 | 27.52 | 39.19 | 13.34 | 24.91 | 37.19 | 11.9 | 21.53 | 32.39 |
| VideoMAE | environment | Normal | 24.44 | 38.22 | 52.48 | 22.97 | 34.77 | 49.51 | 18.67 | 28.57 | 42.68 |
| | | Error | 7.53 | 13.54 | 20.52 | 6.93 | 11.4 | 18.36 | 5.63 | 8.55 | 15.13 |
| | person | Normal | 26.78 | 37.43 | 46.28 | 25.68 | 34.79 | 44.6 | 22.02 | 29.43 | 39.81 |
| | | Error | 16.98 | 27.43 | 37.76 | 16.46 | 25.53 | 36.03 | 14.64 | 22.03 | 32.07 |
| | recordings | Normal | 26.27 | 37.15 | 46.93 | 24.71 | 34.06 | 45.03 | 21.51 | 29.36 | 40.44 |
| | | Error | 15.43 | 25.94 | 33.97 | 14.44 | 23.23 | 32.35 | 12.96 | 19.83 | 28.99 |
| Omnivore | environment | Normal | 34.65 | 47.91 | 60.63 | 33.06 | 44.77 | 58.36 | 28.59 | 38.38 | 51.9 |
| | | Error | 12.51 | 19.6 | 27.06 | 11.66 | 17.54 | 24.45 | 9.94 | 14.63 | 20.96 |
| | person | Normal | 32.5 | 44.45 | 52.47 | 31.13 | 41.53 | 50.91 | 28.39 | 37.03 | 47.97 |
| | | Error | 21.28 | 31.51 | 40.93 | 20.12 | 28.81 | 39.6 | 18.08 | 24.96 | 36.77 |
| | recordings | Normal | 30.22 | 42.43 | 52.11 | 28.94 | 39.47 | 50.49 | 25.15 | 32.65 | 46.51 |
| | | Error | 19.54 | 31.28 | 41.24 | 18.4 | 28.66 | 39.33 | 16.27 | 24.28 | 35.35 |

Table 13: This table compares results on using test sets with only error and normal videos. We note the superior performance of models when evaluated on test sets containing only normal videos.

## G.3   MULTI STEP LOCALIZATION

The task of multi-step localization entails simultaneous recognition and localization of steps performed in a procedural activity. Here, we cast this as a Temporal Localization problem where we extract features using pretrained action recognition models and train an ActionFormer head. Here, we provide further results obtained when (1) Use Omnivore to extract features of segments of lengths 1sec, 3sec, and 4 seconds (see Table 14) (2) Performance of trained models on the evaluation set where either only normal videos are present, or either error videos are present. (see Table 13)

| Backbone | Data split | 0.1 mAP | R@1x | R@5x | 0.3 mAP | R@1x | R@5x | 0.5 mAP | R@1x | R@5x |
| --- | --- | --- | --- | --- | --- | --- | --- | --- | --- | --- |
| omnivore 1s | E | 67.51 | 64.45 | 62.31 | 85.32 | 82.82 | 78.11 | 38.32 | 36.54 | 33.41 |
| | P | 75.96 | 73.35 | 70.34 | 92.14 | 90.51 | 88.24 | 45.82 | 44.12 | 41.16 |
| | R | 73.71 | 71.45 | 68.14 | 92.08 | 89.82 | 86.38 | 42.76 | 40.52 | 37.19 |
| omnivore 3s | E | 72.99 | 70.05 | 66.57 | 86.03 | 83.68 | 81.02 | 43.47 | 41.83 | 38.87 |
| | P | 78.63 | 76.96 | 74.61 | 93.27 | 91.23 | 89.18 | 50.25 | 48.54 | 44.85 |
| | R | 76.82 | 74.9 | 71.94 | 91.33 | 89.61 | 88.11 | 49.23 | 47.84 | 44.76 |
| omnivore 4s | E | 71.85 | 69.79 | 64.93 | 88.12 | 86.33 | 83.15 | 43.13 | 41.54 | 38.95 |
| | P | 79.33 | 77.39 | 74.24 | 93.46 | 91.67 | 89.95 | 50.69 | 49.49 | 46.19 |
| | R | 78.61 | 76.59 | 73.81 | 93.04 | 90.99 | 88.8 | 50.24 | 48.62 | 45.64 |

Table 14: This table presents results on features extracted with varying length segments using Omnivore. We observe that when features are extracted using 4sec, our trained ActionFormer performs much better compared to the ones trained on features extracted using 1sec and 3sec video segments.

## G.4 ZERO SHOT ERROR RECOGNITION

Table 15: Zero Shot Error Recognition

| Method | AUC | EER |
|---|---|---|
| **SSMCTB**(Madan et al., 2022) | 50.65 % | 49.65 % |
| **SSPCAB**(Ristea et al., 2022) | 50.25 % | 49.74 % |

To introduce an additional baseline for error recognition, we reformulate the task as a self-supervised problem focused on frame-level error recognition. More specifically, we use anomaly detection methods to classify each frame in each video as either normal or abnormal, where the latter is defined as an instance that deviates from the expected behavior (the frame where participants made errors).

We used two self-supervised anomaly detection methods from literature, self-supervised masked convolutional transformer block (SSMCTB) (Madan et al., 2022) and self-supervised predictive convolutional attentive block (SSPCAB) (Ristea et al., 2022), and trained them on top of ResNet-50 (He et al., 2015), where the latter serves as a neural, image-based feature extractor. Both models were trained using reconstruction loss (Madan et al., 2022). We used normal recordings for training and both normal and error recordings for testing. We evaluated the benchmark models using the frame-level area under the curve (AUC) and Equal Error Rate (EER) scores. Table 15 shows the results. We observe that SSMCTB is slightly better than SSPCAB. The AUC scores displayed in this context demonstrate only marginal improvement over random chance. This emphasizes the considerable difficulty of the task and underscores the necessity for more specialized approaches to achieve effective error recognition in a self-supervised manner.

## G.5 PROCEDURE LEARNING

Move to Appendix: Action segmentation differs from procedure learning, which focuses on segmenting actions without considering their relevance to task completion; on the contrary, procedure learning aims to find commonalities among the key steps needed to complete the task captured in multiple videos.

Table 16: **Procedure Learning.** The results showcase the performance of models trained using methods $\mathcal{M}_1$ (Dwibedi et al., 2019) and $\mathcal{M}_2$ (Bansal, Siddhant et al., 2022). Where $\mathcal{M}_1$ employs Cycleback Regression Loss ($\mathcal{C}$) and $\mathcal{M}_1$ employs a combination of both Cycleback Regression Loss ($\mathcal{C}$) and Contrastive - Inverse Difference Moment Loss ($\mathscr{C}$). We note that we only train embedded networks using loss functions from these methods and retain the Pro-Cut Module for assigning frames to key steps. Here, $\mathcal{P}$ represents precision, $R$ represents recall, and $I$ represents IOU.

| Recipe | Random | | | $\mathcal{M}_1$ | | | $\mathcal{M}_2$ | | |
|---|---|---|---|---|---|---|---|---|---|
| | $\mathcal{P}$ | $\mathcal{R}$ | $\mathcal{I}$ | $\mathcal{P}$ | $\mathcal{R}$ | $\mathcal{I}$ | $\mathcal{P}$ | $\mathcal{R}$ | $\mathcal{I}$ |
| BlenderBananaPancakes | 7.40 | 3.83 | 2.26 | 12.65 | 9.50 | 5.16 | 15.54 | 9.96 | 5.72 |
| BreakfastBurritos | 9.66 | 4.04 | 2.59 | 18.72 | 11.46 | 6.77 | 16.58 | 10.77 | 5.87 |
| BroccoliStirFry | 4.21 | 3.81 | 1.73 | 9.92 | 9.11 | 3.93 | 8.20 | 8.10 | 3.85 |
| ButterCornCup | 8.37 | 3.91 | 2.16 | 13.82 | 11.85 | 5.79 | 15.07 | 12.30 | 5.82 |
| CapreseBruschetta | 9.34 | 3.96 | 2.52 | 25.55 | 12.89 | 7.52 | 20.53 | 9.09 | 5.59 |
| CheesePimiento | 9.10 | 3.87 | 2.41 | 19.74 | 10.48 | 6.44 | 17.49 | 10.32 | 6.26 |
| Coffee | 6.54 | 3.87 | 2.17 | 13.68 | 9.91 | 5.49 | 15.76 | 10.25 | 5.63 |
| CucumberRaita | 8.90 | 3.64 | 2.44 | 13.58 | 7.92 | 5.14 | 16.15 | 9.97 | 6.09 |
| DressedUpMeatballs | 7.28 | 3.80 | 2.26 | 15.20 | 10.80 | 6.05 | 17.59 | 10.27 | 5.81 |
| HerbOmeletWithFriedTomatoes | 6.82 | 4.05 | 1.98 | 14.66 | 14.98 | 5.50 | 14.64 | 11.34 | 6.29 |
| MicrowaveEggSandwich | 8.81 | 3.98 | 2.61 | 16.25 | 10.44 | 6.16 | 19.16 | 11.29 | 6.99 |
| MicrowaveFrenchToast | 9.03 | 3.74 | 2.49 | 16.82 | 7.90 | 5.07 | 17.31 | 8.82 | 5.66 |
| MicrowaveMugPizza | 7.53 | 3.90 | 2.38 | 12.82 | 9.78 | 5.27 | 12.69 | 9.18 | 5.18 |
| MugCake | 5.45 | 4.00 | 2.12 | 16.12 | 12.95 | 6.87 | 10.32 | 8.85 | 4.40 |
| PanFriedTofu | 5.35 | 3.97 | 1.54 | 8.86 | 10.39 | 3.75 | 9.34 | 12.44 | 3.87 |
| Pinwheels | 6.54 | 4.28 | 2.13 | 13.58 | 11.96 | 5.92 | 16.08 | 13.06 | 7.05 |
| Ramen | 6.85 | 4.12 | 1.87 | 11.09 | 9.97 | 4.48 | 12.90 | 10.92 | 5.07 |
| SautedMushrooms | 6.08 | 3.81 | 2.02 | 15.06 | 12.22 | 6.16 | 19.54 | 13.83 | 7.42 |
| ScrambledEggs | 4.74 | 3.95 | 1.89 | 11.11 | 11.08 | 5.27 | 11.70 | 10.96 | 5.27 |
| SpicedHotChocolate | 14.08 | 3.82 | 3.09 | 29.82 | 10.58 | 8.49 | 29.79 | 11.04 | 8.74 |
| SpicyTunaAvocadoWraps | 6.25 | 3.90 | 2.21 | 15.62 | 10.52 | 5.67 | 12.47 | 9.61 | 5.25 |
| TomatoChutney | 5.45 | 3.89 | 1.85 | 12.25 | 10.68 | 5.42 | 12.25 | 10.68 | 5.42 |
| TomatoMozzarellaSalad | 10.88 | 3.91 | 2.38 | 19.77 | 10.21 | 6.01 | 19.20 | 10.48 | 5.96 |
| Zoodles | 7.91 | 4.08 | 2.22 | 18.32 | 12.80 | 6.37 | 18.32 | 12.80 | 6.37 |
| **Average of 24 recipes** | **7.61** | **3.92** | **2.22** | **15.62** | **10.85** | **5.78** | **15.78** | **10.68** | **5.82** |

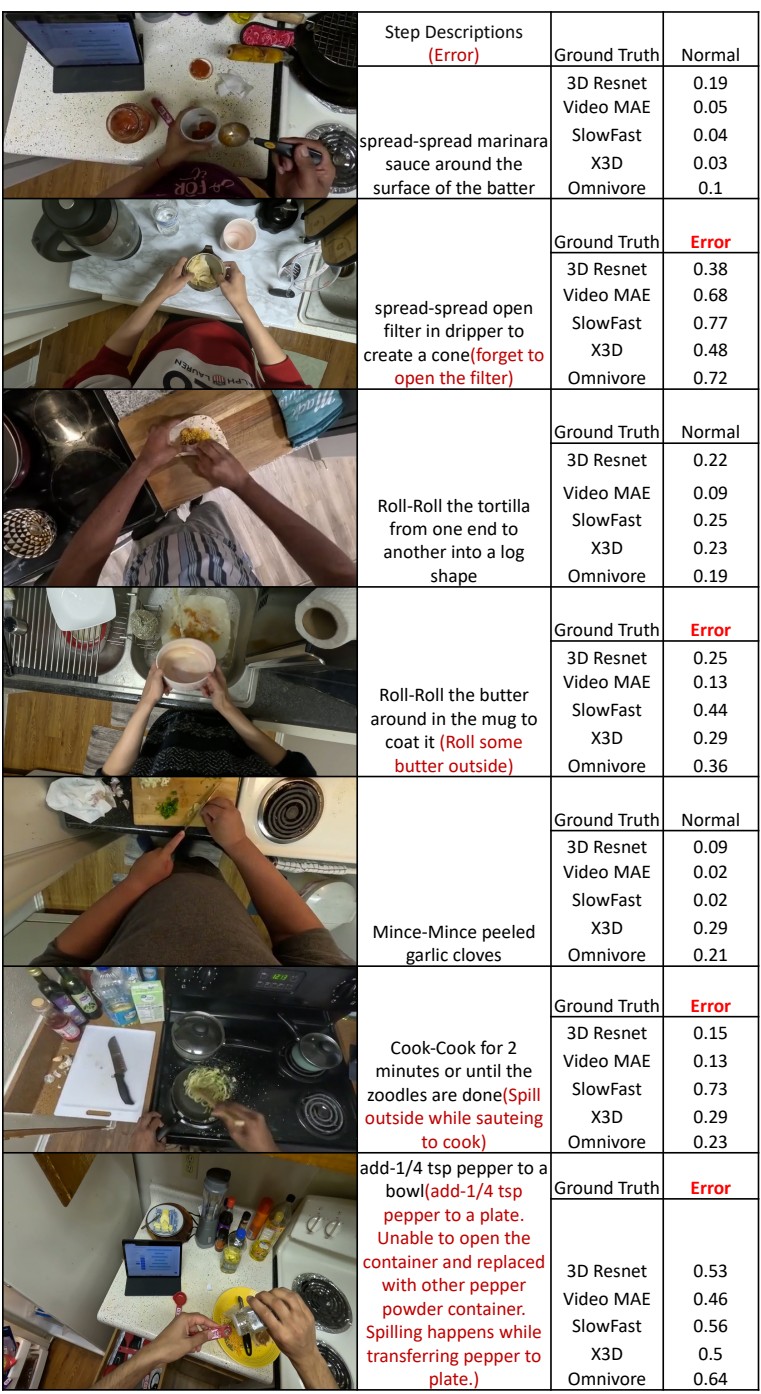

| | Step Descriptions (Error) | Ground Truth | Normal |
|---|---|---|---|
| | | 3D Resnet | 0.19 |
| | | Video MAE | 0.05 |
| | spread-spread marinara sauce around the surface of the batter | SlowFast | 0.04 |
| | | X3D | 0.03 |
| | | Omnivore | 0.1 |
| | | Ground Truth | **Error** |
| | | 3D Resnet | 0.38 |
| | spread-spread open filter in dripper to create a cone(forget to open the filter) | Video MAE | 0.68 |
| | | SlowFast | 0.77 |
| | | X3D | 0.48 |
| | | Omnivore | 0.72 |
| | | Ground Truth | Normal |
| | | 3D Resnet | 0.22 |
| | Roll-Roll the tortilla from one end to another into a log shape | Video MAE | 0.09 |
| | | SlowFast | 0.25 |
| | | X3D | 0.23 |
| | | Omnivore | 0.19 |
| | | Ground Truth | **Error** |
| | | 3D Resnet | 0.25 |
| | Roll-Roll the butter around in the mug to coat it (Roll some butter outside) | Video MAE | 0.13 |
| | | SlowFast | 0.44 |
| | | X3D | 0.29 |
| | | Omnivore | 0.36 |
| | | Ground Truth | Normal |
| | | 3D Resnet | 0.09 |
| | | Video MAE | 0.02 |
| | | SlowFast | 0.02 |
| | | X3D | 0.29 |
| | Mince-Mince peeled garlic cloves | Omnivore | 0.21 |
| | | Ground Truth | **Error** |
| | | 3D Resnet | 0.15 |
| | Cook-Cook for 2 minutes or until the zoodles are done(Spill outside while sauteing to cook) | Video MAE | 0.13 |
| | | SlowFast | 0.73 |
| | | X3D | 0.29 |
| | | Omnivore | 0.23 |
| | add-1/4 tsp pepper to a bowl(add-1/4 tsp pepper to a plate. Unable to open the container and replaced with other pepper powder container. Spilling happens while transferring pepper to plate.) | Ground Truth | **Error** |
| | | 3D Resnet | 0.53 |
| | | Video MAE | 0.46 |
| | | SlowFast | 0.56 |
| | | X3D | 0.5 |
| | | Omnivore | 0.64 |

Figure 47: The first column presents visual representations or video images. The second column offers descriptions of each step and uses red text to highlight any errors present. The third column outlines the methods evaluated. The last column displays the ground truth, indicating whether it's an Error or Normal segment. The following rows show the predicted probabilities for the given example. A class label of 1 implies an error; Examples identified with higher probabilities are classified as errors.

