# OpenReview forum: "Put on your detective hat: What’s wrong in this video?"
_ICLR.cc/2024/Conference — Submitted to ICLR 2024_

### Official Review · Reviewer_tDxh · 2023-10-28

**Soundness:** 2 fair
**Presentation:** 3 good
**Contribution:** 2 fair
**Rating:** 5
**Confidence:** 4

**Summary:**

This paper presents a new dataset for error recognition in procedure videos. Table 1 compares this new dataset to existing datasets. The process of data collection is described. Baselines are provided for the two tasks of "Error Recognition" and "Multi Step Localization".

**Strengths:**

The paper is well presented: (1) the motivation is clear (2) the data collection process is clear.

The paper provides baseline methods for the proposed two tasks: (1) error recognition (2) multi-step localization.

**Weaknesses:**

1. The additional benefit of this new dataset is unclear. Looking at Table 1, the added benefit compared to existing work "Assembly 101" seems to be limited to adding "depth" channel. However, it is unclear whether the "depth" channel is really useful in the context of error recognition. The authors need to provide more evidence on why this new dataset is significantly more useful than "Assembly 101".

2. The collected sensor data such as "depth", "camera trajectory", and "hand joints" are not used or analyzed by the baselines provided.

3. The AUC score between Table 2 (Error Recognition) & Table 3 (Early Error Prediction) looks surprisingly similar. Shouldn't the task of "Early Error Prediction" be much harder than the typical "Error Recognition" task?

**Questions:**

1. In Figure 4, Table 2-3, can you highlight the best performing model?

2. It's better to present the the distribution of different error categories in the main paper, e.g., Figure 16-17?

---

> ### Author Response · Authors · 2023-11-23
> **Author response to Reviewer tDxh**
>
> > The additional benefit of this new dataset is unclear.
>
> Firstly, Error recognition as a task is quite challenging, and we conjecture that it requires the development of techniques that understand the context, meaning, and cause of various errors.
>
> Our dataset is different from Assembly 101 in the following ways:
>
> - **Domain:** We stress the fact that Assembly 101 explores the domain of Assembly and Disassembly tasks, and we focus on Cooking activities.
>    - In assembly, the majority of the errors performed can be corrected, but cooking as a domain includes activities which do not have corrective actions.
>    - Unlike assembly, where shapes and colours of the objects remain fixed during the whole procedure. Cooking involves continuous changes in the shape and colour of ingredients.
> - **Environment:** Assembly 101 is a dataset that is collected in a lab environment, and we collect our dataset in real kitchen environments.
> - **Modalities:** As we primarily employ HoloLens2 in capturing the data, we leverage the wealth of information it enables to capture. For example, our synchronized dataset enables the complete reconstruction of a 3D map of the environment while the activity is being performed.
>
> In conclusion, we emphasize that the category of errors we explore is different from the category of errors explored by activities in the assembly domain.
>
> ---
>
> > The collected sensor data such as "depth", "camera trajectory", and "hand joints" are not used or analyzed by the baselines provided.
>
> We believe that the task of error recognition is a challenging task and requires complete awareness of the 3D environment. So, we employed Hololens2 to capture the data. In the current version of the manuscript, we do not have results on incorporating data captured from other modalities into the proposed tasks. We shall add the results incorporating data captured from all the modalities in future versions of this manuscript.
>
> ---
>
> > The AUC score between Table 2 (Error Recognition)
>
> Notice that although the AUC scores are roughly the same, the F1 scores are always lower because recall is low.
>
> ---
>
> > In Figure 4, Table 2-3, can you highlight the best-performing model?
>
> Sure, we have updated the manuscript following your suggestion
>
> ---
>
> > It's better to present the distribution of different error categories in the main paper, e.g., Figure 16-17.
>
> Sure, we have updated the manuscript following your suggestion

---

### Official Review · Reviewer_bmra · 2023-11-02

**Soundness:** 2 fair
**Presentation:** 3 good
**Contribution:** 2 fair
**Rating:** 5
**Confidence:** 4

**Summary:**

The paper proposes an Egocentric 4D (RGBD) dataset, for understanding step by step procedural activities and the error recognition. The dataset intends to tackle the problem of fine-grained error detection in activities, also anticipating them. The end goal is to mimic activities error in various activities like medical operation or chemical operation. The dataset consists of kitchen recipe videos (384 videos) for 24 tasks, by 8 different actors. The annotation of the datasets ranges from coarse-grained: correct vs error (binary classification) to fine-grained errors on various time stamps of error. The paper also proposes 3 baseline performance metrics : binary error classification, multi-step localization (TAL) and procedural learning.

**Strengths:**

•	Paper is very written in simple terms, easy to follow language.

•	Proposed errors like technique error, and measurement error will test models fine-grained understanding of activities

•	The work is definitely a constructive work towards Coarse grained & Fine grained errors understanding problem, which definitely seems like relatively new field.

•	Extensive fine-grained annotation would greatly benefit future works.

•	Higher ratio of error to normal videos, establishes superiority of the work.

**Weaknesses:**

•	Compared to Assembly 101 (error detection), the paper seems like an inferior / less complicated dataset. Claims like higher ratio of error to normal videos needs to be validated.

•	Compared to datasets, the dataset prides itself on adding different modalities especially depth channel (RGB-D). The paper fails to validate the necessities of such modality. One crucial different between assembly dataset is use of depth values. What role does it play in training baseline models? Does it boost the model’s performance if these weren’t present. In current deep learning area, depth channels should be reasonably be producible via the help of existing models.

•	I’m not convinced that the binary classification is a justifiable baseline metrics. While I agree with the TAL task is really important here and a good problem to solve, I’m not sure how coarse grained binary classification can assess models understanding of fine-grained error like technique error.

•	Timing Error (Duration of an activity) and Temperature based error, does these really need ML based solutions? In sensitive tasks, simple sensor reading can indicate error. I’m not sure testing computer vision models on such tasks is justifiable. These require more heuristics-based methods, working with if-else statement.

•	Procedure Learning: its very vaguely defined, mostly left unexplained and seems like an after thought. I recommend authors devote passage to methods “M1 (Dwibedi et al., 2019)” and “M2 (Bansal, Siddhant et al., 2022)”. In Table 5, value of lambda? Is not mentioned.

•	The authors are dealing with a degree of subjectivity in terms of severity of errors. It would have been greatly beneficial, if the errors could be finely measured. For example if the person uses a tablespoon instead of teaspoon, is still an error? Some errors are more grave than others, is there a weighted scores? Is there a way to measure level of deviation for each type of error or time stamp of occurrence of error. Is one recipe more difficult than the other recipe.

**Questions:**

1.	Please validate claims like higher ratio of error to normal videos

2.	Please provide the utility of depth channels in error detection tasks or provide baseline performances.

3.	I feel like binary classification section can be minimized, and instead procedural activity section needs to be emphasized and described in much more detail.

4.	Show baseline results for each type of errors. A measure of how difficult certain error would help future work.

5.	Some form of comparisons of errors needs to be there, or a severalty level of errors.

---------------

Post Rebuttal

-------------

1.	In terms of statistics, still not convinced if the proposed dataset is better than Assembly 101. None of the baseline models have been shown to be have any adverse impact of these proposed added difficulties : “Shape and color changes ingredients”, “real-kitchen environments”. Higher ratio of error to normal videos (not validated). Reviewer I weakness-1 pointed out the same issue.

2.	Added additional modality of Depth is still not resolved. As a reviewer, I’m not sure what role depth channel plays in the error detection task. The authors have mentioned again in the rebuttal of presence of depth channel as a superiority over Assembly 101 dataset, but no proof how it’s useful with the task at hand (i.e. error detection) or does Depth channel play any role in this dataset. Reviewer I weakness-1 & Reviewer 3 weakness-1 (&2) pointed out the same issue.

3.	Binary classification of the as an error detection baseline metrics is not justified. (Similar issue pointed out by Reviewer I weakness 2). I fully agree that the baselines do not provide many insights.

4.	I’m not entirely convinced metrics based (Time / Temperature) errors are the right metrics to measure error detection capabilities of via Commuter Vision or Machine Learning algorithms. (Similar issue pointed out by Reviewer I weakness 2)

---

> ### Author Response · Authors · 2023-11-23
> **Author response to Reviewer bmra**
>
> > Compared to Assembly 101 (error detection), the paper seems like an inferior / less complicated dataset. Claims like a higher ratio of error to normal videos need to be validated.
>
> Firstly, Error recognition as a task is quite challenging, and we conjecture that it requires the development of techniques that understand the context, meaning, and cause of various errors. Secondly, we respectfully deny the statement that this paper is an inferior dataset compared to Assembly 101. We justify the statement with the following reasons
>
> - **Domain:** We stress the fact that Assembly 101 explores the domain of Assembly and Disassembly tasks, and we focus on Cooking activities.
>   - In assembly, the majority of the errors performed can be corrected, but cooking as a domain includes activities which do not have corrective actions.
>   - Unlike assembly, where shapes and colours of the objects remain fixed during the whole procedure. Cooking involves continuous changes in the shape and colour of ingredients.
> - **Environment:** Assembly 101 is a dataset which is collected in a lab environment, and we collect our dataset in real-kitchen environments.
> - **Modalities:** As we primarily employ HoloLens2 in capturing the data, we leverage the wealth of information it enables to capture. For example, our synchronized dataset enables the complete reconstruction of a 3D map of the environment while the activity is being performed.
>
> In conclusion, we emphasize that the category of errors we explore is different from the category of errors explored by activities in the assembly domain.
>
> ---
>
> > Compared to datasets, the dataset prides itself on adding different modalities especially depth channel (RGB-D). The paper fails to validate the necessities of such modality.
>
> We believe that the task of error recognition is a challenging task and requires complete awareness of the 3D environment. So, we employed Hololens2 to capture the data.
>
> You are right that the current monocular depth estimation techniques are indeed quite powerful and can equal the capacity of a low-quality depth sensor. But, we leverage the wealth of information provided by Hololens2, thus enabling the creation of a less noisy 3D map of the environment.
>
> In the current version of the manuscript, we do not have results on incorporating depth data into the proposed tasks. We shall add the results incorporating data captured from all the modalities in future versions of this manuscript.
>
> ---
>
> > I’m not convinced that the binary classification is a justifiable baseline metrics
>
> We do agree that binary classification is a preliminary baseline and conjecture that any technique developed for the task of error recognition should interpret the action, understand the context and infer the cause.
>
> As this manuscript introduces the dataset, we followed the convention of providing a supervised baseline using transfer learning techniques for the specific tasks.
>
> ---
>
> > Timing Error (Duration of an activity) and Temperature based error, does these really need ML based solutions? In sensitive tasks, simple sensor reading can indicate error.
>
> It is an interesting question. In our experiments (see Table 5) on the proposed tasks and categorization of common errors that could occur while cooking, we believe that timing, temperature and measurement are the most challenging errors to detect.
>
> Whenever there's a conditional transition based on a change of state involving a heating component, sensor-based measurements fall short. For example, \textit{Saute the tofu on a pan for 3 minutes or until it turns brown}. Here, having a perception component that is capable of judging the state change to brown is necessary.
>
> ---
>
> > Procedure Learning: its very vaguely defined, mostly left unexplained
>
> We implemented your suggestion and added an extra paragraph explaining this section. Please look at the updated section on Procedure Learning.
>
> ---
>
> > The authors are dealing with a degree of subjectivity in terms of severity of errors. It would have been greatly beneficial,
>
> In the current version of the manuscript, we flag any deviation from the procedure as an error. As stated in the above question, error recognition completely depends on the context; your example of using a tablespoon instead of a teaspoon may not be a significant error in the procedure when the step is to **stir**. However, the step involves **adding an ingredient like salt**, which completely changes the taste of the recipe, and this is indeed very significant.
>
> We do understand that not all errors have the same impact on the procedure. It is an interesting idea to automatically infer the impact of an error on the current activity.
>
> ---
>
> > Show baseline results for each type of error. A measure of how difficult certain errors would help future work.
>
> We placed this information initially in the supplementary section of the paper and, following your suggestion, transferred it to the main paper.

---

### Official Review · Reviewer_81qP · 2023-11-03

**Soundness:** 2 fair
**Presentation:** 2 fair
**Contribution:** 1 poor
**Rating:** 3
**Confidence:** 4

**Summary:**

This paper introduces a new dataset for Error Recognition in procedure videos. The dataset consists of 384 videos (~94.5 hours) capturing 8 subjects on 10 kitchens, while the subjects are cooking 24 different recipes. The dataset is provided with other modalities such as depth, IMU, camera trajectories, however, baseline experiments use only RGB. The paper provides 3 sets of baselines: Error Recognition (supervised and early prediction), Multi-Step Localization, and Procedure Learning. In term of dataset contribution, the proposed dataset does not provide anything significantly different from previous ones (see detailed explanation in weakness). In term of experiments, the provided baselines do not bring any interesting insights about the new datasets. Written presentation is fair, but not great, some parts are unclear.

**Strengths:**

- The paper dedicates efforts to build a new benchmark for procedural videos, any effort in dataset building is a great contribution to the community.

**Weaknesses:**

1. The proposed dataset has nothing standing out from previous ones. By looking at Table 1, the proposed dataset are bucked in small or medium compared with the existing ones in terms of number of videos or number of hours, and even number of tasks. The only difference may be that the proposed dataset provides more modalities. However, experiments shown in the paper only used RGB so far. Another claim made by the paper is that it has more error instances (compared by error:normal ratio) than Assembly101, which is true. However, the ways of capturing error videos have some problems: in three scenario of capturing error videos (sec 3.1.2), the first twos were scripted, the last one is instructed. Although it helps providing more error videos, however, those error videos will be intentional (the mistakes won't look realistic). In practice, the unintentional mistakes or errors are more relevant and often happened in reality.
2. The baselines provide not much insights.
- For Error Recognition (Table 2 & 3), the observation is Omnivore is the best backbone / embedding for error / normal video classification. The early prediction problem is still formulated as classification with partly-observed data. The same finding is confirmed (Omnivore works best for this, this brings no surprise as both are technically the same problem, the later one is a bit harder).
- For Multi-Step Localization, the problem is formulated as supervised TAL and the same set of features are used and a ActionFormer head was use for localization. The same finding is that Omnivore works best.
- For procedure learning: Two baselines (Dwibedi et al. 2019 and Siddhant et al. 2022) were used and provided similar performance. No real insights were observed in this experiments.
- Since this paper is about the new dataset which is claimed to focus on error recognition, however there not much new insights about the significance of bringing more error videos to procedural video dataset: neither in the way data is captured or significant baselines, experiments to demonstrate why it matters?

3. Writing is not clear in some parts
- In 4.2, is the supervised TAL trained task-agnostically or task-specifically, meaning TAL is trained for all or per tasks (24 recipes)

**Questions:**

- Minor comments
Section 4.2: what does "refer to 16" mean?

---

> ### Author Response · Authors · 2023-11-23
> **Author response to Reviewer 81qP**
>
> > The proposed dataset has nothing standing out from previous ones.
>
> Firstly, Error recognition as a task is quite challenging, and we conjecture that it requires the development of techniques that understand the context, meaning, and cause of various errors. Secondly, we respectfully question the statement the proposed dataset has nothing standing out from previous ones. We justify the statement with the following reasons
>
> - **Domain:** We stress the fact that Assembly 101 explores the domain of Assembly and Disassembly tasks, and we focus on Cooking activities.
>   - In assembly, the majority of the errors performed can be corrected, but cooking as a domain includes activities which do not have corrective actions.
>   - Unlike assembly, where shapes and colours of the objects remain fixed during the whole procedure. Cooking involves transformations in the shape or colour of ingredients.
> - **Environment:** Assembly 101 is a dataset that is collected in a lab environment, and we collect our dataset in real kitchen environments.
> - **Modalities:**  As we primarily employ HoloLens2 in capturing the data, we leverage the wealth of information it enables to capture. For example, our synchronized dataset enables the complete reconstruction of a 3D map of the environment while the activity is being performed.
>
> In conclusion, we emphasize that the category of errors we explore is different from the category of errors explored by activities in the assembly domain.
>
> ---
>
> > However, the ways of capturing error videos have some problems
>
> We respectfully deny this statement.
> - **Intentional videos based on scripts**
>   - It has been a common practice to use scripted videos for activity understanding. Popular dataset [1] is constructed based on pre-prepared scripts.
>   - Since the submission of this manuscript, two works have been published in a similar direction, and both explored assembly tasks [2-3] and both used pre-prepared scripts to capture data.
> - **Unintentional videos**
>   - Although participants followed scripts to perform activities. Due to the complex nature of cooking tasks and the participants' lack of experience in cooking, we do capture unintentional errors.
>   - We have updated the following sections of the manuscript to explicitly state this information Table 1, Figure 3 and Section 3.1.2
> - **Visual Appearance**
>   - It is important to note that the visual outcome of the majority of the errors remains the same i,e independent of errors generated intentionally by following a script or unintentionally.
>
> ---
>
> > The baselines provide not much insights.
>
> Error recognition and robust multi-step localization with errors are harder tasks which require the development of techniques that semantically understand the environment, perceptually identify the changes in the state of the activity and infer the cause. We do understand that the provided baselines are preliminary, but they provide a starting point for the development of task-specific methods.
>
> The purpose of this manuscript is to explore and categorize common errors that could occur in the cooking domain and follow the convention of evaluating the dataset using task-specific (error-recognition and multi-step localization) transfer learning techniques.
>
> ---
>
> > Writing is not clear in some parts
>
> As stated in the paper, we extracted features from 4 pre-trained models, used three criteria to construct training, validation and test splits and trained 12 models. In all these scenarios, we train models task-agnostically.
>
> ---
>
> > Minor comments Section 4.2: what does "refer to 16" mean?
>
> It is a typo. Thanks for catching it.
> It should be \textit{refers to Figure 16}, where figure 16 is from the supplementary material.
>
> ---
>
>
>
> - [1] Hollywood in homes: Crowd-sourcing data collection for activity understanding. In European Conference on Computer Vision, 2016.
> - [2] IndustReal: A dataset for procedure step recognition handling execution errors in egocentric videos in an industrial-like setting 2024.
> - [3] Weakly-Supervised Action Segmentation and Unseen Error Detection in Anomalous Instructional Videos.ICCV 2023

---

### Meta-Review · Area_Chair_TRqr · 2023-12-09

**Metareview:**

The paper received overall negative ratings (3, 5, 5). The main criticism was that the proposed dataset does not bring significant extra value to warrant publication compared to existing ones, especially Assembly101, which also contains mistakes and is significantly larger. Another concern was that, while the authors claim the extra modalities captured as one of the strengths, the paper does not demonstrate their value in error detection. There were additional concerns, mainly raised by reviewer 'bmra' that are valid. The authors provided a rebuttal, re-emphasizing the unique aspects of the proposed dataset and mentioning that they will incorporate extra modalities in future versions of the paper. After carefully reading the reviews and the rebuttal, this meta-reviewer agrees with the reviewers that the significance of the submission—in terms of both dataset and experimental analyses—isn't significant enough to warrant publication in this venue.

**Justification For Why Not Higher Score:**

I agree with the reviewers that the proposed dataset isn't a significant step forward compared to existing ones, and the extra value it brings  isn't significant enough to warrant a publication.

**Justification For Why Not Lower Score:**

N/A

---

### Decision · Program_Chairs · 2024-01-16

Reject